# PARTNR: A Benchmark for Planning and Reasoning in Embodied Multi-agent Tasks

**Matthew Chang, Gunjan Chhablani, Alexander Clegg, Mikael Dallaire Cote, Ruta Desai, Michal Hlavac, Vladimir Karashchuk, Jacob Krantz, Roozbeh Mottaghi, Priyam Parashar, Siddharth Patki, Ishita Prasad, Xavier Puig, Akshara Rai, Ram Ramrakhya, Daniel Tran, Joanne Truong, John M. Turner, Eric Undersander, Tsung-Yen Yang**

Work done at FAIR Meta, Alphabetical author order

https://aihabitat.org/partnr/

## ABSTRACT

We present a benchmark for Planning And Reasoning Tasks in humaN-Robot collaboration (**PARTNR**) designed to study human-robot coordination in household activities. **PARTNR** tasks exhibit characteristics of everyday tasks, such as spatial, temporal, and heterogeneous agent capability constraints. We employ a semi-automated task generation pipeline using Large Language Models (LLMs), incorporating simulation-in-the-loop for the grounding and verification. **PARTNR** stands as the largest benchmark of its kind, comprising 100,000 natural language tasks, spanning 60 houses and 5,819 unique objects. We analyze state-of-the-art LLMs on **PARTNR** tasks, across the axes of planning, perception and skill execution. The analysis reveals significant limitations in SoTA models, such as poor coordination and failures in task tracking and recovery from errors. When LLMs are paired with *real* humans, they require 1.5x as many steps as two humans collaborating and 1.1x more steps than a single human, underscoring the potential for improvement in these models. We further show that fine-tuning smaller LLMs with planning data can achieve performance on par with models 9 times larger, while being 8.6x faster at inference. Overall, **PARTNR** highlights significant challenges facing collaborative embodied agents and aims to drive research in this direction.

## 1 INTRODUCTION

Imagine a robot that collaborates with humans in daily activities, akin to human-to-human interactions. This scenario requires two key features: dynamic collaboration between the robot and the human, and the use of natural language for interaction. Current benchmarks in embodied AI satisfy one or the other condition; either robots operate in isolation (Shridhar et al., 2020; Krantz et al., 2020; Majumdar et al., 2024), or tasks are not specified in natural language (Yenamandra et al., 2023; Puig et al., 2024; Szot et al., 2023). Despite significant progress in the field of embodied AI, there remains a gap in realistic benchmarks that evaluate robots in collaborative settings. To bridge this gap, we introduce Planning And Reasoning Tasks in humaN-Robot collaboration (**PARTNR**), a novel benchmark that evaluates the ability of embodied AI agents to collaborate with humans across a range of household activities in simulated indoor environments (Figure 1).

**PARTNR** consists of 100,000 natural language instructions paired with tailored evaluation functions, focusing on four task types: (1) constraint-free, where sub-tasks can be completed in any manner by either agent, (2) spatial tasks that contain spatial constraints, (3) temporal tasks that require ordered execution, and (4) heterogeneous tasks that include actions that cannot be completed by one of the agents. Beyond the conventional challenges of long-horizon planning, partially observed environments, and large state and action spaces, **PARTNR** emphasizes the need for effective collaboration.

Curating such a benchmark of large-scale, natural language tasks with tailored evaluation functions presents significant challenges. Current benchmarks typically rely on either templated tasks (Shridhar et al., 2020; Zhang et al., 2024a) or tasks and evaluations crafted by humans (Li et al., 2023a), restricting the diversity or scale of the datasets. Instead, we propose a semi-automated approach using

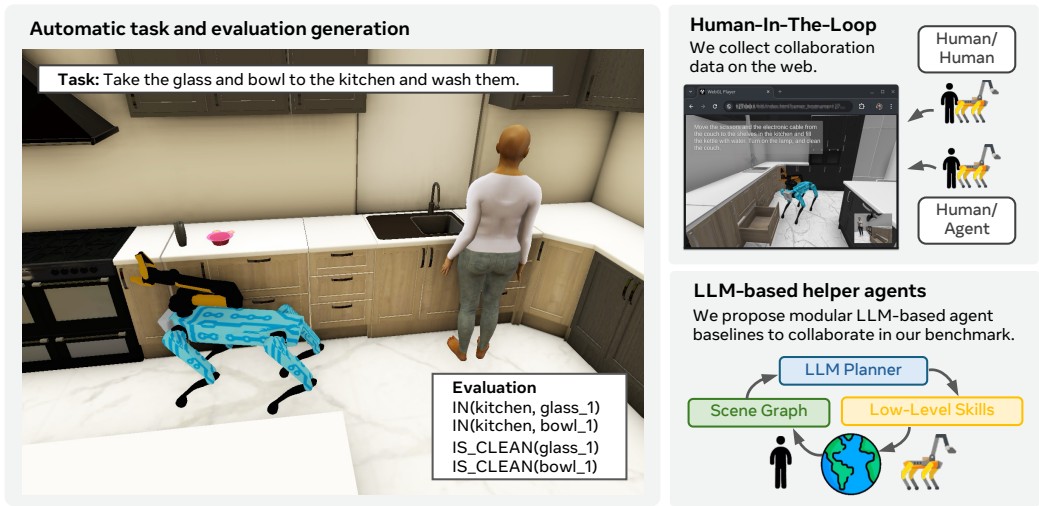

Figure 1: We present **PARTNR**, a benchmark for planning and reasoning in embodied multi-agent tasks, featuring 100,000 everyday tasks and evaluation functions generated semi-automatically, spanning 60 houses and 5,819 unique objects. We analyze LLM-based planning agents and also provide a human-in-the-loop tool to evaluate how agents collaborate with real humans.

Large Language Models (LLMs) with simulation-in-the-loop grounding. First, an LLM generates task and evaluation functions, which are grounded in the objects and furniture of a simulated house. Next, we employ simulation-aided filtering to remove hallucinations, complemented by human annotation to enhance diversity and accuracy. Subsequently, a set of 1,000 verified instructions and evaluation functions are utilized to guide an LLM through in-context prompting to create 100,000 tasks.

As **PARTNR** consists of natural language tasks and LLMs have shown strong results in planning (Yao et al., 2023; Ahn et al., 2022; Huang et al., 2022), we explore prompting and fine-tuning LLMs, to assess their effectiveness in collaborative scenarios. We study the effect of observability of the environment (i.e., full or partial), centralized vs. decentralized multi-agent control, learned or privileged oracle robot skills, and different ways of grounding the 3D world information for LLM-based planning. Beyond these automated evaluations with *synthetic* human partners, we also perform evaluations with *real* humans-in-the-loop, with people performing the task alone, with a human partner, or with an LLM-guided robot partner. Overall, we find that LLMs struggle at coordination, task tracking and dealing with perception and skill errors. While humans are able to solve 93% of **PARTNR** tasks, SoTA LLMs can only successfully complete 30% under non-privileged conditions. Moreover, in decentralized multi-agent settings, task completion takes 1.3x more steps than single-agent, due to poor tracking of partner actions, resulting in extraneous actions. In contrast, human pairs outperform single humans, in our human-in-the-loop experiments, highlighting potential for improving LLM collaboration strategies. LLMs also struggle to recover from skill failures and perception grounding errors, resulting in lower performance when privileged skills and privileged perception are removed. When comparing model sizes, we observe that a smaller fine-tuned Llama3.1-8B achieves a similar performance to a Llama3.1-70B without finetuning, while being 8.6x faster. This faster inference plays an important role when evaluated with real humans-in-the-loop, where the finetuned model takes fewer steps and offloads more tasks from the human.

In summary, **PARTNR** enables reproducible, large-scale, and systematic evaluations of embodied agents in a wide variety of collaborative scenarios. Through systematic evaluation, we reveal critical insights into the current limitations of LLM-based planners, opening interesting future research directions. All code, datasets, and human demonstrations on **PARTNR** tasks will be open-sourced.

## 2 RELATED WORK

**Language-based benchmarks in Embodied AI.** A large body of work on language benchmarks in Embodied AI has focused on navigation (Anderson et al., 2018; Krantz et al., 2020; Chen et al., 2019)

| | Environment | Multi-Agent | Language | Action Space | Task Types | Num tasks |
|---|---|---|---|---|---|---|
| Overcooked (Carroll et al., 2019) | 2D | ✓ | | HL | C | 4 |
| RoboGen (Wang et al., 2024) | 3D | | ✓ | LL+HL | CST | 106 |
| GenSim (Katara et al., 2023) | 3D | | ✓ | LL | CS | 100 |
| RoCo (Mandi et al., 2024) | 3D | ✓ | ✓ | LL | CS | 6 |
| FurnMove (Jain et al., 2020) | 3D-S | ✓ | | LL | C | 30 |
| RoboCasa (Nasiriany et al., 2024) | 3D-S | ✓ | ✓ | LL | CST | 100 |
| ALFRED (Shridhar et al., 2020) | 3D-S | | ✓ | HL | CST | 25,743 |
| BEHAVIOR-1K (Li et al., 2023a) | 3D-M | | | LL+HL | CST | 1,000 |
| WAH (Puig et al., 2021) | 3D-M | ✓ | | HL | C | 1,211 |
| Co-ELA (Zhang et al., 2024a) | 3D-M | ✓ | ✓ | HL | C | 44 |
| **PARTNR** | 3D-M | ✓ | ✓ | LL+HL | CSTH | 100,000 |

Table 1: **Comparison to similar embodied benchmarks.** We compare **PARTNR** to embodied AI benchmarks, focusing on natural language and multi-agent collaboration tasks. Comparison axes are – **Environment:** Household single room (S), household multi-room (M). **Action Space:** High-Level Actions (HL), Low-level Actions (LL). **Task Types:** Constraint-free (C), Spatial (S), Temporal (T), Heterogeneous (H) **Num tasks:** We measure tasks as the number of unique scene-goal pairs.

or Embodied Question Answering (Das et al., 2018; Majumdar et al., 2024) which involve navigation and information gathering but do not require agents to modify their environments. Closer to our work are instruction-following benchmarks (Shridhar et al., 2020; 2021; Puig et al., 2018; Wang et al., 2024; James et al., 2020; Gong et al., 2023), where agents interact with environments to complete tasks described via language, though the diversity of tasks is limited. In contrast, we leverage LLMs to generate diverse task definitions and scene initializations, and extend them to *multi-agent* settings. The idea of scaling up task generation using LLMs has been explored in a few recent works (Katara et al., 2023; Wang et al., 2024; Xian et al., 2023; Nasiriany et al., 2024). However, these works tend to focus on single-agent tasks that span relatively short horizons, while we consider long-horizon, multi-agent problems. Table 1 compares relevant benchmarks with **PARTNR**.

**Embodied multi-agent benchmarks.** Multiple works have proposed embodied multi-agent benchmarks (Puig et al., 2023; Agashe et al., 2023; Zhang et al., 2024a; Jain et al., 2019; Suarez et al., 2019). Many of these benchmarks focus on coordination in simple 2D environments, limiting their applicability to real world settings (Agashe et al., 2023; Carroll et al., 2019). Recent works have developed benchmarks studying collaboration in more realistic environments and activities (Puig et al., 2021; Zhang et al., 2024a; Jain et al., 2019; Puig et al., 2024; Szot et al., 2023), focusing on rearranging objects or furniture in large, partially observable 3D environments (Puig et al., 2021; 2024; Jain et al., 2019; Szot et al., 2023), or manipulating objects in a counter-top space (Mandi et al., 2024). However, these benchmarks are typically limited to a predefined and reduced set of tasks, often not described in natural language and primarily involving object rearrangement. In contrast, **PARTNR** covers an open set of tasks, each described in natural language, requiring agents to rearrange objects with spatial and temporal constraints, as well as requiring heterogeneous actions that can only be done by the human agent, (e.g., washing dishes or turning on the oven).

**LLMs for decision making.** Several works use LLMs as interactive policies, highlighting challenges in grounding them with observations and actions (Huang et al., 2022; Yao et al., 2023; Ahn et al., 2022; Huang et al., 2023; Zeng et al., 2022; Zheng et al., 2023; Guo et al., 2024; Liu et al., 2024). Some approaches improve grounding by prompting LLMs with demonstrations and task-specific constraints (Huang et al., 2022; Yao et al., 2023), or by integrating LLMs with external modules for multi-modal reasoning (Ahn et al., 2022; Huang et al., 2023; Zeng et al., 2022). Toolformer (Schick et al., 2023) allows LLMs to call APIs for information retrieval or environmental interaction. For instance, APIs can be used to call low-level policies (Driess et al., 2023), to leverage VLMs for obtaining the state of the world (Huang et al., 2023; Zhang et al., 2024b), or to another LLM serving as a world model (Zhao et al., 2024). SayPlan (Rana et al., 2023) maintains a persistent graph of the current world-state (Gu et al., 2024; Werby et al., 2024), enabling detailed semantic and geometric queries. Our work synthesizes these ideas by encoding the environment into a graph, using tools to extract relevant information, and executing tasks through motor skills. Another line of work fine-tunes LLMs with data from the target environments, by learning input and output adaptors (Li et al., 2022; Szot et al., 2024; Xiang et al., 2023). We explore low-rank adaptation of LLMs with multi-agent data to enhance coordination and efficiency. While fewer studies focus on LLMs in multi-agent collaboration (Zhang et al., 2024a; Park et al., 2023; Li et al., 2023b; Zhou et al., 2024), one notable example is CoELA (Zhang et al., 2024a) that design collaborative agents, though limited in task

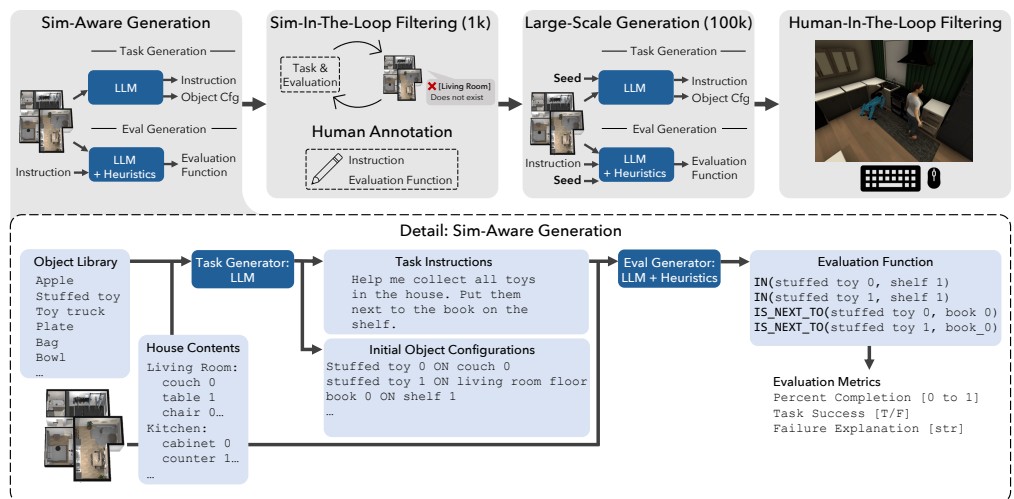

Figure 2: **The PARTNR generation pipeline.** Task and evaluation generators produce episodes, which are filtered and annotated for correctness. These episodes are then treated as seeds to achieve 100k-scale. Finally, episodes are vetted during human-in-the-loop collection.

diversity. Our work addresses a broader range of tasks and considers agents with varying capabilities, pushing the boundaries of multi-agent system collaboration in more complex scenarios.

## 3 BENCHMARK GENERATION

We introduce **PARTNR**, a benchmark aimed at training and evaluating robots at solving natural language tasks in collaboration with humans. **PARTNR** covers four types of tasks: (1) Constraint-free tasks, where sub-tasks can be completed in any manner by either agent. For example, *"Let's move all dirty plates to the sink."* (2) Spatial tasks that require reasoning about the spatial positioning of objects. For instance, *"Let's place the books on the shelf **next to each other**."* (3) Temporal tasks, where the sequence in which sub-tasks are executed is important. For example, *"Let's remove the candles from the dining table **before** bringing the plates to the table."* (4) Heterogeneous tasks, involving actions that are beyond the robot's capabilities. For example, *"Let's **wash** the dishes before putting them in shelves."* In scenarios where the robot's skills do not support washing, completing this task requires reasoning about agent capabilities. Our benchmark consists of natural language instructions and corresponding evaluation functions, both of which are generated at-scale using LLMs. Specifically, we generate 1,000 human-verified instructions and corresponding evaluation functions and use them as in-prompt examples to scale to 100,000 tasks in other scenes with different layouts and objects. A unique aspect of our automatic generation is the integration of an embodied simulator within the generation loop, which significantly reduces LLM errors such as hallucinations and infeasible actions.

**Task Instruction**

*"Let's tidy up the family room. The toys go in the toy box. After that, set the plants on the shelf next to each other."*

**Evaluation Function**

```
Propositions:
0  is_inside(["toy_fire_truck_0"], ["toy_box_0"])
1  is_inside(["toy_food_0"], ["toy_box_0"])
2  is_on_top(["plant_0"], ["shelf_0", "shelf_1"])
3  is_on_top(["plant_1"], ["shelf_0", "shelf_1"])
4  is_next_to(["plant_0"], ["plant_1"])

Dependencies:
0  WhileSatisfied([4], depends_on=[2,3])
```
*Verify "is_next_to" when the plants are placed*
```
Constraints:
0  TemporalConstraint(
       [(0,2), (1,2), (0,3),
        (1,3), (0,4), (1,4)]
   )
```
*The toys must be rearranged first*
*All placements are terminal*
```
1  TerminalSatisfactionConstraint([0,1,2,3,4])
```

Figure 3: **Task and evaluation example.** Language tasks have inherent complexity and ambiguity; both of which are supported by the structures of our evaluation functions.

### 3.1 SIMULATION-IN-THE-LOOP TASK INSTRUCTION GENERATION

While LLM-based task generation has been studied in literature before (Katara et al., 2023; Wang et al., 2024; Xian et al., 2023; Nasiriany et al., 2024), these generations are not grounded beyond

user-created in-context prompts. In **PARTNR**, we use a simulation-in-the-loop generation technique to ground the LLM in the environment, agents and available actions. Specifically, we instantiate a simulation environment in the Habitat 3.0 simulator (Puig et al., 2024), populated with the HSSD dataset (Khanna et al., 2024), consisting of 60 unique houses and 5,819 OVMM objects (Yenamandra et al., 2023). The simulated house is parsed into a list of rooms and available furniture, and passed to an LLM, along with all available objects. Using this information, the LLM is asked to generate free-form, viable tasks in the scene, along with an initial scene state description. For example, if the generated task is *"Clear dishes from the living room"*, the LLM should generate an initial scene with multiple dishes in the living room. At this stage, additional objects are also added to the scene to create clutter in the environment. Once generated, the tasks, initial states, and clutter are instantiated in the simulator, and infeasible instructions are filtered. For example, if the house does not have a living room, *"Clear dishes from the living room"* is invalid. Similarly, if the generated task requires actions not supported by the simulator, such as folding, the task is filtered. Generally, the rate of hallucinations is high, leading to a significant number of episodes being discarded. We observe that after filtering for infeasible instructions, the diversity in generated instructions is typically limited. For example, most of the instructions use the same objects (e.g., dishes) or similar rooms (e.g., kitchen or dining room). To increase diversity of the generated tasks, we manually annotate them to ensure task and object diversity, such as maintaining a balanced distribution of constraint-free, spatial, temporal, and heterogeneous tasks by modifying the instructions to elicit specific characteristics. This process results in 1,000 human annotated and simulation-verified tasks (Appendix A.5.2).

Such manual annotation is not practical for large-scale generation. Instead, we leverage the human-annotated 1,000 instructions to scale generation by using them as in-prompt examples. We prompt the LLM with both a house description and an example task, and instruct it to modify the task to fit the new house. For example a task like *"Clear all dishes from the living room"* is modified to *"Clear all toys from the bedroom."* This allows us to maintain the diversity of the original annotated instruction set, while ensur-

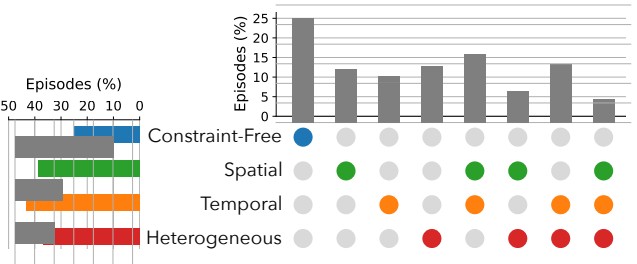

Figure 4: **Distribution of task types in PARTNR.** The left plot displays the percentage of tasks with each characteristic. Constraint-free tasks by definition exclude the other types. The top right bars correspond to the dot combination below.

ing a high likelihood of successful instantiation in the simulator. Qualitatively, we filtered or edited $\sim 90\%$ of free-form generated instructions and only $\sim 10\%$ of scaled instructions. We use *LLama3-70B-Instruct* (Dubey et al., 2024) for all instruction generation. Finally, all tasks go through a human-in-the-loop filtering. In this step, humans attempt to solve the tasks using our human-in-the-loop tool (Appendix A.12) and eliminate physically infeasible instructions that are difficult to detect, such as requiring an object to be at two locations. Figure 2 provides an overview of our pipeline. Details on the generation process can be found in Appendix A.3 and prompts in Appendix A.13.

## 3.2 EVALUATION FUNCTION GENERATION

To determine if an agent successfully completed the instruction *"Clear all dishes from the living room"*, we need an evaluation function that can validate the removal of all spoons, forks, and other dishes from any of the living rooms. However, manually annotating all necessary rearrangements and state changes of a task is time intensive and since all tasks are unique, impractical at scale. Similar to instruction generation, we employ an LLM to create an evaluation function that assesses task completion without requiring any manual annotations. Specifically, we leverage the ability of LLMs to generate predicate-based Python programs using three types of APIs: a list of **propositions** indicating *what* relations between entities must be satisfied, a set of **dependencies** indicating *when* propositions should be queried, and a set of **constraints** indicating *how* propositions must be satisfied. We define an expressive vocabulary of each of these components to afford evaluation of all tasks in the benchmark (e.g., Figure 3). Closely related evaluation systems include defining tasks in PDDL (Ghallab et al., 1998) or BDDL (Srivastava et al., 2022). We choose to build a new Python-based evaluation system since neither have the expressivity to evaluate **PARTNR** tasks while maintaining human and LLM interpretability; for instance, BDDL does not support time-varying evaluation. Since **PARTNR** tasks have temporal dependencies (e.g. multi-step rearrangement), the input to the evaluation function is

the complete sequence of simulator states during task execution. The evaluation function returns 3 metrics: (1) Percent Complete ($PC \in [0, 1]$), the % of propositions that are satisfied w.r.t. constraints, (2) Success ($S \in \{True, False\}$), measuring if a task was successfully completed, defined as $S := (PC = 1)$, and (3) Failure Explanation ($FE$), a human and LLM interpretable language description of why the agents failed to accomplish the task. See Appendix A.4 for details.

We use *CodeLLama-70B-instruct* (Roziere et al., 2023) for evaluation function generation. Exemplified in Figure 3, producing perfect evaluation functions is non-trivial. The LLM must correctly classify the entire space of possible actions against natural language instructions and the specific simulation environment, which can be quite complex. For example, in Figure 3, the instruction *"set the plants on the shelf"* refers to *"the shelf"*, but two shelves exist in the room. The evaluation function must allow either shelf while requiring placement of all plants, and finally account for a next-to relation. Any error or missing value in either a proposition or constraint invalidates the evaluation function. Consequently, we observe a large error rate in LLM generation, particularly pertaining to incorrect propositions and temporal sequencing constraints.

To alleviate these inaccuracies, we follow a similar semi-automated procedure to instruction generation. We first generate evaluation functions for the 1,000 human-annotated instructions and perform manual annotation to correct them (Appendix A.5.3). This results in a dataset of 1,000 human-verified instruction and evaluation pairs. Next, we generate evaluations for the scaled 100,000 instruction set. Recall that the scaled instructions are generated by prompting the LLM with an example instruction from the annotated set. We retrieve the corresponding annotated evaluation function and prompt the LLM with it. This is similar to approaches such as retrieval-augmented generation (Lewis et al., 2020) and improves the accuracy of evaluation function generation from 50% to 92% as measured through manual inspection (Appendix A.5.2). As a final step, we ask human users to solve all **PARTNR** tasks using our human-in-the-loop evaluation tool (Appendix A.12). All tasks that cannot be solved by humans over 6 retries (3 single-user, 3 multi-user tries) are deemed infeasible, and removed from the dataset. We find that about 90% of instructions, and 92% of evaluation functions from automated generation are accurate, resulting in a combined generation accuracy of $90 \times 92 = 83\%$ from automated generation, and 98% after human annotation and filtering.

### 3.3 THE **PARTNR** DATASET

The **PARTNR** dataset comprises of 100,000 episodes in 37 train scenes, 1,000 episodes in 13 validation scenes, and 1,000 episodes in 10 test scenes from the HSSD dataset (Khanna et al., 2024). After scaled generation, all validation and test set episodes are human annotated for correctness, as well as a 2,000-episode subset of train. See Appendix A.5.1 for correctness analysis of scale-generated episodes. Below, we analyze the characteristics and diversity of this dataset.

**Characteristics**: As described earlier, **PARTNR** focuses on four task types: constraint-free, spatial, temporal, and heterogeneous. We show the distribution of these task types in the test split in Figure 4; validation split is similar. **PARTNR** evaluates collaboration along these axes both independently and jointly. Secondary characteristics of interest include dependent rearranges (e.g., *"Place them on the **same** table"*) and multi-step rearrangement of the same object (e.g. *"Move the cup to the **sink**, wash it, then place it in the **cabinet**"*). 7% of tasks include dependent rearranges and 6% include multi-step rearrangement. Tasks average 4.7 propositions to be satisfied (indicative of number of steps required to complete tasks). For analysis of linguistic phenomena and more characteristics, see Appendix A.2.

**Diversity**: The diversity of tasks in **PARTNR** is largely enabled by simulation-in-the-loop generation, which utilizes rich HSSD scenes, and the OVMM object set. Consequently, **PARTNR** tasks reference and require reasoning about 155 unique object types, 20 furniture classes and 13 room types. Note that each instruction, instantiated in each house, brings its own diversity. For example, *"move the laptop to the office table"*, grounds *office* and *table* uniquely in each house, as well as different instances of *laptop* in different instructions. Further discussion can be found in Appendix A.2.

## 4 EXPERIMENTS AND ANALYSIS

We investigate how state-of-the-art planning and perception methods handle natural language tasks in new environments and coordinate with unseen partners using **PARTNR**. Since **PARTNR** consists of diverse spatio-temporal tasks specified in language, we primarily use LLMs in our baselines for planning, and study variants in (1) zero-shot prompting, retrieval-augmented generation or fine-tuning,

(2) centralized versus decentralized planning, (3) partially versus fully observed environment, (4) learned versus oracle low-level robot skills, and (5) privileged versus non-privileged perception.

Our experiments are conducted in the Habitat 3.0 simulator (Puig et al., 2024) with a simulated Spot robot (BostonDynamics; Yokoyama et al., 2023). We adopt a two-layer hierarchical control architecture, similar to (Puig et al., 2024; Szot et al., 2021), as illustrated in Figure 5, for the robot and simulated human. At the high level, a planner selects skills from a predefined skill library (e.g., navigate, pick, place, open, close). We also use a textual world graph with a three-layer hierarchy representing rooms, furniture, and movable objects. Each node in the graph stores a semantic category (e.g., kitchen, table or cup), 3D information (e.g., position or bounding box), and states (e.g., clean, powered on). See Appendix A.6 and Figure 10 for details.

## 4.1 BASELINES

We evaluate baselines along the following axes:

1. **Variations of high-level planner:**
   - Heuristic expert: This approach utilizes expert-designed heuristics and privileged information about the task, environment and evaluation function to pre-plan all steps for human and robot based on their capabilities. For instance, both agents might rearrange objects, but only humans perform cleaning, filling, and toggling on/off tasks.
   - Zero-shot ReAct (ReAct): We use ReAct (Yao et al., 2023) with an API library of functions or *tools* that enable the LLM to take actions. As observation, we provide the LLM with a concise, current world graph description plus a history of actions. The LLM uses this information to choose an action from `[ExploreRoom, Navigate, OpenFurniture, CloseFurniture, PickObject, PlaceObject, Wait, Done]` for the robot. See Appendix A.15 for prompts and Appendix A.8 for API details (human and robot).
   - ReAct with Retrieval-Augmented Generation (ReAct-RAG): We also evaluate ReAct with RAG (Lewis et al., 2020) to investigate whether examples of planning on similar tasks improves the performance of ReAct. We construct a database of planning examples by collecting the successful traces from ReAct from a the 2,000 task training subset (see 3.3). During test time, the most relevant planning trace from the train dataset is selected based on sentence similarity and added to the LLM's prompt (Pang et al., 2024; Madaan et al., 2024).
   - Finetuned LLMs (Finetuned): We also investigate finetuning a smaller LLM (Llama3.1-8B) as our high-level planner, using successful traces from the ReAct baselines (Hsieh et al., 2023) that use Llama3.1-70B. Using the React-RAG dataset, we split every episode into a sequence of high-level planning actions, filtering for only actions that were executed successfully. For every action, we build an input containing the world-graph and history of actions, similar to ReAct (see Appendix A.9 for more details). We then finetune an LLM to predict the action from the ReAct episode given this input, using a low-rank adapter (Hu et al., 2021). This model has reduced latency and computational demands, suitable for real world deployment.

   All model generations are constrained to only output valid actions on observed objects using constrained generation (Geng et al., 2023). The constrained generation greatly reduces the hallucinations and 'grammatical' errors typical of LLMs. An episode is finished when both agents call `Done` or reach maximum simulation steps or LLM calls. Refer to Appendix A.8 for details.

2. **Centralized versus decentralized planning:** To study the overhead of coordination in multi-agent **PARTNR** tasks, we compare centralized and decentralized planners. In centralized, a single LLM decides actions for both agents, with complete information about both agent's states, effectively removing any need for coordination between the agents. In decentralized, each agent is controlled by a different LLM, and each LLM needs to reason about the other agent's actions.

3. **Partial versus full observability:** To evaluate if SoTA language models can explore new environments and identify task-relevant objects, we consider a partially observed setting where the planner knows the house's layout but not the object locations, requiring exploration. This is in contrast to a fully observed setting, where all object locations are known in advance.

4. **Learned versus oracle low-level robot skills:** We examine the impact of learned neural-network skills versus oracle skills (with privileged simulation information) on overall performance in **PARTNR** tasks. We create a library of learned skills for pick, place, navigate, open and close actions (Appendix A.7 provides more details) and compare performance with oracle skills.

5. **Privileged versus non-privileged perception:** To study perception challenges such as inaccurate detection and approximate localization, we used a non-privileged world graph with modified

ConceptGraphs (Gu et al., 2024), built from agents' RGBD observations only. As agents explore and take actions, this world graph is updated using onboard sensing (details in Appendix A.6). In contrast, with privileged perception, this information is available from the simulation.

## 4.2 RESULTS AND ANALYSIS

**Metrics.** We evaluate performance across different settings using four key metrics. First, the *simulation steps* metric measures the number of steps required for agents to complete the task within the simulation environment, serving as an indicator of efficiency. Second, the *success rate* reflects the completion of the task i.e. whether 'all' task constraints are satisfied. Given the complexity and long-horizon nature of **PARTNR** tasks, agents often partially complete the task. To account for this, we also report *percent complete*, which quantifies the ratio of completed task 'propositions' (percent complete = 1 for successful tasks). Lastly, we assess the reasoning efficiency of the planners through the *planning cycles* metric, which counts the number of high-level LLM calls each planner makes in the course of an episode. We cap the maximum planner calls at 50 in all experiments.

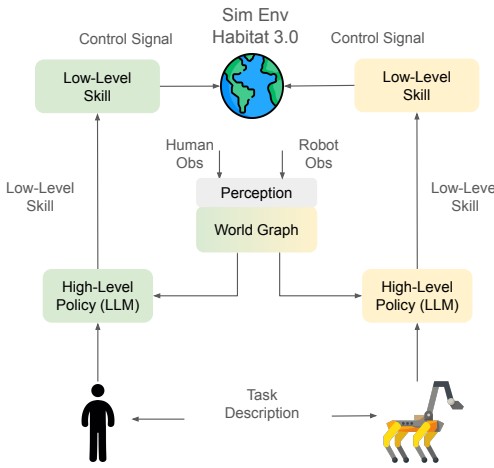

Figure 5: **Decentralized architecture.** The human and robot agents use a 2-layer hierarchical architecture, with high-level LLM planners that call low-level skills. Both agents build a world graph, updated using observations and actions.

### 4.2.1 TASK PERFORMANCE ANALYSIS

Table 2 presents a comprehensive evaluation of the planning approaches defined in Section 4.1 using the *Llama3.1-70B-Instruct* model (Dubey et al., 2024) for ReAct baselines, and a finetuned *Llama3.1-8B* base model for the Finetuned baseline. Since **PARTNR** tasks are multi-agent, we also need a simulated human partner, which we control with a ReAct approach, using *Llama3.1-70B-Instruct*. Our main findings are detailed below.

**PARTNR tasks are challenging for LLM-based planners.** LLM-based baselines across all observability and controllability conditions perform worse than the privileged heuristic expert, due to errors in task tracking (not completing all steps, performing them in the wrong order, or undoing completed steps), and semantic errors (placing objects on the wrong furniture, or moving the wrong object), indicating a gap in LLM task planning.

**LLMs struggle with coordination in decentralized settings.** Decentralized ReAct baselines which do not have privileged access to partner's intent are significantly slower at task completion than centralized ReAct (3295 steps with decentralized-partial in row(e) versus 2298 with centralized-partial in row(d)). This shows that reasoning about the partner e.g., knowing or inferring partner's intent can improve task efficiency in **PARTNR** tasks, and current SoTA LLMs perform poorly at this. Moreover, decentralized ReAct with two agents is even *slower* than ReAct with a single-agent (3295 steps with multi-agent in row(e) versus 2519 with single-agent in row(a)), indicating that LLMs suffer from a significant coordination "burden". This co-ordination burden is further highlighted in our analysis on *extraneous effort* in Section 4.2.2, where we find that agents end up repeating parts of the task or performing irrelevant actions with much higher frequency in decentralized settings.

**LLMs are unable to recover from learned skill failures.** When replacing oracle skills with learned skills, the success rate decreases from 0.73 to 0.57 (row(e) vs. row (h)). This decline can be attributed to the higher failure rate and increased number of simulation steps required by learned skills compared to privileged oracle skills. The LLMs struggle to recover from skill errors like failing to pick up an object or performing incomplete exploration, resulting in a lower success rate. Future research could investigate training large models with low-level skills in the loop, enabling them to learn recovery and replanning strategies in the face of such failures.

**LLMs exhibit a high degree of sensitivity to errors in perception.** When we replace privileged perception with off-the-shelf perception modules, success rate significantly declines (from 0.57 with a privileged, partial world graph in row(h) to 0.30 with Concept-Graphs (Gu et al., 2024) in row(i)). LLMs rely heavily on accurate world descriptions provided by the world graph and struggle to correct

| Method | Controllability | Skills | Observability | Sim Steps ↓ | Success Rate ↑ | Percent Complete ↑ | Planning Cycles ↓ |
|---|---|---|---|---|---|---|---|
| (a) ReAct-Single | Single Agent | Oracle | Partial | 2519.02 ± 57.48 | 0.73 ± 0.01 | 0.85 ± 0.01 | 18.68 ± 0.33 |
| (b) Heuristic-Expert | Centralized | Oracle | Full | 1260.88 ± 26.97 | 0.84 ± 0.01 | 0.94 ± 0.01 | N/A |
| (c) ReAct | Centralized | Oracle | Full | 1347.43 ± 33.80 | 0.74 ± 0.01 | 0.88 ± 0.01 | 17.49 ± 0.34 |
| (d) ReAct | Centralized | Oracle | Partial | 2298.13 ± 61.39 | 0.74 ± 0.01 | 0.85 ± 0.01 | 20.73 ± 0.51 |
| (e) ReAct | Decentralized | Oracle | Partial | 3295.20 ± 76.27 | 0.73 ± 0.01 | 0.86 ± 0.01 | 15.24 ± 0.31 |
| (f) ReAct + RAG | Decentralized | Oracle | Partial | 3467.47 ± 82.39 | 0.71 ± 0.01 | 0.84 ± 0.01 | 14.75 ± 0.31 |
| (g) Finetuned | Decentralized | Oracle | Partial | 3228.96 ± 75.14 | 0.70 ± 0.01 | 0.84 ± 0.01 | 12.85 ± 0.24 |
| (h) ReAct | Decentralized | Learned | Partial | 6494.88 ± 181.52 | 0.57 ± 0.02 | 0.76 ± 0.01 | 22.72 ± 0.58 |
| (i) ReAct | Decentralized | Learned | ConceptGraph | 12490.80 ± 208.90 | 0.30 ± 0.01 | 0.56 ± 0.01 | 23.84 ± 0.45 |

Table 2: **Analysis of planner baselines in various settings.** We compare performance using simulation steps, success rate and percent complete on the tasks, and the average number of planning cycles used by the baselines (described in Section 3).

errors such as misclassification (e.g., shelves misidentified as tables) or incorrect room assignments (e.g., a table in the living room mislabeled as being in the bedroom). Multi-modal models like VLMs might be stronger at recovering from such failures, which we leave for future work.

**Finetuned 8B model performs on par with a ReAct with a 70B model, while being 8.6x faster.** We find that the finetuned planner with a small 8B model performs on par with ReAct, which uses a much larger 70B model (a 0.73 success rate with the 70B model in row(e), versus 0.70 with the finetuned 8B model in row(g)). At the same time, we find that the finetuned model is 8.6 times faster at inference. This indicates that the finetuning effectively distills task-relevant information from the training set and generalizes to new test tasks. When deployed with humans-in-the-loop, the finetuned model takes fewer steps and offloads more sub-tasks than the 70B model (see Table 3).

### 4.2.2 ANALYSIS OF COLLABORATIVE BEHAVIOR AND EFFICIENCY

Our analysis in Table 2 revealed challenges in LLM collaboration, prompting a deeper investigation into specific collaborative behaviors, explained below and detailed in Appendix A.11 and Table 12.

**Robots offload up to 60% of tasks.** We evaluate the robot's ability to offload tasks from the human, measuring the ratio of sub-tasks performed by the robot to the total sub-tasks in successful **PARTNR** tasks. Despite similar success rates between single- and multi-agent (0.73 vs. 0.74), the robot offloads about 60% of sub-tasks in decentralized multi-agent, reducing human effort (Table 12).

**Decentralized agents are prone to performing extraneous tasks.** The agents sometimes end up performing sub-tasks that are not useful for the task such as rearranging an object that is not required by the task or repeating a sub-task already performed by the other agent. To capture such extraneous agent effort, we measure the portion of agent actions that did not increase the percent complete metric i.e., did not contribute to task progress, over the total number of successful agent actions in an episode. We find a 300% increase in extraneous effort in decentralized multi-agent settings compared to single-agent (Table 12), indicating a significant coordination burden.

**Temporal and heterogeneous tasks are challenging for LLMs.** LLMs struggle in temporal and heterogeneous tasks. Task success drops by 27% for temporal tasks and 20% for heterogeneous tasks compared to constraint-free tasks for ReAct (Table 13). This highlights the limitations of LLMs in reasoning about agent capabilities and following strict ordering constraints.

### 4.3 HUMAN-IN-THE-LOOP EVALUATION

We build on the human-in-the-loop infrastructure from Habitat 3.0 (Puig et al., 2024) and adapt it to a server-client architecture, with the server hosted on AWS capable of supporting multiple clients (see Appendix A.12). This allows us to run at-scale evaluation of our tasks with 129 non-expert human participants. We collect single-user and multi-user data on 1000 tasks from the validation and test set using this tool. In the single-user setting, a single participant completes the whole task, by driving the human agent in the simulator via keyboard/mouse controls Figure 14 in appendix shows our HITL interface. In multi-user, two participants complete the task together by controlling a human and a robot agent. The goal of these experiments is to study multi-user dynamics at **PARTNR** tasks, and see if multiple humans collaborating are more efficient than single human. Finally, we run a human-AI experiment where a human participant collaborates with a robot controlled by an

| Method | Success Rate ↑ | Percent Complete ↑ | Sim Steps ↓ | Task Offloading ↑ | Exploration Efficiency ↓ | Extraneous Effort ↓ |
|---|---|---|---|---|---|---|
| Single-user | $0.93 \pm 0.01$ | $0.96 \pm 0.00$ | $3046.99 \pm 80.79$ | N/A | $2459.22 \pm 26.75$ | $0.09 \pm 0.01$ |
| Multi-user | $0.93 \pm 0.01$ | $0.96 \pm 0.00$ | $2369.55 \pm 49.33$ | $0.59 \pm 0.01$ | $1762.47 \pm 13.99$ | $0.15 \pm 0.01$ |
| Human-ReAct | $0.91 \pm 0.01$ | $0.96 \pm 0.02$ | $4267.71 \pm 83.40$ | $0.16 \pm 0.01$ | $2624.39 \pm 26.05$ | $0.12 \pm 0.01$ |
| Human-Finetuned | $0.92 \pm 0.01$ | $0.96 \pm 0.00$ | $3443.33 \pm 61.46$ | $0.26 \pm 0.01$ | $2164.94 \pm 21.31$ | $0.13 \pm 0.01$ |

Table 3: **Human-in-the-Loop Evaluation.** We evaluate the performance of a 2-person human team and human-LLM teams, comparing them to solo human performance on **PARTNR** tasks using metrics described in Section 4.1. Additional results and analysis in Appendix A.12.

LLM (using the ReAct and Finetuned models from Section 4.1). This experiment aims to evaluate LLM-controlled agents at collaborating with unseen, real humans. Table 3 shows the success rate (SR) and percent complete (PC) of tasks from the validation set in a single-user, multi-user, human-ReAct and human-Finetuned setting. Additionally, we measure the number of steps taken by each approach to complete the task, and the ratio of work completed by the robot i.e., task offloading. We also measure exploration efficiency in human-in-the-loop, by measuring the steps taken to pick the first object, and extraneous effort, indicating actions that were not useful for task completion. These results are summarized in Table 3. Some key findings are below (more results and analysis in A.12.4):

**Humans are significantly better than LLMs at PARTNR tasks.** In both single and multiple human environments, the success rate achieved is 0.93 on the **PARTNR** benchmark. In contrast, the ReAct model without any privileged information, achieves a significantly lower success rate of 0.30 (row (i) of Table 2). This highlights a significant gap in the performance of LLMs in planning tasks. Note that the LLM baselines like ReAct and Finetuned achieve a success rate of 0.92 and 0.91 when evaluated with *real* humans (Table 3), because humans are able to adapt to LLM mistakes. On the other hand, the simulated human in Table 2 is an LLM, which is unable to recover from partner mistakes.

**Finetuned LLMs perform better than ReAct when coordinating with real humans.** When deployed with real humans-in-the-loop, the finetuned model is faster than ReAct at task completion (3443 steps with finetuned versus 4267 with ReAct). It is also able to offload more tasks from humans than ReAct (26% with finetuned as compared to 16% with ReAct). This reflects that smaller models with faster inference can improve human experience in real-world deployment.

**LLMs struggle at coordination, hampering human performance.** Despite the Finetuned being faster than ReAct when collaborating with humans, both approaches are *slower* than the human doing the task alone. In contrast, two humans working together complete the task faster than a single human (2369 steps vs. 3046 with multi- and single-user respectively). This result is in line with the automated evaluation we observed in Table 1, where multi-agent LLMs are also *slower* than a single-agent LLM. This result further reinforces that LLMs suffer at coordination; while humans are able to coordinate and divide tasks between each other, decentralized LLMs are unable to do so.

**LLMs are able to offload tasks from humans.** Despite the aforementioned increase in the number of steps for task completion, robots guided by the finetuned model successfully offload 26% of tasks from humans. This indicates that LLMs can still offer assistance when collaborating with real human partners. Nonetheless, there remains significant potential for improvement.

## 5 CONCLUSION

We present **PARTNR**, a benchmark for reasoning and planning in multi-agent embodied tasks, featuring 100,000 natural language tasks instantiated in 60 simulated, multi-room houses with 5,819 unique objects. We use a semi-automated LLM-powered pipeline for large-scale instruction and evaluation function generation that uses simulation-in-the-loop grounding. **PARTNR** exhibits characteristics of everyday tasks, such as temporal and spatial constraints, and allows systematic evaluation of planning approaches. We find a significant gap between SoTA LLMs and human-level performance at **PARTNR** tasks. While our best LLM baseline only succeeds at 30% of tasks with no privileged information, humans are able to solve 93% of the tasks. Moreover, LLMs face challenges in coordinating with both LLM-based agents and real human partners. Human-in-the-loop evaluations, involving real humans collaborating with LLM-guided robots, reveal that LLM-guided partners decrease human efficiency compared to working solo. This suggests that LLM-based agents require significant improvements to become effective collaborative partners in embodied tasks. **PARTNR** serves as a challenging benchmark that highlights the substantial limitations of current models.

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

# A   APPENDIX

We present **PARTNR**, a benchmark for reasoning and planning in multi-agent embodied tasks, featuring 100,000 natural language, everyday tasks. We show at-scale generation of these tasks using LLMs with simulation in the loop for grounding and human in the loop for filtering. We also evaluate several LLM-based planning models on these tasks and highlight avenues for future work. This appendix provides additional details on these contributions and is organized as follows:

## A.1   OPEN-SOURCING **PARTNR** DATASET AND CODEBASE

Accompanying this paper, we will release the code and data necessary to reproduce our experiments. Released code includes our **PARTNR** benchmark tasks, metrics, baseline oracle skills, large planning model framework, and dataset generation utilities. Released data includes extensions of the Habitat Synthetic Scenes Dataset (HSSD)  (Khanna et al., 2024), generated benchmark task episodes, and model weights for our trained neural network skills and fine-tuned large planning model.

The publicly released codebase accompanying **PARTNR** will be contained in a public github repository and depend on the most recent version of the AI Habitat platform (habitat-lab and habitat-sim (v0.3.2)) (Puig et al., 2024) which it extends to define collaboration tasks and skills.

For the purpose of this submission, the anonymized code is included in the supplementary zip file. To preserve anonymity and respect supplement size limits, the data and assets required to reproduce the results are not included in the initial upload, but will be released along with the public code repository at a later date.

In order to model the space of rich indoor collaboration tasks we propose with **PARTNR**, we extended HSSD with additional asset authoring and annotation. To enable more realistic indoor object manipulation, we added articulated 3D furniture models such as drawers, cabinets, and appliances. These models were converted from rigid source assets in HSSD and swapped into the scenes. We prepared 60 scenes divided into train, val, and test splits to support our experiments. Each scene is manually adjusted by a human to ensure simulation robustness and minimize potential issues. Furniture is annotated with a set of Receptacles (surfaces which support small object placement such as shelves and drawers) and can be opened and closed by the agents. Receptacles are further filtered contextually in each scene to ensure that the active set is accessible to the agents. Additional annotations include point or marker sets for each furniture, region annotations, and semantic classification of objects. The marker sets indicate either a spread of surface points (for distance/occlusion checking) or the location of key points of interest such as faucets (for cleaning/filling) and handles (for opening/closing) necessary for low-level skill training and oracle skill execution. Region annotations included per-scene region volumes (e.g., kitchen, living-room, bedroom, etc.) for checking and specifying the location of objects and furniture. Semantic annotations indicate the object category or class (e.g. table, chair, cup, toy) to support open language prompt grounding and semantically guided task generation.

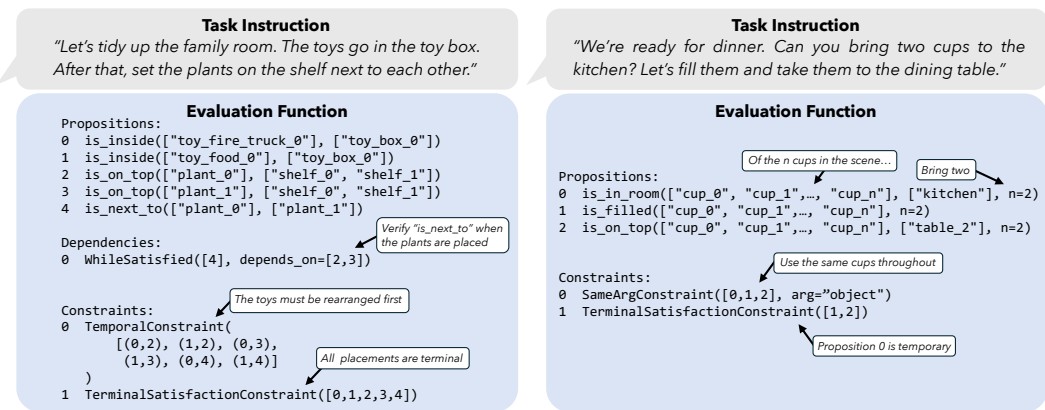

Figure 6: **Additional task and evaluation examples.** Expands on Figure 3 to show a task and evaluation containing subset counts and dependent rearranges (right).

## A.2 DATASET DETAILS AND ADDITIONAL ANALYSIS

We expand on the details and analysis of the **PARTNR** dataset of Section 3.3, including analysis of linguistic phenomena, secondary task characteristics, and the distribution of sampled entities in generated tasks. See Figure 6 for additional task and evaluation examples.

**Linguistic Phenomena.** In Table 4, we present an analysis of linguistic phenomena manually annotated over 50 random episodes (similar to Ku et al. (2020) and Chen et al. (2019)). We analyze the following phenomena:

- **Class Reference:** refers to a semantic class of objects, furniture, or rooms in the scene. These references typically, but not necessarily, follow from the classes defined in OVMM.
- **Instance Reference:** refers to a unique object, furniture, or room; this disambiguation between entities of a class is typically achieved by visual description.
- **Co-Reference:** an expression that refers to an entity defined elsewhere in the instruction.
- **Passive Voice:** the instruction is phrased such that the object is receiving the rearrangement or state change; the request is typically asked instead of commanded.
- **Active Voice:** the instruction is phrased such that the rearrangement or state change is to be completed with an object; the request is typically commanded instead of asked.
- **High-Level Goal Spec:** this sets the operating context for the task before the particulars of rearrangement or state change are specified.
- **Agentic Reference:** a reference to one of the agents performing the task. Typically used to incite a suggested task division between the human and robot.

We observe that **PARTNR** tasks have a high rate of entity class references, such as *the table*, (6.38/episode), and common occurrences of instance references and co-references. This signals a need for capable natural language understanding, scene grounding, and co-reference resolution. Tasks and sub-tasks are predominantly issued using active voice (92%), but 14% of tasks include at least one occurrence of passive voice. Half of tasks involve a high-level goal specification, which commonly serves to reduce the search space. For example, a task starting with *Let's clean up all the toys in the playroom* constrains object search to that room and softly constrains the placement of those objects to locations in which toys would commonly be stored. Finally, agentic references are present in 14% of episodes.

**Secondary Task Characteristics.** In Table 5, We present an analysis of secondary task characteristics present in **PARTNR** as derived automatically from evaluation functions of all episodes in our dataset. We find rare but present occurrences of subset counts, where agents must reason about manipulating a subset of a set of objects (e.g. *bring two cups...* when more than two cups exist in the scene). We find that every episode contains at least one occurrence of resolvable ambiguity, where an instruction makes an object/furniture/room reference that may be resolved by more than one entity instance.

| Linguistic Phenomenon | $p$ | $\mu$ | Example |
|---|---|---|---|
| Class Reference | 100 | 6.38 | *The table* |
| Instance Reference | 12 | 0.14 | *The coffee table* |
| Co-Reference | 50 | 0.64 | *That, Those, it, ...* |
| Passive Voice | 14 | 0.18 | *Can you bring me...?* |
| Active Voice | 92 | 2.40 | *Set it on the stool.* |
| High-Level Goal Spec | 50 | 0.50 | *Let's tidy up. Move...* |
| Agentic Reference | 14 | 0.16 | *Do... While I...* |

Table 4: **Analysis of linguistic phenomena in the PARTNR dataset.** $p$ is the % of instructions that contain the phenomenon while $\mu$ is the average number of times the phenomenon occurs within each instruction. A random sample of 50 Test episodes were included in manual annotation.

| Secondary Characteristics | $p$ | $\mu$ | Example |
|---|---|---|---|
| Subset Count | 1 | 0.01 | *Bring two cups...* |
| Resolvable Ambiguity | 68 | 1.68 | *Move the pants to any chair.* |
| Dependent Rearrange | 7 | 0.09 | *Place them on the same table.* |
| Multi-Step Rearrange | 6 | 0.11 | *Cup to sink. Cup to cabinet.* |

Table 5: **Analysis of secondary task characteristics in the PARTNR dataset.** $p$ is the % of instructions that contain the phenomenon while $\mu$ is the average number of times the phenomenon occurs within each instruction. All episodes in Test were included using automatic annotation.

Dependent rearrangements build on this task characteristic; when multiple objects/furniture/rooms can satisfy a sub-task, a task may require that same entity to solve another sub-task. An example of this is the same table being used to solve the placement of both a spoon and a bowl (*"bring the spoon and bowl to the same table in the living room."*). This occurs in 31% of episodes and the resulting dynamic dependency is a challenging aspect of collaboration. Finally, 6% of tasks include multi-step rearrangement of the same entity. For example, moving a cup to the sink and then to the cabinet after it is washed. Tasks with multi-step rearrangement have two such rearrangements on average.

**Distribution of Entities in PARTNR Tasks.** In Figure 7, we examine the distribution of task-relevant objects, furniture, and rooms. We define task-relevant objects to be objects requiring rearrangement or a state change, task-relevant furniture to be target furniture for rearrangement, and task-relevant rooms to be target rooms for rearrangement. Object categories are shown in Figure 7a, furniture categories are shown in Figure 7b, and room categories are shown in Figure 7c. The PARTNR dataset contains a long tail of semantic object categories, and within those categories, a wide variety of objects. Thus, for agents to perform well in PARTNR tasks, they must display visual reasoning and collaboration behaviors that generalize across the semantic particulars of a task. The skew of distributions in Figure 7b and Figure 7c can be understood by the relative occurrences of rooms and objects in HSSD scenes, e.g. there are more tables than couches on average.

The episodes are completely unique not just through instruction, but also through randomization of object instances, initial object locations, count of clutter objects, and agent starting positions. So, the same low-level action trajectories would not transfer to a different task.

We run an analysis on the instructions, removing all nouns (as proxy for object ID), and checking for uniqueness. We find that 800 val-1k episodes and 5193 train-10k episodes retain their uniqueness, implying 80% uniqueness in val, and 52% uniqueness in scaled train episodes. Moreover, instructions are unique with a minimum edit distance of at least 5 to account for slight variation.

## A.3 SIMULATION-IN-THE-LOOP LARGE-SCALE TASK GENERATION

In this section, we describe in detail the simulation-in-the-loop task generation pipeline. We follow a 4-step generation pipeline:

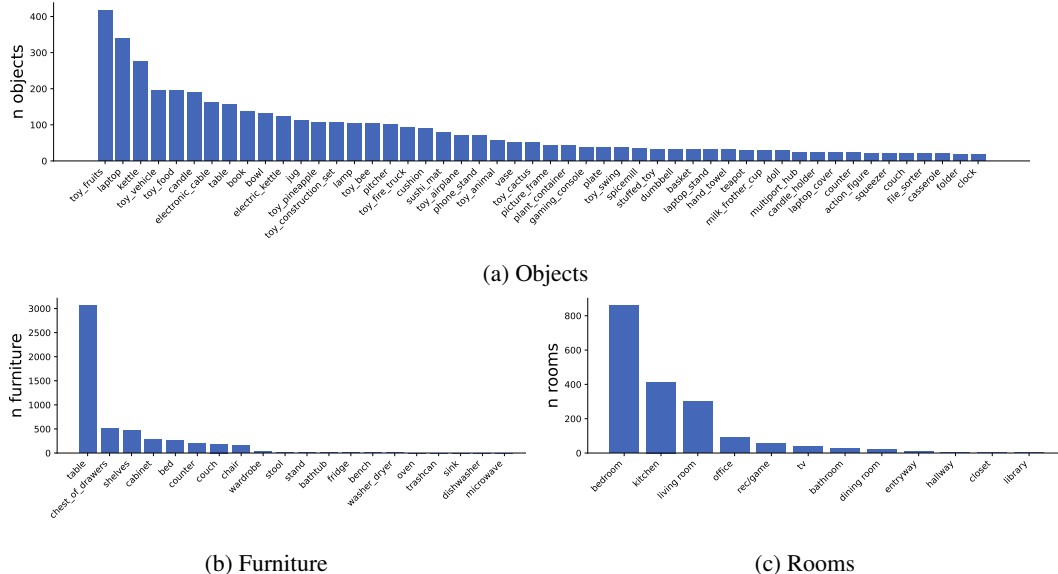

(a) Objects

(b) Furniture

(c) Rooms

Figure 7: **The distribution of task-relevant object categories, furniture categories, and room categories in the PARTNR dataset.** The object distribution is truncated to the 50 most common entities, down from 155 total object categories. The count of entities is derived by the total number of occurrences of that entity in the Test split evaluation data.

1. **Simulation-in-the-loop small-scale free-form LLM generation:**

   In **PARTNR**, we initiate the process by setting up a simulation environment using the Habitat 3.0 simulator, populated with the HSSD dataset which includes 60 unique houses and 5,819 OVMM objects. This simulated environment is parsed to identify a list of rooms and available furniture. This information, along with a list of all available objects, is then passed to a Language Model (LLM). The LLM utilizes this data to generate free-form, viable tasks within the scene, accompanied by an initial scene state description. For instance, if the task is "Clear dishes from the living room," the LLM would generate an initial scene depicting multiple dishes in the living room. To add complexity, additional objects are introduced to create clutter.

2. **Simulation-in-the-loop filtering and annotation.**

   The tasks and initial states generated by the LLM are instantiated within the simulator. At this stage, tasks that are infeasible due to the layout of the house or the capabilities of the simulator are filtered out. For example, a task like "Clear dishes from the living room" would be discarded if the simulated house lacks a living room. Similarly, tasks requiring unsupported actions, such as "folding," are also filtered out. This filtering process is crucial as it significantly reduces the number of unrealistic or impossible tasks, although it also tends to limit the diversity of the generated instructions. The full list of allowed actions in the simulator are: [open, close, power on, power off, move, clean, fill, pour, pick, place] and related actions. Next, we use manual annotation to diversity of the resulting tasks. To counter the limited diversity resulting from automated filtering, we manually annotate the tasks to ensure a balanced distribution of various types of tasks, such as constraint-free, spatial, temporal, and heterogeneous tasks. This manual intervention involves modifying the instructions to incorporate different objects and settings, resulting in a curated set of 1,000 human-annotated and simulation-verified tasks. This step is essential for maintaining quality and diversity but is impractical for scaling up due to its labor-intensive nature.

3. **Large-scale generation.**

   To scale the generation of diverse tasks without extensive manual effort, we leverage the 1,000 human-annotated instructions as examples in the LLM prompts. By providing the LLM with both a house description and an example task, we instruct it to adapt the task

> to fit the new setting, such as changing "Clear all dishes from the living room" to "Clear all toys from the bedroom." This approach helps preserve the diversity of the tasks while enhancing the likelihood of successful instantiation in the simulator.

4. **Human-in-the-loop filtering.**

   The scaled tasks undergo a final human-in-the-loop filtering where physically infeasible instructions are eliminated, ensuring the practicality and realism of the tasks. This ensures that tasks such as "Move 4 cups to the dining table", where there are only 2 cups in the scene are removed. Or "First move the cup from the kitchen to the living room, then place a jug in the kitchen, next to the cup", which consists of a physically infeasible instruction.

### A.3.1 HABITAT 3.0 AND HSSD EXTENSION

We generate **PARTNR** using modified HSSD scenes Khanna et al. (2024) and the Habitat 3.0 Puig et al. (2024) simulator due to its humanoid simulation capabilities and availability of features which support modeling of collaborative tasks as discussed in Appendix A.1.

Our extensions to the Habitat platform include a set of features targeting: object state manipulation (e.g., clean/dirty, powered on/off and filled/un-filled), evaluation of object relative relationships (e.g., next-to, above, within, on-top, on-floor, in-region, etc), and procedural clutter generation utilities enabling generation of valid initial scene contents pre-conditioned on the output of LLM-generated requirements from open-language prompts (see Section 3). For example, in order to evaluate the task of rearranging a tea set from furniture A to furniture B, we must first generate a scene with both types of furniture in accessible locations and a tea set already sitting on or inside of furniture A.

### A.4 THE **PARTNR** EVALUATION SYSTEM

In this section, we formalize the components of the evaluation system, define the resulting metrics, and present the details and prompts used for LLM-based evaluation generation.

### A.4.1 EVALUATION PREDICATES

We use logical predicates to query the state of objects, furniture, and rooms at the current timestep in the simulator. The evaluation system operates on the resulting binary state values. The details of all logical predicates are in Table 6.

### A.4.2 PROPOSITIONS

The primary component of a task evaluation function is a list of propositions. We define a proposition as an evaluation predicate instantiated with argument values. Propositions additionally enable the evaluation of instructions with ambiguous references (*"on a table"* — which table?) and subset counts (*"two spoons"* — any two of the n total spoons). Ambiguity is enabled by extending the predicate arguments to lists. Subset counts are enabled by optional arguments `number`, which defines the subset size, and `arg_match`, which is a boolean indicating whether all entities in the subset must be satisfied with the same second argument. Suppose we want to evaluate the task *"Bring a spoon to the table."*. If we have a single spoon and a single table, the proposition is straightforward:

$$is\_on\_top([spoon\_1], [table\_1]).$$

If we have multiple spoons and need just one (an ambiguous instruction), the following proposition is used:

$$is\_on\_top([spoon\_1, spoon\_2, spoon\_3], [table\_1]).$$

In the above case, a list is treated as a `OR` of entities. The same holds for multiple tables:

$$is\_on\_top([spoon\_1, spoon\_2, spoon\_3], [table\_1, table\_2]),$$

in which any spoon may be placed on any table. If the instruction specifies bringing two spoons to the table, the `number=2` argument is added:

$$is\_on\_top([spoon\_1, spoon\_2, spoon\_3],$$
$$[table\_1, table\_2], number=2).$$

| Predicate Name | Category | Description |
|---|---|---|
| is_on_top($o_1, o_2$) | Rearrange | $o_1$ is considered on top of $o_2$ if a downward raycast from any of the $o_1$ bounding box corners intersects with $o_2$. |
| is_inside($o_1, o_2$) | Rearrange | $o_1$ is considered inside $o_2$ if a threshold number of opposing raycasts hit the same object. Ray casts are 1.00m and the threshold number is 2. |
| is_in_room($o_1, r_1$) | Rearrange | $o_1$ is considered to be in room $r_1$ if at least 25% of keypoints (bounding box corners and center) are contained in the 3D region. |
| is_on_floor($o_1$) | Rearrange | $o_1$ must be within 0.04m vertically of the navigation mesh. |
| is_next_to($o_1, o_2$) | Spatial | The bounding box of $o_1$ must overlap vertically with the bounding box of $o_2$ and the horizontal L2 distance between bounding boxes is less than or equal to 0.50m. |
| is_clustered($o_1, ..., o_n$) | Spatial | Each $o_i$ satisfies is_next_to($o_i, o_j$) for some $j \neq i$. |
| is_clean($o_1$) | State | $o_1$ has affordance cleanable and the state machine indicates that $o_1$ is clean. |
| is_dirty($o_1$) | State | $o_1$ has affordance cleanable and the state machine indicates that $o_1$ is not clean. |
| is_filled($o_1$) | State | $o_1$ has affordance fillable and the state machine indicates that $o_1$ is filled. |
| is_empty($o_1$) | State | $o_1$ has affordance fillable and the state machine indicates that $o_1$ is not filled. |
| is_powered_on($o_1$) | State | $o_1$ has affordance powerable and the state machine indicates that $o_1$ is powered on. |
| is_powered_off($o_1$) | State | $o_1$ has affordance powerable and the state machine indicates that $o_1$ is powered off. |

Table 6: **Logical predicates to evaluate the relations of objects, furniture, and rooms in the PARTNR.** Predicates exist for measuring rearrangement of objects (category=Rearrange), spatial placements relative to objects or furniture (category=Spatial), and states of objects or furniture (Category=State). For all predicates, $o_i \in \{\text{object, furniture}\}$ and $r_i$ is a room.

Finally, the instruction may specify that two spoons should be brought to the *same* table. `arg_match=True` enables this:

```
is_on_top([spoon_1, spoon_2, spoon_3],
          [table_1, table_2], number=2, arg_match=True).
```

Propositions are represented as JSON within dataset files and as Python function calls (as shown above) during human annotation for interpretability.

### A.4.3 DEPENDENCIES

Evaluation functions operate over a sequence of simulation states. By default, all propositions in the list of evaluation propositions will be evaluated at every time step. However, many tasks in the **PARTNR** benchmark involve a required temporal order during execution. Take for example the instruction *"Move the cup from the table to the sink. Then, return the cup to the table."* This consists of a multi-step rearrangement where the proposition checking that the cup is on the table is *dependent* on a different proposition first being satisfied. We define a proposition dependency as the following:

```
PropositionDependency(
            proposition_indices, depends_on, relation_type),
```

where the argument `proposition_indices` is a list of integers indicating the dependent propositions, the argument `depends_on` is a list of integers indicating the head propositions, and the argument `relation_type` indicates the condition that the `depends_on` propositions must take in order for `proposition_indices` to be evaluated for satisfaction. The following relation types are supported:

- **after_satisfied:** evaluate propositions in `proposition_indices` after the propositions in `depends_on` have been satisfied.
- **after_unsatisfied:** evaluate propositions in `proposition_indices` after propositions in `depends_on` have been satisfied at some point in the past but are no longer satisfied.
- **while_satisfied:** evaluate propositions in `proposition_indices` when propositions in `depends_on` are currently satisfied.

As a concrete example, suppose the instruction is *"Move the ball and bat to the kitchen and set them next to each other. Then, move them to the closet."* The propositions for this task would be

```
0 is_in_room([ball], [kitchen])
1 is_in_room([bat], [kitchen])
2 is_next_to([ball], [bat])
3 is_in_room([ball], [closet])
4 is_in_room([bat], [closet]).
```

and the dependencies would be

```
0 PropositionDependency([2], [0, 1], "while_satisfied")
1 PropositionDependency([3, 4], [0, 1], "after_satisfied").
```

The first ensures that the `is_next_to` predicate is only queried during kitchen placements and the second ensures that placing the ball and bat in the closet is not checked at the start of the task execution, which would be inadvertently satisfied upon scene initialization. Each evaluation function has a (possibly empty) list of such proposition dependencies.

### A.4.4 CONSTRAINTS

**PARTNR** tasks often require constraining *how* tasks are completed. Examples of this include completing evaluation propositions in temporal order (*"Do [x] before doing [y]"* and enforcing links between ambiguities (*"Fill one of the cups, then put that cup on the table"*). We define a set of constraint types that enables evaluating such task complexities:

- **TemporalConstraint:** A directed acyclic graph (DAG) over the indices of propositions that defines the temporal requirement of when propositions should be satisfied. If propositions are satisfied out of order, then the task is marked unsuccessful. The temporal constraint is defined as

$$\texttt{TemporalConstraint(graph\_edges)},$$

where `graph_edges` is a set of pairwise temporal constraints. For example, an edge "$0 \rightarrow 1$" indicates that the proposition at index $0$ must be satisfied at an earlier time step than the proposition at index $1$.

- **SameArgConstraint:** Requires the argument used to satisfy a proposition to be the same within a pre-determined set of propositions. The same argument constraint is defined as

$$\texttt{SameArgConstraint(proposition\_indices, arg\_names)},$$

where `proposition_indices` is a list of proposition indices to link and `arg_names` is a list specifying which component of each proposition to link.

- **DifferentArgConstraint:** Requires the argument used to satisfy a proposition to be unique within a pre-determined set of propositions. The different argument constraint is defined as

$$\texttt{DifferentArgConstraint(proposition\_indices, arg\_names)},$$

where `proposition_indices` is a list of proposition indices to link and `arg_names` is a list specifying which component of each proposition to link.

- **TerminalSatisfactionConstraint:** Some propositions, once satisfied, should remain satisfied at the end of the episode. Others are expected to become unsatisfied, such as in multi-step rearrangements or object state changes of a single object. The terminal satisfaction constraint is defined as

$$\texttt{TerminalSatisfactionConstraint(proposition\_indices)}$$

where `proposition_indices` is a list of proposition indices that should be satisfied at time $t = T$ for an episode rollout of duration $T$.

### A.4.5 EVALUATION METRICS

A task evaluation function is constituted of propositions, dependencies, and constraints as defined above. From this data, the task evaluation function serves to determine what percentage of the task is complete for a given human-robot collaboration rollout. From a sequence of simulation states, we evaluate the truth values of each proposition with respect to both the dependencies (*when* the propositions must be evaluated) and the constraints (*how* the propositions must be satisfied). The Percent Completion ($PC$) is defined as the ratio of satisfied propositions to the total number of propositions. Success is defined as $S := (PC = 100)$. Both Percent Completion and Success metrics solely evaluate task completion and thus are agent-agnostic. Given that **PARTNR** is designed to evaluate multi-agent collaboration, we note that these metrics can be combined with duration-based metrics (e.g. simulation steps or time) to measure efficiency; multi-agent aspects like task division and partnered exploration serve to optimize task efficiency. The benefit of an agent-agnostic metric is flexibility to evaluate any number of agents performing the task with respect to the high-level goal.

### A.4.6 EVALUATION FUNCTION GENERATION

This section provides detail on evaluation function generation beyond the overview provided in Section 3. In particular, Figure 8 shows the first two steps of Figure 2 (sim-in-the-loop generation and filtering + annotation) in greater detail as they pertain to evaluation generation. Notably, we take a three step process to generating evaluation functions; first, an LLM generates a list of evaluation propositions which are parsed into a usable format. Then, an LLM infers the temporal constraint over these propositions by predicting topological generations of a temporal graph. For example, the prediction $[[0, 1], [2, 3]]$ implies that propositions at indices $0, 1$ can be completed in either order, and both must be completed before propositions at indices $2, 3$. This prediction is stored and evaluated more generally as edges of a directed acyclic graph (DAG). We found that the assumption of topological generations sufficiently expresses the tasks in **PARTNR** while being simpler

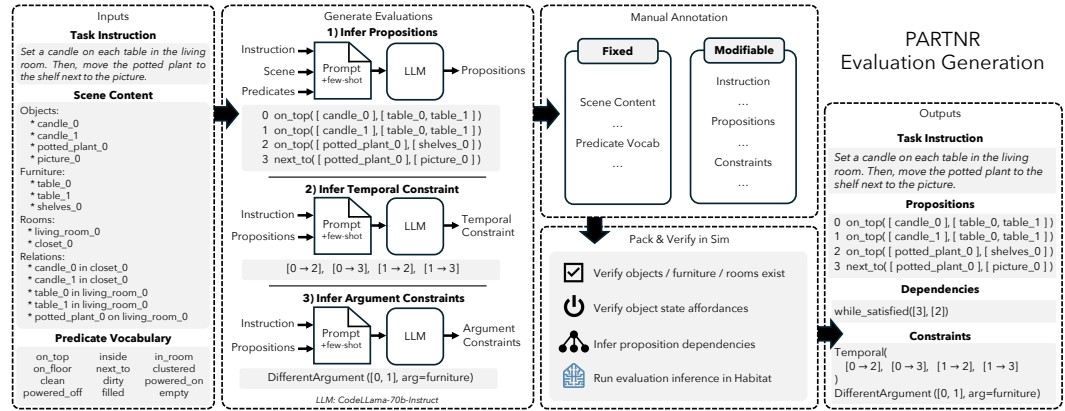

Figure 8: **Evaluation function generation pipeline overview.** A three-step LLM-generation process produces propositions and constraints during *Generate Evaluations*. The evaluation function then is saved to a file for optional human annotation during *Manual Annotation*. Finally, the evaluation function is packed and verified in simulation during *Pack & Verify in Sim* to ensure that all entities and affordances exist.

| | Correctness (%) | | |
| --- | --- | --- | --- |
| | Task | Evaluation | Combined |
| Constraint-Free | 89 | 97 | 86 |
| Spatial | 87 | 85 | 74 |
| Temporal | 91 | 93 | 85 |
| Heterogeneous | 93 | 91 | 85 |
| Average | 90 | 92 | 83 |

Table 7: **Manually-annotated generation accuracy of 100k-scale PARTNR tasks and evaluation functions.** Altogether, we find that 83% of episodes are generated without any task or evaluation function errors. Analysis performed on 100 sampled episodes of each task type via PrediViz.

for an LLM to generate than a DAG. The third step is predicting argument constraints, in which the LLM is provided the instruction and propositions list and must predict a list of constraints, either `SameArgConstraint` or `DifferentArgConstraint`. All LLM queries are performed against CodeLlama70b-Instruct (Roziere et al., 2023). All prompts used for evaluation function generation are included in Appendix A.14.

## A.5 HUMAN ANNOTATION AND ACCURACY ASSESSMENT FOR PARTNR DATA

### A.5.1 GENERATION ACCURACY FOR TASKS AND EVALUATIONS

It is important for tasks in PARTNR to be solvable by collaboration agents and for the associated evaluation functions to accurately reflect the task being performed. In this section, we analyze the accuracy of 100k-scale PARTNR generation with respect to both of these criteria using the PrediViz tool (Appendix A.5.2). In Table 7, we demonstrate that the accuracy of task generation ranges from 87-93% depending on task type and averages 90%. The accuracy of evaluation generation ranges from 85-93% depending on task type and averages 92%. Combining these numbers yields an overall joint accuracy of 83% for our 100k-scale dataset. In Table 8, we annotate the failure modes that lead to unsolvable tasks and incorrect evaluation functions. Common task-related failure modes include:

- **Hallucination.** The produced instruction references objects, furniture, or rooms that do not exist in the scene the instruction was generated for. Example: *"Move the clothes to the washing machine."* produced for an environment that does not contain a washing machine.
- **Unresolvable Ambiguity.** The produced instruction contains ambiguous directives that cannot be reasonably resolved without further communication or a detailed understanding of

| | Task Failures | | Evaluation Failures | |
| | Mode | % | Mode | % |
|---|---|---|---|---|
| Constraint-Free | Hallucination | 7 | Incorrect Ambiguity | 2 |
| | Already Satisfied | 2 | Incorrect Furniture | 1 |
| | Contradiction | 2 | - | - |
| Spatial | Hallucination | 6 | Incorrect Temporal Grouping | 7 |
| | Unresolvable Ambiguity | 5 | Incorrect Predicate (Other) | 5 |
| | Already Satisfied | 2 | Incorrect Predicate (Room vs Furniture) | 3 |
| Temporal | Hallucination | 4 | Incorrect Temporal Grouping | 3 |
| | Contradiction | 3 | Incorrect Predicate (Other) | 3 |
| | Already Satisfied | 2 | Incorrect Predicate (Room vs Furniture) | 1 |
| Heterogeneous | Hallucination | 3 | Incorrect Predicate (Room vs Furniture) | 5 |
| | Unresolvable Ambiguity | 2 | Incorrect Object/Furniture/Room | 2 |
| | Contradiction | 1 | Incorrect Predicate (Other) | 2 |

Table 8: **Top three failure modes of 100k-scale task and evaluation generation reported for each task type.** Failures of task generation are led by the hallucination of non-existent entities, while failures of evaluation generation are led by incorrect temporal predictions and incorrect predicate functions. Analysis performed on 100 sampled episodes of each task type via PrediViz.

the task-issuer's preferences. Example: *"Set the table for dinner"*; how many place settings are necessary? Should we set the formal dining table or nook table? What cutlery is needed?

- **Contradiction.** The produced instruction involves two or more sub-tasks that cannot simultaneously be satisfied. Example: *"Set the scissors on the coffee table. Set the bowl on the counter next to those scissors."*

- **Already Satisfied.** The produced instruction dictates sub-tasks that are all already satisfied at the start of the episode. Example: *"Move the laptop to the living room and turn it on"*, when the laptop is already powered on and in the living room.

According to Table 8, hallucinations are the most common failure mode for task generation. While simulation-in-the-loop filtering avoids this issue for evaluation generation, such filtering is inconsistent for tasks; language has a looser grounding to scene entities than statements of propositional logic. For example, a home might not have a formal dining room, but a table in the living room may serve the purpose of a dining table. Moving on to evaluation generation, common failures are as follows:

- **Incorrect Temporal Grouping.** The instruction implies a temporal order among sub-tasks (either explicitly via sequencing words, or implicitly via multi-step manipulations) and the predicted temporal constraint fails to reflect this order over the propositions. Example: allowing propositions to be satisfied in any order for the task *"First, return the plates to the kitchen. Then, tidy up the living room."*

- **Incorrect Predicate (Other).** The evaluation function uses the wrong predicate function to evaluate the task. Example: using `is_powered_on` instead of `is_filled` when the instruction asks to fill the kettle.

- **Incorrect Predicate (Room vs Furniture).** A task specifies that an object should be rearranged to another room, but the propositions overly-constrain the rearrangement to a target furniture for placement. This failure is separate from the one above because it the most common, and it indicates the tendency for the LLM to produce single solution instances rather than reflect the full space of ambiguity. Example: producing a proposition like `is_on_top([electronic_cable], [bed])` for the task *"Move the electronic cable to the bedroom."*

- **Incorrect Object/Furniture/Room.** incorrect entities are selected for satisfying proposition. Example: the instruction calls for rearranging the cushion to a living room table, but `table_4` in the proposition `is_on_top([cushion_0], [table_4])` exist in bedrooms.

- **Incorrect Ambiguity.** For an instruction that can be satisfied $n$ different ways, the evaluation affords $m$ options for solution, where $m \neq n$. Example: *"Move a toy to the kid's room"*,

where two or more toys exist in the toy bin but the evaluation function does not list out all possible toys for rearrangement.

According to Table 8, the primary failures modes of evaluation generation are incorrect temporal grouping and incorrect predicates. Regarding temporal grouping, we observe that many sub-task require multiple propositions to evaluate. Grouping these propositions consistently within the temporal constraint is a source of error. Take for example the instruction *"Set the shirt and pants next to each other on the counter. Then, move them to the dresser."* In this case, five propositions will exist; four for placements and one for the spatial relation. The temporal prediction may erroneously link the spatial relation with the dresser placements rather than the counter placements.

On average, this process takes 30 seconds per task. To further ensure task feasibility, we use human-in-the-loop to solve the tasks. This ensures that criteria such as object reachability and the absence of conflicting task constraints are met. This process is easily scalable on the web, as it does not require expert annotation, and takes 2 minutes 6 seconds per task on average. Our analysis indicates that automatic generation significantly reduces annotation time by a factor of 7x.

### A.5.2 Visualization of **PARTNR** Tasks and Evaluation Functions

LLM-generated **PARTNR** episodes may contain errors so we evaluate their correctness with human annotators. This is a verification problem in which the generated instruction, evaluation function, and contents of the scene must be compared against each other. See Figure 6 for an example of this in code. To make this process faster, easier, and more accurate, we designed a visualization and annotation system (PrediViz) that illustrates the state of the world and the evaluation function relative to a task instruction (Figure 9). We chose a 2D illustration style to capture the relevant structure in the data:

- **Rooms.** Each room is drawn as a box with a name provided underneath. Names repeat if there are multiple instances of that room category in the scene. We model the rooms as a fully-connected graph of accessibility. We treat rooms as spatially independent and reorder or wrap them as needed.

- **Objects.** We visualize objects as boxes with category names underneath. Each proposition of the evaluation function is assigned a unique color to separate them from other propositions. For consistency, objects are colored by the color of the first proposition they appear in.

- **Receptacles.** Receptacles are the furniture in each room. We designed a bespoke set of icons for the 25 categories of furniture such that they can be easily subitized – you can tell a chair is a chair and a table is a table just by glancing at it.

- **States.** Both objects and receptacles have states that change over time (empty/filled, dirty/-clean, powered on/off). These states are displayed using motifs for objects and textual labels for receptacles, when necessary.

- **Affordances.** Different furniture affords different object placements. For example, a cup can be placed *on top* of the fridge or *inside* it, but only *on top* of a table. These relationships are annotated by hand at the category-level. We visualize the target affordance as a dark box indicating where an object is supposed to go. The initial affordances are provided with the metadata to initialize the objects on corresponding receptacles or rooms.

- **Placements.** To represent the requested movement of an object, we draw an arrow from the object's initial position to the intended position. A simple placement is represented by a single arrow. Multiple allowable placement targets (e.g. multiple tables in the room) are represented by multiple arrows. We use solid arrows for AND placements (e.g. *"place the doll and the toy truck on the couch"*) and dotted arrows for OR placements (e.g. *"place the doll or the toy truck on the couch"*). We also have support for choosing $k$ out of $n$ objects for placement (e.g. *"place two out of three dolls on the couch"*), for which we use a numerical label on the dotted lines.

- **Temporal Constraints.** To visualize temporal constraints, we split the instruction into multiple frames. For example, if the instruction requires us to *"place the doll on the couch and the toy truck in the chair, then put the stuffed toy inside a chest of drawers,"* we create one frame to represent the first half of the statement (i.e. *"place the doll on the couch and*

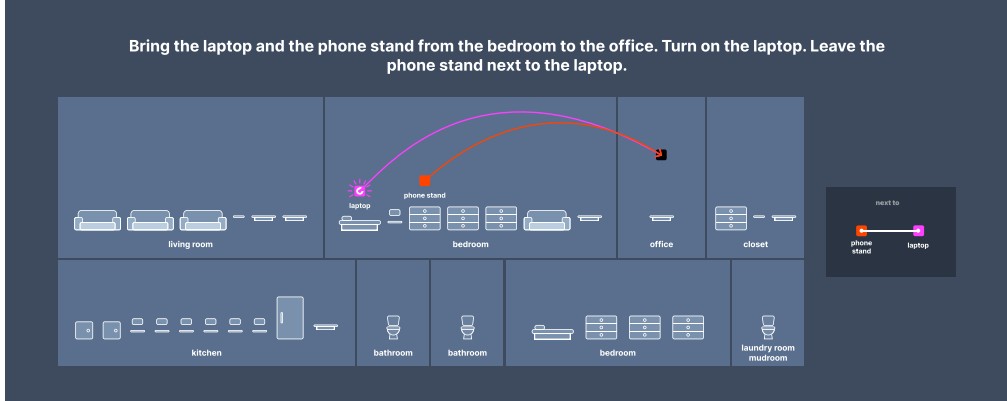

(a) Example task #1

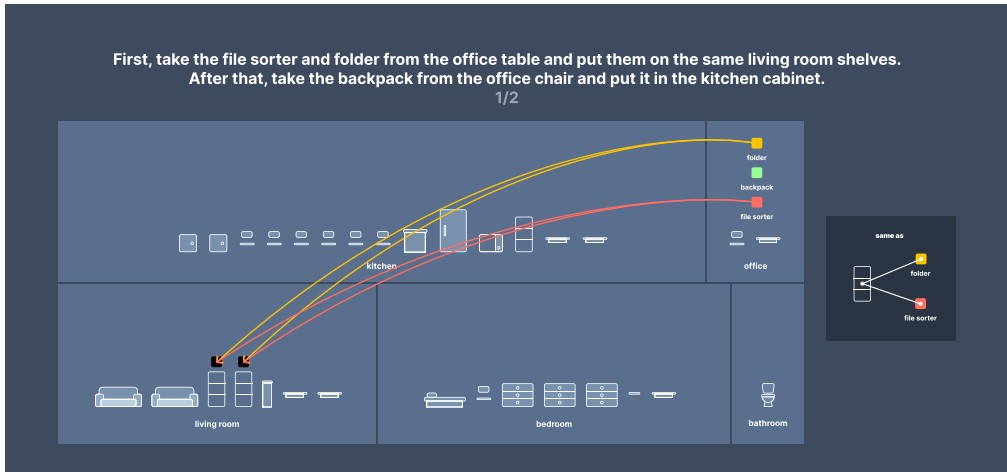

(b) Example task #2: first temporal frame

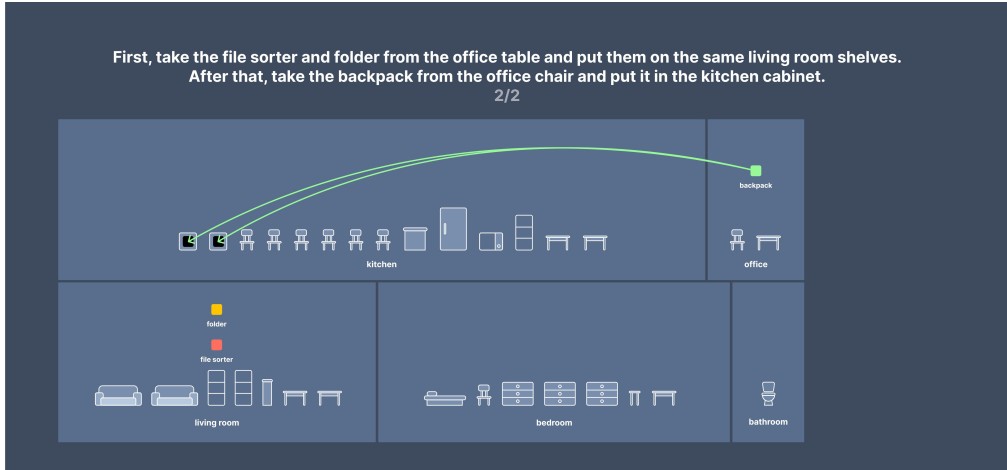

(c) Example task #2: second temporal frame

Figure 9: **PARTNR tasks visualized in PrediViz.** The design distills the task and scene to only the components necessary for verification. In example task #2, the split frames signify that the agents must first rearrange the file sorter and folder (Figure 9b), *then* rearrange the backpack (Figure 9c).

*the toy truck in the chair"*) and another for the second half (i.e. *"put the stuffed toy inside a chest of drawers"*).

- **Special Relations.** We also illustrate special relations like `next_to`, `same_as`, and `different_from` in the style of an informational legend shown on the side.

Resulting visualizations are wrapped in a web-based annotation tool that affords binary verification and failure model labeling. See Figure 8 for results derived from this tool. We ran a small-scale experimental study (n=22) comparing human annotation using PrediViz to a text-based representation. Using PrediViz, verification was 2.6 times faster, 8% more accurate, and perceived as 24% easier.

### A.5.3 HUMAN-ASSISTED CORRECTION ANNOTATION

Below is an example task and evaluation function saved in plain text for human annotation. The instruction, propositions, and constraints can be modified as necessary to ensure the task is feasible and that the evaluation function reflects it. Annotators also have access to a file containing the objects, furniture, rooms, and relations thereof to reference during this process.

---

**Dataset Correction Annotation Trial**

```
# type: ignore

# ----------------------------------------
# INSTRUCTION
#    modify as necessary, but keep in mind the scene is fixed.
# ----------------------------------------
instruction = """
Help me organize the entryway. First, place the phone, watch, and keychain on the table
next to each other.
"""

# ----------------------------------------
# PROPOSITIONS
#    is_on_top(objects, receptacles, number=1, is_same_receptacle=False)
#    is_inside(objects, receptacles, number=1, is_same_receptacle=False)
#    is_in_room(objects, rooms, number=1, is_same_room=False)
#    is_next_to(entities_a, entities_b, number=1, is_same_b=False, l2_threshold=0.5)
#    is_on_floor(objects, number=1)
#    Args:
#        objects/receptacles/entities_*: OR of a list
#        number: n objects/entities_a must satisfy
#        is_same: the same entity must satisfy all n objects
# ----------------------------------------
propositions = [
    is_on_top(['cellphone_0'], ['table_4']),
    is_on_top(['watch_0'], ['table_4']),
    is_on_top(['keychain_0'], ['table_4']),
    is_next_to(['cellphone_0'], ['watch_0']),
    is_next_to(['watch_0'], ['keychain_0']),
]

# ----------------------------------------
# TEMPORAL GROUPS
#    Place propositions in groups s.t. one group must be satisfied before the next.
#    Example:
#        [ [0, 1], [2, 3] ] means props 0 & 1 must be satisfied before props 2 & 3.
# ----------------------------------------
temporal_groups = [
    [0, 1, 2, 3, 4],
]

# ----------------------------------------
# TIE CONSTRAINTS
#    options: SameArgConstraint, DifferentArgConstraint
#    Args:
#        proposition_indices: List[int]
#        arg_indices: List[int]
#    Example:
#        SameArgConstraint([0, 2], [1, 1]). Means: Propositions 0 & 2 must
#        match values on the argument at argument index 1 and 1, respectively.
# ----------------------------------------
tie_constraints = [
]
```

---

```
# ----------------------------------------
# TERMINAL SATISFACTION CONSTRAINT:
#    We assume all propositions must remain satisfied to the end of the episode.
#    if a proposition *should* become unsatisfied, add its index here.
# ----------------------------------------
exclude_final_constraint = []

# ----------------------------------------
# mark True if the task has a fatal issue
# ----------------------------------------
skip_episode = False
reason = ""
```

## A.6 WORLD GRAPH: THE PERCEPTION FRAMEWORK FOR LLM AGENTS

Scene-graph style hierarchical graphs have been shown to be effective for planning problems Gu et al. (2024); Agia et al. (2022); Rana et al. (2023). Inspired by such prior work, as illustrated in Figure 10, our world graph is a hierarchical multi-edge directed graph with $K = 3$ levels for representing the entities in the world. The nodes at first level correspond to rooms in the environment, followed by furniture at second, and objects and agents at third level. The root of this graph is an abstract house-node denoting the environment where tasks are taking place. Apart from the semantic information, each node also stores the 3D location of the entity and its affordance states e.g., clean/dirty, on/off, open/close, etc. The graph can then be serialized or accessed via specialized tools by ReAct policies. Prompts in Section A.15 provide an example of how the world graph is used by the LLMs in our baselines.

### A.6.1 BUILDING AND UPDATING THE WORLD GRAPH

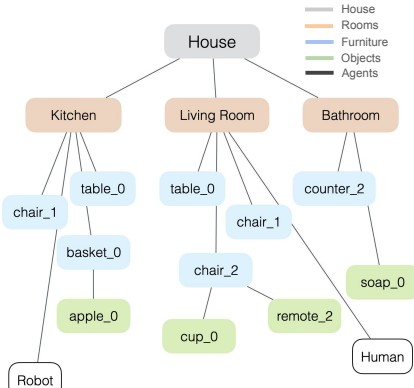

Figure 10: **World graph.** The high-level policies of the human and robot agents leverage a hierarchical world graph containing information about rooms, furniture, and objects in the environment. This graph is updated based on the observations and actions of both agents.

The initial graph is built by reading the ground-truth room region annotations and furniture placements associated with a particular scene. The location of furniture and the region boundaries are used to associate each furniture to a specific room. This creates the initial two tiers, `Room` and `Furniture`, of the privileged world-graph. In partial-observability setting this is all the planner gets to start planning. However, in the full-observability setting, object-to-furniture assignments are read from each episode's initialization information and a third `Object` tier is added to the world graph.

Under partial-observability setting during execution, we use the panoptic sensors attached to both agents to detect all visible objects in the current-frame. Then ground-truth simulation information is used to extract the housing furniture or agent for each of these objects, as well as the location, and this new information is added to the maintained graph. For full-observability setting, each frame latest graph is mined from simulator using current information for objects and overwrites the previous world-graph.

It is not guaranteed in our setup that the images will pick up an object that was placed, filled or powered on/off in the previous step. Therefore, in partial-observability setting, we also add deterministic updates to the graph based on previous action and its result, e.g. successful placement triggers deletion of edge between agent who was holding the object and an addition of edge between placed object and the furniture it was placed on per action arguments.

### A.6.2 NON-PRIVILEGED PERCEPTION USING CONCEPTGRAPHS

In order to study the dependence of the planner on the underlying world-graph, we also follow a modified version of ConceptGraphs (CG) pipeline Gu et al. (2024) to create the initial layout that is used to initialize our world-graph representation.

**Modified Pipeline for Creating ConceptGraphs.** We adapt the original CG pipeline to only use Meta-CLIP models for getting object name and category; using YOLO and SAM for object segmentation; and LLaMA3.1 for annotating room-label and inter-object spatiosemantic relations. Instead of using LLaVA and GPT for getting open-vocab descriptive object names, we use multi-perspective averaged CLIP embeddings of each object to classify its category given our closed object-vocabulary. For the required room-to-furniture relations, we extend CG pipeline by adding another prompt and query Llama3.1 model to assign room-labels to each entity given the categories of 10 closest entities to it.

**Updating ConceptGraphs.** Like the privileged world-graphs there are two modes of updating a ConcepGraphs-initialized non-privileged world-graph:

1. **Observations.** Using the same panoptic sensors we extract all the visible objects in current frame for both the agents. We use depth-sensors to extract the point-cloud associated with this object. We use this 3D location to first check if this is a redetection of a previously detected object. We use location, category and whether object-is-being-held-by-agent features to assess if this is a degenerate detection. If this is a new object detection then we use its location along-with bounding-boxes of existing furniture to check if this object is contained-within or on-top-of any of the recognized furniture-pieces. If it is then the world-graph is updated with this node and the edge.

2. **Actions.** Just like privileged world-graph, we can not guarantee sensors will pick up changed state of an object that is placed, powered, etc. Thus we add similar action based updates to this version as well. A special case is when non-privileged graph is upated by human agent's action arguments which are grounded in privileged world-graph and may refer to same physical entity with different given names, e.g. `backpack_0` is `backpack_153` in non-privileged graph. We use a simple proximity and category matchig heuristic to match human's arguments to known entities in non-privileged graph, falling back on proximity based matching when no entity of same category are found.

**Using ConceptGraphs with Simulated Skills** Simulated skills require sim-handles of the placement furniture to snap objects onto them. The furniture in non-privileged graph do not have these sim-handles by design. Thus we come up with a simple proximity and category-name based matching to match a ground-truth furniture entity to a detected furniture-entity. If we can not find any ground-truth entity of the same category close to the detected furniture, we fallback on matching to the closest entity.

**Prompts and Models used in ConceptGraphs Creation.** CLIP model and pretrained backbone checkpoints: `ViT-H-14-quickgelu`, `metaclip_fcc`. Object detector: YOLO checkpoint `yolov8x-worldv2`. LLM: `Llama3.1-70B-Instruct`

---

**Room Annotation Prompt for Modified CG**

```
<|begin_of_text|><|start_header_id|>system<|end_header_id|>You are an expert on house
layouts. You will be given an input which will describe QUERY_OBJECT. This object will
be described by its name and the 10 pieces of furniture closest to it. You will assign a
```

---

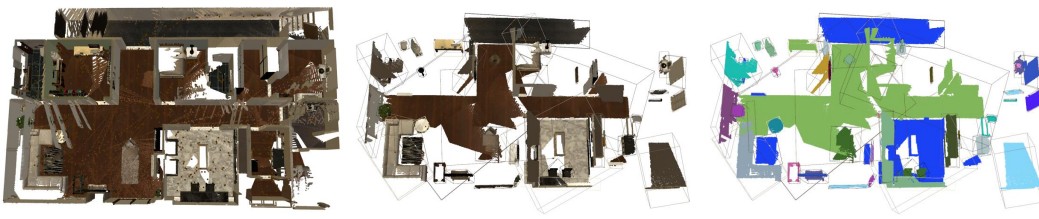

Figure 11: **ConceptGraph pipeline.** Left to right: Point-cloud of the scene built from a trajectory; All objects detected by ConceptGraphs pipeline after assigning a category-name based on CLIP; Semantic visualization of the apartment with color-coded class categories per furniture

```
ROOM_NAME to this object. The input will be a JSON with fields "QUERY_OBJECT_NAME" and
"CLOSEST_OBJECTS".  Your output should also be a JSON consisting of key "ROOM_LABEL".
You should only output the JSON and nothing else.

You should only assign one of the following labels:
1. bedroom
2. living room
3. kitchen
4. dining room
5. hallway
6. bathroom
7. unknown: only when none of the above strings describe the object<|eot_id|>
```

### A.7 LEARNED LOW-LEVEL ROBOT SKILLS

We provide the implementation details for training the learned low-level robot skills: [Explore, Navigate, OpenFurniture, CloseFurniture, PickObject, PlaceObject]. These skills are based on two types of basic skills: Navigate skill, which outputs robot base control velocity command given robot depth and sensor observations, and Manipulation skill, which outputs joint angles and base control velocity commands to reach the target location given robot depth and sensor observations. We then use these two skills to get the above robot skills.

The LLM receives a binary signal of successful or unsuccessful execution, as well as a simple explanation of the failure. For example, if navigation fails because the robot was not able to reach the target in the maximum allowed number of steps, then the LLM receives an 'observation' - "Unsuccessful execution - skill timed out on the way to target". There are some mistakes such as the robot picking up the wrong object, the observation says "Successful execution!" (since the robot is holding an object). The LLM needs to take into account the new scene graph to recover from these mistakes, however, it is unable to do so.

Most salient failure cases per skill are as below:

- Navigation : Not able to reach the target in allowed steps (timing out)
- Pick : Picking the wrong object, being too far away from target object, unable to pick because of learned skill failure (object out of reach of arm, or requiring complex motion planning to reach)
- Place: Target place position too far, skill timing out, learned skill failure (target place location out of reach).

#### A.7.1 NAVIGATE SKILL

We follow the learned low-level robot skill from Puig et al. (2024) to get the skill. We briefly describe the details for completeness. The goal of Navigate skill is to navigate to the target object given the target object location. The observation space includes (1) an arm depth camera ($224 \times 171$ with hFOV of $55$), (2) the relative pose of the target object location in 2-dim polar coordinate system. The action space includes (1) the linear and (2) angular robot base velocities with the range of $-10$ and $+10$ m/s. The reward function is to encourage the robot to move forward while facing the target location with a correct orientation. An additional navigation success reward is given if the robot can navigate close enough to the target object, and a collision penalty is given if the robot collides with obstacles in the scene. Moreover, a slack reward is given to let the robot navigate to the target in as few steps as possible. Finally, the skill is trained with DD-PPO distributing training.

Below is the list of skills based on navigate skill.

- Navigate: As described above.
- Explore: It is a composed skill that involves calling Navigate skills sequentially given a sequence of navigation waypoints.

#### A.7.2 MANIPULATION SKILL

We follow the work from Puig et al. (2024) to train the skill. We briefly describe the details for completeness. The goal of Manipulation skill is to drive the robot's arm and base to reach the

target object's location. The observation space includes (1) an arm depth camera ($224 \times 171$ with hFOV of $55$), (2) the relative pose of the target object in a 3-dim Cartesian coordinate system, (3) the 7-dim arm joint angles, (4) a binary holding detector, and (5) the relative pose of arm end-effector to the target resting location in a 3-dim Cartesian coordinate system. The action space includes (1) the linear and (2) angular base velocities with the range of $-10$ and $+10$ m/s, (3) the delta arm joint angles applied to the arm with the range of $-5 \times 10^{-2}$ and $+5 \times 10^{-2}$ (7-dim), and (4) a binary command to grasp or release the object. The reward function is to encourage the arm to move toward the object. In addition, a success reward is given if the robot interacts with the right target object. Moreover, a slack reward is given to let the robot complete the task in as few steps as possible. Finally, the skill is trained with DD-PPO distributing training.

Below is the list of skills based on `manipulate` skill.

- `OpenFurniture`: Given the drawer handle location, `manipulate` skill drives the arm and the base to the target. The articulated furniture is opened if the gripper location is close enough to the handle location.

- `CloseFurniture`: Given the drawer handle location, `manipulate` skill drives the arm and the base to the target. The articulated furniture is closed if the gripper location is close enough to the handle location.

- `PickObject`: Given the target object location, `manipulate` skill drives the arm and the base to the target. The object is snapped to the gripper if the gripper location is close enough to the target object location.

- `PlaceObject`: Given the target place location, `manipulate` skill drives the arm and the base to the target. The object is desnapped from the gripper to the target place location if the gripper location is close enough to the target place location.

## A.8 IMPLEMENTATION DETAILS FOR REACT AGENTS

For all experiments, LLM inferrence is performed on two Nvidia A100 GPUs using the `gpt-fast` inference engine PyTorch (2023). Inference on LLama-3.1-70B (using tensor parallelism over two A100s), resulted in an average generation speed of 11.43 tokens/s. Each planning step required an average of 52 tokens resulting in a latency of 4.55 seconds per planning step. The average wall time to complete and entire episode (planning steps for both agents and simulation time) was 36.0 minutes. The finetuned model based on Llama-3.1-8B required an average of 0.53s per planning step. For those experiments, simulation time and human agent inference time remained unchanged, giving a final wall time of 25.3 minutes per episode.

All decentralized baselines had a maximum timeout of 50 replanning calls, while centralized baselines had a maximum timeout of 100 replanning calls (to account for the fact that one planner would need to plan for both agents). Additionally all baselines had a maximum timeout of 20000 simulation steps.

### A.8.1 SKILL API LIBRARY

Below is a list of the skills available across all baselines. Agents acting in the robot role do not have access to state-modifying actions (Clean, Fill, Pour, PowerOff, PowerOn). The ReAct agents considered in the main paper do not have access to perception tools (FindAgentActionTool, FindObjectTool, FindReceptacleTool, FindRoomTool). We additionally study ReAct agents that query the environment via those tools, which we name ReAct-Tools.

- **Clean** : Used for cleaning an object.
- **Close** : Used for closing an articulated entity.
- **Explore** : Search a specific room by visiting various receptacles or furnitures in that room.
- **Fill** : Used for filling an object.
- **Navigate** : Used for navigating to an entity.
- **Open** : Used for opening an articulated entity.
- **Pick** : Used for picking up an object. The agent cannot hold more than one object at a time.

- **Place** : Used for placing an object on a target location. This requires the name of the object to be placed, the name of the furniture where it should be placed, spatial relation ("on" or "within") describing the relation between the object and furniture. The object to be placed must already be held by the agent (i.e. picked previously). Additionally, you can request to place the object near another object. For that you can optionally provide a spatial constraints ("next_to") and the name of the reference object. To place next to an object, the reference object must already be on the target furniture. API tempate: `Place[<object_to_be_placed>, <spatial_relation>, <furniture to be placed on>, <spatial_constraint>, <reference_object>]`. spatial_constraint and reference_object should be set to "None" when necessary.

- **Pour** : Used for pouring from one container to another. This skill will pour into the specified container from whichever container is currently held by the agent.

- **PowerOff** : Used for turning off a powered object

- **PowerOn** : Used for turning on a powered object

- **Rearrange** : Used for moving an object from its current location to the target location. This requires the name of the object to be rearranged, the name of the furniture where it should be placed, spatial relation ("on" or "within") describing the relation between the object and furniture. This skill will automatically pick the specified object and move to the target furniture and attempt to place it. Additionally, you can request to place the object near another object. For that you can optionally provide a spatial constraints ("next_to") and the name of the reference object. To place next to an object, the reference object must already be on the target furniture. API tempate: `Rearrange[<object_to_be_placed>, <spatial_relation>, <furniture to be placed on>, <spatial_constraint>, <reference_object>]`. spatial_constraint and reference_object should be set to "None" when necessary.

- **Wait** : Used to make agent stay idle for some time

- **FindPartnerAgentActionTool** : This tool will return a summary of the other agent's actions.

- **FindObjectTool** : Used to find the exact name/names of the object/objects of interest. An LLM will be used to distill relevant objects from the user query. Example (FindObjectTool[toys on the floor] or FindObjectTool[apples])

- **FindReceptacleTool** : Used to know the exact name of a receptacle. An LLM will be used to distill relevant receptacles from the user query. Example (FindReceptacleTool[a kitchen counter])

- **FindRoomTool** : Used to know the exact name of a room in the house. An LLM will be used to distill relevant rooms from the user query. Example (FindRoomTool[a room which might have something to eat])

- **Done** : Used to indicate that the agent has finished the task.

### A.8.2 CONSTRAINED GENERATION

We follow the procedure described in Geng et al. (2023), constraining token sampling to only select tokens that consistent with at least one accepting string in the specified grammar. For each call to the LLM we build a grammar which will only accept valid tool calls on observed entities. Below is the base grammar used tool calls for all experiments. For experiments utilizing a summary of the world representation (i.e. ReAct, Finetuned see Section 4.1) the perception tools (`FindObjectTool`, `FindReceptacleTool`, etc.) are omitted. The rules for `object`, `furniture`, and `room` are set dynamically based on the agent's current world graph. This ensures skills are called only for entities that the agent knows exist.

---

**Tool Call Grammar**

```
root ::= Navigate | Pick | Place | Open | Close | Rearrange | PowerOn | PowerOff | Clean
| Fill | Pour | Explore | Wait | FindReceptacleTool | FindObjectTool |
FindAgentActionTool | FindRoomTool | Done
Navigate ::= "Navigate[" nav_target "]"
Pick ::= "Pick[" object "]"
Place ::= "Place[" object "," WS spatial_relation "," WS furniture "," WS
((spatial_constraint "," WS obj_or_furniture )| (("none" | "None") WS "," WS ("none" |
"None"))) "]"
Open ::= "Open[" furniture "]"
```

```
Close ::= "Close[" furniture "]"
Rearrange ::= "Rearrange[" object "," WS spatial_relation "," WS furniture "," WS
((spatial_constraint "," WS obj_or_furniture )| (("none" | "None") WS "," WS ("none" |
"None"))) "]"
PowerOn ::= "PowerOn[" obj_or_furniture "]"
PowerOff ::= "PowerOff[" obj_or_furniture "]"
Clean ::= "Clean[" obj_or_furniture "]"
Fill ::= "Fill[" object "]"
Pour ::= "Pour[" object "]"
Explore ::= "Explore[" room "]"
Wait ::= "Wait[" "]"
FindReceptacleTool ::= "FindReceptacleTool[" free_text "]"
FindObjectTool ::= "FindObjectTool[" free_text "]"
FindAgentActionTool ::= "FindAgentActionTool[" "]"
FindRoomTool ::= "FindRoomTool[" free_text "]"
Done ::= "Done[]"
nav_target ::= (furniture | room | object)
object ::= "object_1" | "object_2" | ...
obj_or_furniture ::= (furniture | object)
furniture ::= "furniture_1" | "furniture_2" | ...
room ::= "room_1" | "room_2" | ...
spatial_constraint ::= "next_to"
spatial_relation ::= "on" | "within"
free_text ::= [ "'.:,!a-zA-Z_0-9]*
WS ::= [ ]*
```

### A.8.3 RETRIEVAL-AUGMENTED GENERATION FOR REACT AGENTS

Retrieval-Augmented Generation (RAG) is the method of optimizing LLM text generation by querying an external database. However, there are several challenges to applying RAG in our setup. First, when implementing RAG, it is necessary to provide an external database from which LLMs can retrieve information. However, it is unclear how to effectively generate such a dataset and determine which content is most beneficial for solving the task. Second, once LLMs retrieve the information, it is unknown where to ingest this information into the generation process. To solve the above challenges, we develop an approach inspired by the recent literature that uses an LLM to generate its training dataset to iteratively optimize performance (Pang et al., 2024; Madaan et al., 2024). As shown in Fig. 12, we first construct the dataset by collecting the successful traces generated by LLMs for solving the training tasks. Then, during test time, we select the most relevant trace by comparing the sentence similarity between a test instruction and the ones in the dataset. The selected trace is passed back to the LLM's promppt as a successful example trace. This represents a refinement process that uses its own past success experience to increase the chance of solving downstream tasks. Figure 12 illustrates the high-level idea.

Specifically, we use the **PARTNR** train dataset to generate ReAct traces (where the human agent is ReAct-Tools and the robot agent is ReAct). In total, we generate 925 traces that successfully solve the tasks to form the RAG dataset. During evaluation time, we use sentence similarity computed by `all-mpnet-base-v2` Reimers & Gurevych (2019) to select the most similar instruction to the instruction at hand in the dataset, followed by adding the trace into the ReAct prompt.

## A.9 FINETUNING LLM AGENTS

In this section, we describe how we finetune an LLM to build our Finetuned baseline. We detail the process to generate the data for finetuning our model and the training details below.

### A.9.1 DATA GENERATION FOR FINETUNING

We train the model using successful traces from the ReAct baseline, which obtains the best decentralized results. In particular, we run this baseline together with a ReAct-Tools human on the training set, and keep the episodes with 1.0 success rate. If an episode reaches a 100% success rate some time during the task, but fails at the end, we keep the actions up until the success step and replace the last action with a Done[] action to finish the episode. This process results in 1,226 episodes. We then split each episode into the sequence of robot actions, and filter out those that resulted in failure. Our training set is constructed by building a prompt for each of the successful actions, as shown in Sec. A.15. This process results in 15,889 training samples. Note that each action prompt contains the

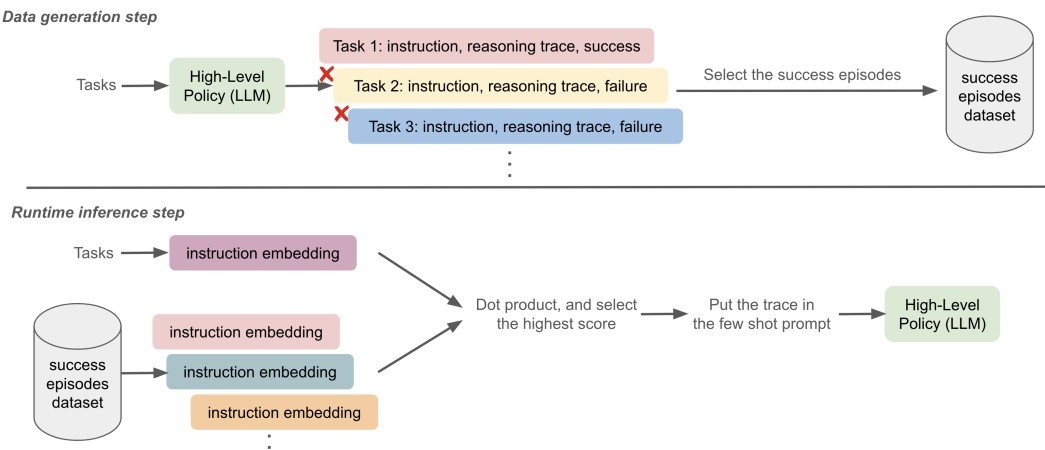

Figure 12: **Overview of ReAct-RAG.** RAG consists of two steps: (1) We first use a training dataset to obtain traces from LLMs and log traces with task instructions. (2) During runtime, we retrieve the traces by doing dot product on the instructions and obtaining the trace with the highest score, and finally put the trace in the prompt. We ensure that the dataset used for the first step is different from the one in the second step.

current state of the environment and the previous robot and human actions, but filters out the thought produced by the human or robot.

We also explored using the Heuristic-Expert baseline as a data source for finetuning, but we did not observe improvements in the resulting model. Given that this baseline plans using the ground truth evaluation function, we hypothesized it would help distill the natural language task into the right predicates. We followed the same process described above, obtaining 1250 episodes and 13939 training samples. We trained a model with ReAct, Heuristic-Expert and both data sources. We show the evaluation results for each model in Table 9, with the model trained with only React data performing the best.

| Data source | Sim Steps ↓ | Success Rate ↑ | Percent Complete ↑ | Planning Cycles ↓ |
|---|---|---|---|---|
| Heuristic-Expert | $3477.82 \pm 78.19$ | $0.63 \pm 0.02$ | $0.79 \pm 0.01$ | $15.50 \pm 0.24$ |
| ReAct | $3228.96 \pm 75.14$ | $0.70 \pm 0.01$ | $0.84 \pm 0.01$ | $12.85 \pm 0.24$ |
| ReAct + Heuristic-Expert | $3552.96 \pm 61.95$ | $0.69 \pm 0.01$ | $0.83 \pm 0.01$ | $14.47 \pm 0.30$ |

Table 9: **Performance of Finetuned model when using different data sources for finetuning.** We measure performance using simulation steps required to finish the episode, success rate and percent complete on the tasks, and the average number of planning cycles performed by the planner. Mean and standard error are reported over the validation set. Heuristic centralized expert has access to the task evaluation function.

### A.9.2 IMPLEMENTATION DETAILS

We train the model to predict, for every example, the action taken by the agent, which corresponds to the text after the `<|reserved_special_token_0>|` token.

We use a low rank adapter Hu et al. (2021) with $r = 132, \alpha = 128$, dropout=0.01, on top of the value and query projection layers $W^V, W^Q$. We train all models on 4 A100 GPUs, with a batch size of 2 per GPU. The models are trained for 40,000 steps, which takes around 24 hours.

### A.10 ADDITIONAL RESULTS

To supplement the results in Table 2, we have included Table 10, containing results from additional baselines on the validation set, and Table 11 containing results on the **PARTNR** test set. ReAct rows

| Method | Controllability | Skills | Observability | Sim Steps ↓ | Success Rate ↑ | Completion Rate ↑ | Planning Cycles ↓ |
|---|---|---|---|---|---|---|---|
| Heuristic-Expert | Centralized | Oracle | Full | 1260.88 ± 26.97 | 0.84 ± 0.01 | 0.94 ± 0.01 | N/A |
| ReAct | Centralized | Oracle | Full | 1347.43 ± 33.80 | 0.74 ± 0.01 | 0.88 ± 0.01 | 17.49 ± 0.34 |
| ReAct | Decentralized | Oracle | Full | 1915.63 ± 56.84 | 0.74 ± 0.01 | 0.86 ± 0.01 | 14.20 ± 0.34 |
| ReAct | Centralized | Oracle | Partial | 2298.13 ± 61.39 | 0.74 ± 0.01 | 0.85 ± 0.01 | 20.73 ± 0.51 |
| ReAct | Decentralized | Oracle | Partial | 3295.20 ± 76.27 | 0.73 ± 0.01 | 0.86 ± 0.01 | 15.24 ± 0.31 |
| ReAct-Tools | Decentralized | Oracle | Partial | 3622.52 ± 79.09 | 0.71 ± 0.01 | 0.85 ± 0.01 | 21.41 ± 0.34 |
| ReAct + RAG | Decentralized | Oracle | Partial | 3467.47 ± 82.39 | 0.71 ± 0.01 | 0.84 ± 0.01 | 14.75 ± 0.31 |
| Finetuned | Decentralized | Oracle | Partial | 3228.96 ± 75.14 | 0.70 ± 0.01 | 0.84 ± 0.01 | 12.85 ± 0.24 |
| ReAct | Decentralized | Learned | Partial | 6494.88 ± 181.52 | 0.57 ± 0.02 | 0.76 ± 0.01 | 22.72 ± 0.58 |
| ReAct | Decentralized | Learned | ConceptGraph | 12274.27 ± 212.65 | 0.25 ± 0.01 | 0.53 ± 0.01 | 26.74 ± 0.45 |
| ReAct-Single | Single Agent | Oracle | Partial | 2519.02 ± 57.48 | 0.73 ± 0.01 | 0.85 ± 0.01 | 18.68 ± 0.33 |
| ReAct-Single | Single Agent | Oracle | Full | 1590.20 ± 42.73 | 0.73 ± 0.01 | 0.85 ± 0.01 | 18.60 ± 0.38 |
| ReAct-8B | Decentralized | Oracle | Partial | 3699.45 ± 87.40 | 0.64 ± 0.02 | 0.80 ± 0.01 | 23.15 ± 0.47 |

Table 10: **Baseline results on PARTNR validation set.** We measure performance using simulation steps required to finish the episode, success rate and completion rate on the tasks, and the average number of planning cycles performed by the planner. Mean and standard error are reported over the validation set. Heuristic centralized expert has access to the task evaluation function. Collaboration enables higher task completion and success as compared to single agent task execution (shown in gray), at the expense of more simulation steps highlighting the coordination "burden".

| Method | Controllability | Skills | Observability | Sim Steps ↓ | Success Rate ↑ | Completion Rate ↑ | Planning Cycles ↓ |
|---|---|---|---|---|---|---|---|
| Heuristic-Expert | Centralized | Oracle | Full | 1184.74 ± 22.88 | 0.69 ± 0.02 | 0.89 ± 0.01 | N/A |
| ReAct | Centralized | Oracle | Full | 1348.71 ± 53.34 | 0.67 ± 0.02 | 0.86 ± 0.01 | 21.14 ± 0.59 |
| ReAct | Centralized | Oracle | Partial | 2590.82 ± 90.71 | 0.56 ± 0.02 | 0.80 ± 0.01 | 25.57 ± 0.64 |
| ReAct | Decentralized | Oracle | Partial | 3353.33 ± 70.03 | 0.63 ± 0.02 | 0.84 ± 0.01 | 17.38 ± 0.33 |
| ReAct-Tools | Decentralized | Oracle | Partial | 3810.15 ± 86.52 | 0.61 ± 0.02 | 0.83 ± 0.01 | 25.79 ± 0.41 |
| ReAct + RAG | Decentralized | Oracle | Partial | 3489.18 ± 79.54 | 0.62 ± 0.02 | 0.83 ± 0.01 | 17.55 ± 0.38 |
| Finetuned | Decentralized | Oracle | Partial | 3460.60 ± 78.33 | 0.51 ± 0.02 | 0.78 ± 0.01 | 14.73 ± 0.25 |
| ReAct | Decentralized | Learned | Partial | 5905.88 ± 162.35 | 0.50 ± 0.02 | 0.76 ± 0.01 | 24.30 ± 0.60 |
| ReAct-Single | Single Agent | Oracle | Partial | 2632.30 ± 60.04 | 0.68 ± 0.01 | 0.85 ± 0.01 | 21.28 ± 0.37 |
| ReAct-Single | Single Agent | Oracle | Full | 1559.73 ± 36.02 | 0.73 ± 0.01 | 0.88 ± 0.01 | 21.06 ± 0.38 |
| ReAct-8B | Decentralized | Oracle | Partial | 4100.21 ± 98.97 | 0.51 ± 0.02 | 0.77 ± 0.01 | 27.65 ± 0.52 |

Table 11: **Baseline results on PARTNR test set.** We measure performance using simulation steps required to finish the episode, success rate and completion rate on the tasks, and the average number of planning cycles performed by the planner. Mean and standard error are reported over the test set. Heuristic centralized expert has access to the task evaluation function.

use the same summary prompting format as the baselines in 2. `ReAct-Tools` requires the agents to use perception tools to observe the environment instead. `ReAct-8B` uses Llama-3.1-8B-Instruct as the robot model for an equal capacity comparison with `Finetuned`.

We also present two additional baselines that use VLMs to extract visual information. We design two new baselines that combine the ReAct approach with a VLM to leverage vision during planning, as detailed below:

In the first baseline, we measure whether visual information can be combined with our graph representation to improve planning. For this, we still provide a privileged graph representation, but also add a caption of the agent's current egocentric observation, generated using Llama-3.2-11B-Vision-Instruct. This provides strictly more information to the planner, as it uses both the visual information and accurate world graph. On the validation set, this baseline achieves a success rate of 0.72 (within the error margin of the baseline that does not use VLM, Table 2e), showing that adding the VLM output did not improve performance.

In the second baseline, we measure whether visual information alone can help perform better planning, when no graph is provided. For this, we use Llama-3.2-11B-Vision-Instruct to describe the visual observations seen by the agent so far. The planner uses these descriptions to generate an action. On the validation set, this baseline obtains a success rate of 0.69, lower than the baseline that uses a graph representation, Table 2 - row (e). Studying how to best leverage visual inputs to improve planning in PARTNR is an important direction that we leave for future work.

### A.11 ANALYSIS OF COLLABORATIVE BEHAVIOR AND EFFICIENCY OF LLM AGENTS

Since task state success and percent complete metrics look at overall team performance for the human and robot agents in our tasks, we also evaluate metrics that allow us to look at different aspects of collaborative behavior of the agents. We measure percentage of sub-tasks done by the robot (task offloading), ratio of unnecessary rearrangements over total successful rearrangements done by both agents (extraneous effort), and number of exploration steps needed before first object is picked (exploration efficiency) to analyze agent behaviors (Table 12). Our main findings are below:

| Method | Sim Steps ↓ | Task Offloading↑ | Extraneous Effort↓ | Exploration Efficiency↓ |
|---|---|---|---|---|
| Decentralized | 3295.20 $\pm_{76.27}$ | 0.596 $\pm_{0.01}$ | 0.21 $\pm_{0.01}$ | 994.88 $\pm_{24.890}$ |
| Centralized | 2298.13 $\pm_{61.39}$ | 0.49$\pm_{0.01}$ | 0.04 $\pm_{0.004}$ | 684.06 $\pm_{21.71}$ |
| Single agent | 2519.02 $\pm_{57.48}$ | - | 0.047 $\pm_{0.01}$ | 1121.65 $\pm_{31.256}$ |

Table 12: **Analysis of collaboration characteristics for LLM agents.** Average and standard errors for task offloading, extraneous effort, and exploration efficiency are reported over the successful episodes from the validation set for LLM agents using ReAct approach in partially observable setting.

**Agents are able to find objects faster when collaborating as compared to solo, but only when they successfully co-ordinate.** The exploration efficiency increases i.e., agents are able to find task-relevant objects in fewer steps, in centralized and decentralized settings. By computing the average number of exploration steps taken before the first object is picked up for a task, we find that single agents require on average 127 steps more to locate objects compared to multi-agent. However, in centralized setting, where the co-ordination between agents is better owing to a single LLM co-ordinating the actions of both agents, shows higher gains in such efficiency as compared to decentralized settings. The challenge LLMs face in coordinating exploration in multi-agent settings also negatively impacts human-LLM team performance in our HITL experiments when paired with humans (Table 3).

**Poor co-ordination also leads to wasted effort and more steps to complete the tasks than solo execution.** Despite multiple agents working together, agents take longer to complete the tasks in decentralized settings as compared to solo execution owing to poor co-ordination. Poor co-ordination is further highlighted by extraneous effort, which increases by 300% in decentralized settings as compared to solo execution. The agents often repeat parts of the task – unsure of whether the other agent really executed that part, and sometimes even undo successfully completed tasks.

| Method | Success Rate per Task-type | | | |
|---|---|---|---|---|
| | Constraint-free | Spatial | Temporal | Object states |
| Decentralized | 0.82 ±0.02 | 0.82±0.02 | 0.60±0.03 | 0.66±0.03 |
| Centralized | 0.84 ±0.02 | 0.85±0.02 | 0.59±0.03 | 0.66±0.03 |
| Single agent | 0.85±0.02 | 0.81±0.02 | 0.58±0.03 | 0.68±0.03 |

Table 13: **Task performance per task type.** Average and standard errors of task success rate for episodes from the validation set categorized by task type. Performance is shown for LLM agents that use ReAct approach in partially observable setting.

**The robot is able to offload more than half of the tasks from the human partner.** The human-robot team takes longer to complete the task in decentralized setting, however, the robot offloads 60% tasks from the human partner, reducing their load of task execution. This highlights the potential of robots assisting humans more effectively as LLMs continue to advance in reasoning, coordination, and planning capabilities.

**LLMs struggle to reason about temporal constraints and agent capabilities while planning PARTNR tasks.** Constraint-free and spatial tasks in PARTNR require the LLMs to reason about only the final configuration and states of objects. Instead, the temporal tasks in PARTNR require tracking states of one or more objects over the entire episode, making them challenging (Table 13). Likewise, heterogeneous tasks necessitate reasoning about task distribution conditioned on each agent's capabilities, which make them challenging.

## A.12   HUMAN-IN-THE-LOOP EVALUATION FOR PARTNR TASKS

### A.12.1   HITL INTERFACE AND WEB HOSTING

We adapt the existing human-in-the-loop (HITL) infrastructure from Habitat 3.0 (Puig et al., 2024) to support a more robust server-client architecture, with the server component hosted on AWS. Habitat3.0 HITL includes the ability to extend functionality to resource-constrained environments such as web browsers and VR devices, making the platform versatile for different user needs and experimental setups. Figure 13 shows our HITL system running on a web browser. Detailed interface is shown in Figure 14. Our adaptation to AWS hosting is crucial for handling multiple clients simultaneously, especially non-experts without access to powerful machines or large Habitat datasets. The server-client architecture not only enhances scalability but also ensures flexibility, allowing the system to accommodate a variety of operating systems and hardware platforms. Furthermore, the system includes a matchmaking service that enables pair-

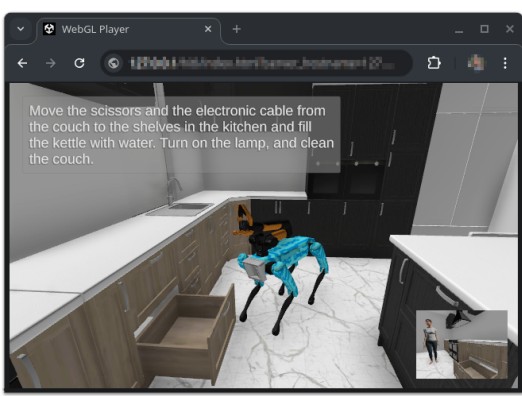

Figure 13: **HITL on Web-browser.** Our HITL system can be deployed on web browsers enabling large-scale collection.

ing participants for multi-user sessions. When a participant requests a task, they are redirected to a "lobby screen" where they are instructed to wait until the next participant arrives.

### A.12.2   PARTICIPANT RECRUITMENT AND QUALITY CONTROL

The study was performed through a 3rd party company specializing in large-scale annotations. The participants were recruited and compensated for their time by this company. The participants were English speakers, 18 years or older from the United States. For training them, we created a project and task overview video and guidelines. The participants were instructed to complete the tasks correctly and efficiently by themselves or with a partner. The participants also went through a tutorial where

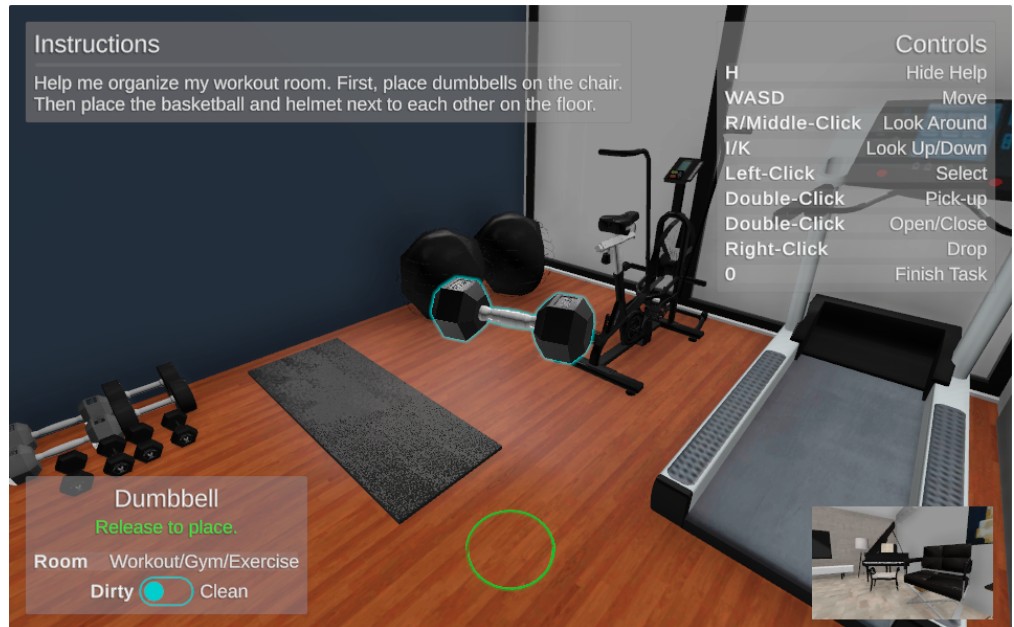

Figure 14: **HITL Interface.** Participants control human and robot agents using keyboard/mouse controls to complete the **PARTNR** tasks. Each participant has access to their partner's viewpoint and thereby current actions via a small viewport on the bottom right.

they performed some tasks to get acquainted with the interface before performing the main tasks. Each task took on average 3-5 minutes to complete. We recruited 129 non-expert participants in total.

**Filtering data:**

Each task was completed up to 3 times, in all settings, until deemed successful through task evaluation (conducted online as the participants complete the tasks). With the Failure explanation output from the evaluation function, users were also given a natural language feedback at the end of an episode, describing what went wrong in an episode, if the task was not successful. For example, if actions were completed in the wrong order, or a spatial constraint like next-to was not respected appropriately. Users can use this information to update their actions in the next episode, improving their overall performance over time. By giving 4 tutorial episodes before the start of each experiment, and also a tutorial episode in the same house, we ensure that the users are deeply familiar with the tool and tasks before starting the actual experiment.

Once each task was completed successfully at least once or retried 3 times, we assimilate data by selecting one of the tries per episode: we select the successful try for successfully completed episodes, and the highest percent complete try for unsuccessful episodes. This collection of 1000 episodes for test and val is used for collecting performance statistics described in the next section.

### A.12.3    HITL EXPERIMENTAL DETAILS

**Task Evaluation and Data Collection:**

Using the enhanced server-client setup, we conduct comprehensive evaluations of 1000 tasks from the validation set, and 1000 tasks from the test set. These evaluations are designed to capture data in both single-user and multi-user scenarios. In the single-user setting, participants individually control a human agent within the simulator using traditional keyboard and mouse inputs, completing tasks without any external assistance. Conversely, the multi-user setting involves collaborative efforts where two participants work together, each controlling either a human or a robot agent. This collaborative approach is specifically designed to study the dynamics between multiple users and to evaluate whether such collaborations lead to more efficient task completion compared to single-user efforts in the **PARTNR** tasks.

**Human-AI Collaboration Experiment:**

| Method | Success Rate ↑ | Percent Complete ↑ | Sim Steps ↓ | Task Offloading ↑ | Exploration Efficiency ↓ | Extraneous Effort ↓ |
|---|---|---|---|---|---|---|
| Validation 1,000 episodes | | | | | | |
| Single-user | $0.93 \pm 0.01$ | $0.96 \pm 0.00$ | $3046.99 \pm 80.79$ | N/A | $2459.22 \pm 26.75$ | $0.09 \pm 0.01$ |
| Multi-user | $0.93 \pm 0.01$ | $0.96 \pm 0.00$ | $2369.55 \pm 49.33$ | $0.59 \pm 0.01$ | $1762.47 \pm 13.99$ | $0.15 \pm 0.01$ |
| Human-ReAct | $0.91 \pm 0.01$ | $0.96 \pm 0.02$ | $4267.71 \pm 83.40$ | $0.16 \pm 0.01$ | $2624.39 \pm 26.05$ | $0.12 \pm 0.01$ |
| Human-Finetuned | $0.92 \pm 0.01$ | $0.96 \pm 0.00$ | $3443.33 \pm 61.46$ | $0.26 \pm 0.01$ | $2164.94 \pm 21.31$ | $0.13 \pm 0.01$ |
| Test 1,000 episodes | | | | | | |
| Single-user | $0.89 \pm 0.01$ | $0.95 \pm 0.00$ | $3937.87 \pm 110.53$ | N/A | $2737.44 \pm 25.27$ | $0.13 \pm 0.00$ |
| Multi-user | $0.85 \pm 0.01$ | $0.95 \pm 0.00$ | $2667.86 \pm 58.07$ | $0.60 \pm 0.01$ | $1889.56 \pm 14.9 = 69$ | $0.20 \pm 0.00$ |
| Human-ReAct | $0.87 \pm 0.01$ | $0.95 \pm 0.00$ | $4080.10 \pm 72.24$ | $0.12 \pm 0.00$ | $2449.50 \pm 19.98$ | $0.18 \pm 0.01$ |
| Human-Finetuned | $0.87 \pm 0.01$ | $0.95 \pm 0.00$ | $3403.03 \pm 62.08$ | $0.26 \pm 0.01$ | $2162.20 \pm 19.40$ | $0.20 \pm 0.01$ |

Table 14: **Human-in-the-Loop evaluation.** We evaluate the performance of 2-person human teams and human-LLM teams, comparing them to solo human performance on **PARTNR** tasks using metrics described in Section 4.1. The human-LLM teams with SoTA LLMs is *slower* than solo human.

In addition to human-only interactions, we conduct experiments where a human collaborates with a robot controlled by a Language Model (LLM), specifically using the ReAct and Finetuned models as described in Section 4.1. The primary goal of these experiments is to evaluate the effectiveness of LLM-controlled agents in real-time collaboration with non-expert humans who have not previously interacted with these AI models. We track and compare the success rates (SR) and the percentage of tasks completed (PC) across various settings including single-user, multi-user, and human-AI collaborations. The results of these experiments are systematically documented and analyzed in Table 14, providing insights into the collaborative capabilities of human-AI pairs.

To enable this setting, we host LLMs on AWS nodes, and query them intermittently based on robot observations and actions. The HITL server now queries two clients - a human and a LLM. The human client sends commands to control the human agent, and the LLM client uses the hosted LLM to control the robot. Different baselines need different numbers of GPUs to keep the inference time reasonable. For hosting and 70B models, as used by the ReAct baselines, we use 4 A100 GPUs per model. For hosting a smaller 8B model used by the Finetuned baselines, we use 1 A100 GPU. This makes deploying smaller Finetuned models much scalable than larger 70B models.

**Efficiency and Task Offloading Metrics:**

To further understand the efficiency of task completion across different experimental setups, we measure several key performance metrics beyond success rate and percent complete. These include the number of steps taken to complete tasks and the exploration efficiency, which is assessed by the number of steps participants take to pick the first object. Additionally, we evaluate the extraneous effort by noting actions that do not contribute directly to task completion. Another critical metric we analyze is the ratio of work completed by the robot, referred to as task offloading. Ideally, in a well-coordinated human-AI team, the task offloading ratio should approach 0.5, indicating an efficient division of labor between the human and the robot.

### A.12.4 HITL ANALYSIS

**Humans are significantly better than LLMs at PARTNR tasks.** Both single and multi-human settings achieve a success rate of 0.93 on **PARTNR** validation tasks, while ReAct without any privileged information only achieves 0.30 (Table 2, row (i)). This indicates a huge gap in LLM planning performance. We observe a slightly lower human performance on the test set (0.89), also in line with a lower LLM performance in Table 11 on this dataset. This indicates that the tasks in our test set are more challenging than the validation set for both humans and LLMs alike, potentially due to human annotations aimed at making them more complex and diverse.

**Finetuned LLMs perform better than ReAct when coordinating with real humans.** When deployed with real humans-in-the-loop on the validation set, the finetuned model is faster than ReAct at task completion (3275 steps with finetuned versus 4484 with ReAct on the validation set). It is also able to offload more tasks from humans than ReAct (26% with finetuned as compared to 16% with ReAct). This reflects that smaller models with faster inference can improve human experience in real-world deployment. This result is also reflected in the test set where finetuned model outperforms

ReAct. Interestingly, the automated eval performance of finetuned is worse than ReAct on the test set, but the HITL performance is better, indicating that faster inference is more critical than task success when working with real humans.

**LLMs struggle at coordination, hampering human performance.** Despite the Finetuned being faster than ReAct when collaborating with humans, both approaches are *slower* than the human doing the task alone. In contrast, two humans working together complete the task faster than a single human (2369 steps vs. 3046 with multi- and single-user respectively). This result is in line with the automated evaluation we observed in Table 1, where multi-agent LLMs are also *slower* than a single-agent LLM. This result further reinforces that LLMs suffer at coordination; while humans are able to coordinate and divide tasks between each other, decentralized LLMs are unable to do so. We observe the same effect in the test set, further reinforcing this finding.

**LLMs are able to offload tasks from humans.** Despite the aforementioned increase in the number of steps for task completion, robots guided by the finetuned model successfully offload 26% of tasks from humans. This indicates that LLMs can still offer assistance when collaborating with real human partners. Nonetheless, there remains significant potential for improvement.

**LLM's inefficiency to explore reduces the team performance when paired with humans.** In multi-user condition, the two humans start in different parts of a house, and explore efficiently to locate task-relevant objects more quickly than a single user – as evidenced by the reduced number of steps before first pick (1762 steps with multi-user vs. 2459 steps for a single user). However, this efficiency is reduced when humans are paired with LLMs (2120 steps with finetuned and 2791 steps with ReAct), indicating that LLMs struggle to coordinate at *both* task completion and exploration.

## A.13 PROMPTS FOR BENCHMARK TASK AND EVALUATION

### A.13.1 TASK GENERATION PROMPTS

Prompts are similar for the different task types, with the primary difference being the in-context examples.

---

**Constraint-free task generation prompt**

```
You are a system that generates tasks for robots to perform with humans.
Do not be verbose. Answer the question with no added qualifications or caveats. Just
directly provide the answer in JSON.

You will be given a description of a house with objects and furniture and your task is
to provide 5 instructions for tasks that a robot and a human could be doing together in
that house, using the objects and furniture.

For each task, provide the initial state of objects in the house, the instruction that
should be performed, and final state of the objects after the instruction is performed.
The initial and final state will contain a list of dictionaries, each with an object
type, the number of objects of that type, their location on a furniture or floor, and
the region of the house where they are in e.g., bedroom.

Follow the next principles:

1. The instruction should be given as if the human doing the task wanted the robot to
perform part of it. In some cases the task will be done together, in other cases, the
human and robot will be doing different tasks.
2. The initial and final state should contain objects of different types, and sometimes
multiple objects of a type.
3. Some of the instructions should be semantically rich, in particular they should refer
to classes or groups of objects.
4. The instructions shouldn't be detailed and explain all the steps, but the high-level.
5. The robot can only rearrange objects and open containers, the human can do more tasks
e.g., turn on lamp, clean plates, fill up pitcher.
6. The instruction should contain a clear goal and at least two steps associated with
the goal.
7. Ensure that instructions are diverse from each other. Some instructions should
contain spatial specifiers such as "next to", "left", "right", "beside", "near",
"front", "side". While some other instructions should contain temporal order, which can
be specified using words such as "after", "then", "before" etc. For instance: "Fill up
the kettle and then turn it on. After that, bring two cups to the dinning table."
```

```
8. The instructions should contain diverse actions such as "turn on/off", "fill",
"clear", "set" etc. and object states such as "clean", "dirty", "open", "close" etc.
while referring to objects.

You will be given all the pieces of furniture in the house.
You will also be given all the different types of objects that you can use. You can
specify multiple instances of an object type.
Make sure you instruction includes the object types and furniture present in the list
below.

The house has the following rooms, each with the following furniture:
{house_furniture}

You can use the following objects:
{objects_list}

Here is an example with two instructions:

JSON_OUTPUT: [
{{
    "initial state": [
        {{
            "object_type": "lamp",
            "how_many": 1,
            "furniture_name": "table_10",
            "region": "living_room_1"
        }},
        {{
            "object_type": "book",
            "how_many": 3,
            "furniture_name": "table_11",
            "region": "living_room_1"
        }},
        {{
            "object_type": "toy_vehicle",
            "how_many": 2,
            "furniture_name": "floor",
            "region": "living_room_1"
        }},
        {{
            "object_type": "toy_cactus",
            "how_many": 1,
            "furniture_name": "table_1",
            "region": "living_room_1"
        }}
    ],
    "final state": [
        {{
            "object_type": "lamp",
            "how_many": 1,
            "furniture_name": "table_10",
            "region": "living_room_1"
        }},
        {{
            "object_type": "book",
            "how_many": 3,
            "furniture_name": "shelves_2",
            "region": "living_room_1"
        }},
        {{
            "object_type": "toy_vehicle",
            "how_many": 2,
            "furniture_name": "bed_2",
            "region": "bedroom_1"
        }},
        {{
            "object_type": "toy_cactus",
            "how_many": 1,
            "furniture_name": "bed_2",
            "region": "bedroom_1"
        }},
    ],
    "instruction": "We need to clean up the living room. Move all toys and books to the
    shelf in the living room.",
```

```
     "reason": "The task involves moving multiple objects to the shelf in the living
     room."
}},

{{
     "initial state": [
          {{
               "object_type": "plate",
               "how_many": 3,
               "furniture_name": "cabinet_2",
               "region": "kitchen_1"
          }},
          {{
               "object_type": "glass",
               "how_many": 2,
               "furniture_name": "counter_1",
               "region": "kitchen_1"
          }},
          {{
               "object_type": "fork",
               "how_many": 5,
               "furniture_name": "cabinet_5",
               "region": "kitchen_1"
          }},

     "final state": [
          {{
               "object_type": "plate",
               "how_many": 2,
               "furniture_name": "table_8",
               "region": "living_room_1"
          }},
          {{
               "object_type": "glass",
               "how_many": 2,
               "furniture_name": "table_8",
               "region": "living_room_1"
          }},
          {{
               "object_type": "fork",
               "how_many": 2,
               "furniture_name": "table_8",
               "region": "living_room_1"
          }},
     ],
     "instruction": "Help me set up a table for dinner in the livingroom for 2 people.
     Place 2 plates and 2 glasses on the table. ",
     "reason": "The task includes semantically rich descriptions (set up a table)."
}}
]

Generate a JSON list with {k} instructions like the examples above.
Your output should only be:
JSON_OUTPUT: result_list
where result_list should be a JSON list with the instructions.
Let's think through this carefully, step by step.
```

### Spatial task generation prompt

```
You are a system that generates tasks for robots to perform with humans.
Do not be verbose. Answer the question with no added qualifications or caveats. Just
directly provide the answer in JSON.

You will be given a description of a house with objects and furniture and your task is
to provide 5 instructions for tasks that a robot and a human could be doing together in
that house, using the objects and furniture.

For each task, provide the initial state of objects in the house, the instruction that
should be performed, and final state of the objects after the instruction is performed.
The inital and final state will contain a list of dictionaries, each with an object
type, the number of objects of that type, their location on a furniture or floor, and
the region of the house where they are in e.g., bedroom.

Follow the next principles:
```

```
1. The instruction should be given as if the human doing the task wanted the robot to
perform part of it. In some cases the task will be done together, in other cases, the
human and robot will be doing different tasks.
2. The initial and final state should contain objects of different types, and sometimes
multiple objects of a type.
3. Some of the instructions should be semantically rich, in particular they should refer
to classes or groups of objects.
4. The instructions shouldn't be detailed and explain all the steps, but the high-level.
5. The robot can only rearrange objects and open containers, the human can do more tasks
e.g., turn on lamp, clean plates, fill up pitcher.
6. The instruction should contain a clear goal and at least two steps associated with
the goal.
7. Ensure that instructions are diverse from each other.
8. All the instructions should contain at least one of the spatial specifiers from this
list: "next to", "left", "right", "beside", "near", "front", "side".
9. The instructions should contain diverse actions such as "turn on/off", "fill",
"clear", "set" etc. and object states such as "clean", "dirty", "open", "close" etc.

You will be given all the pieces of furniture in the house.
You will also be given all the different types of objects that you can use. You can
specify multiple instances of an object type.
Make sure you instruction includes the object types and furniture present in the list
below.

The house has the following rooms, each with the following furniture:
{house_furniture}

You can use the following objects:
{objects_list}

Here is an example with two instructions:

JSON_OUTPUT: [
{{
    "initial state": [
        {{
            "object_type": "vase",
            "how_many": 1,
            "furniture_name": "table_10",
            "region": "living_room_1"
        }},
        {{
            "object_type": "stuffed_toy",
            "how_many": 2,
            "furniture_name": "floor",
            "region": "bedroom_2"
        }},
        {{
            "object_type": "candle",
            "how_many": 1,
            "furniture_name": "chest_of_drawers_2",
            "region": "bedroom_1"
        }}
    ],
    "final state": [
        {{
            "object_type": "vase",
            "how_many": 1,
            "furniture_name": "shelves_2",
            "region": "living_room_1"
        }},
        {{
            "object_type": "stuffed_toy",
            "how_many": 2,
            "furniture_name": "shelves_2",
            "region": "living_room_1"
        }},
        {{
            "object_type": "candle",
            "how_many": 1,
            "furniture_name": "shelves_2",
            "region": "living_room_1"
        }}
    ],
```

```
        "instruction": "Let's decorate! Put the vase on the shelf. Then, set a candle and a
        stuffed_toy on each side of the vase.",
        "reason": "The task includes spatial constraint specified by 'side'."
}},

{{
        "initial state": [
            {{
                "object_type": "plate",
                "how_many": 3,
                "furniture_name": "cabinet_2",
                "region": "kitchen_1"
            }},
            {{
                "object_type": "glass",
                "how_many": 2,
                "furniture_name": "counter_1",
                "region": "kitchen_1"
            }},
            {{
                "object_type": "fork",
                "how_many": 5,
                "furniture_name": "cabinet_5",
                "region": "kitchen_1"
            }},

        "final state": [
            {{
                "object_type": "plate",
                "how_many": 2,
                "furniture_name": "table_8",
                "region": "living_room_1"
            }},
            {{
                "object_type": "glass",
                "how_many": 2,
                "furniture_name": "table_8",
                "region": "living_room_1"
            }},
            {{
                "object_type": "fork",
                "how_many": 2,
                "furniture_name": "table_8",
                "region": "living_room_1"
            }},
        ],
        "instruction": "Help me set up a table for dinner in the livingroom for 2 people.
        Place 2 plates and 2 glasses on the table. There should be a fork next to each
        plate",
        "reason": "The task includes semantically rich descriptions (set up a table) and
        spatial constraints specified by the word 'next'."
}}
]

Generate a JSON list with {k} instructions like the examples above.
Your output should only be:
JSON_OUTPUT: result_list
where result_list should be a JSON list with the instructions.
Let's think through this carefully, step by step.
```

### Temporal task generation prompt

```
You are a system that generates tasks for robots to perform with humans.
Do not be verbose. Answer the question with no added qualifications or caveats. Just
directly provide the answer in JSON.

You will be given a description of a house with objects and furniture and your task is
to provide 5 instructions for tasks that a robot and a human could be doing together in
that house, using the objects and furniture.

For each task, provide the initial state of objects in the house, the instruction that
should be performed, and final state of the objects after the instruction is performed.
The inital and final state will contain a list of dictionaries, each with an object
type, the number of objects of that type, their location on a furniture or floor, and
the region of the house where they are in e.g., bedroom.
```

```
Follow the next principles:

1. The instruction should be given as if the human doing the task wanted the robot to
perform part of it. In some cases the task will be done together, in other cases, the
human and robot will be doing different tasks.
2. The initial and final state should contain objects of different types, and sometimes
multiple objects of a type.
3. Some of the instructions should be semantically rich, in particular they should refer
to classes or groups of objects.
4. The instructions shouldn't be detailed and explain all the steps, but the high-level.
5. The robot can only rearrange objects and open containers, the human can do more tasks
e.g., turn on lamp, clean plates, fill up pitcher.
6. The instruction should contain a clear goal and at least two steps associated with
the goal.
7. Ensure that instructions are diverse from each other.
8. All the instructions should contain temporal order, specified using one of the words
from this list: "after", "then", "before", "finally", "first". For instance: "Fill up
the kettle and then turn it on. After that, bring two cups to the dinning table."
9. The instructions should contain diverse actions such as "turn on/off", "fill",
"clear", "set" etc. and object states such as "clean", "dirty", "open", "close" etc.

You will be given all the pieces of furniture in the house.
You will also be given all the different types of objects that you can use. You can
specify multiple instances of an object type.
Make sure you instruction includes the object types and furniture present in the list
below.

The house has the following rooms, each with the following furniture:
{house_furniture}

You can use the following objects:
{objects_list}

Here is an example with two instructions:

JSON_OUTPUT: [
{{
    "initial state": [
        {{
            "object_type": "lamp",
            "how_many": 1,
            "furniture_name": "table_10",
            "region": "living_room_1"
        }},
        {{
            "object_type": "book",
            "how_many": 3,
            "furniture_name": "table_11",
            "region": "living_room_1"
        }},
        {{
            "object_type": "toy_vehicle",
            "how_many": 2,
            "furniture_name": "floor",
            "region": "living_room_1"
        }},
        {{
            "object_type": "toy_cactus",
            "how_many": 1,
            "furniture_name": "table_1",
            "region": "living_room_1"
        }}
    ],
    "final state": [
        {{
            "object_type": "lamp",
            "how_many": 1,
            "furniture_name": "table_10",
            "region": "living_room_1"
        }},
        {{
            "object_type": "book",
            "how_many": 3,
            "furniture_name": "shelves_2",
```

```
                     "region": "living_room_1"
              }},
              {{
                     "object_type": "toy_vehicle",
                     "how_many": 2,
                     "furniture_name": "bed_2",
                     "region": "bedroom_1"
              }},
              {{
                     "object_type": "toy_cactus",
                     "how_many": 1,
                     "furniture_name": "bed_2",
                     "region": "bedroom_1"
              }},
       ],
       "instruction": "We need to clean up the living room. Move all toys to the bedroom
       and the books to the shelf. After that, turn on the lamp in the living room.",
       "reason": "The task includes temporal constraints specified by 'after'."
}},

{{
       "initial state": [
              {{
                     "object_type": "plate",
                     "how_many": 3,
                     "furniture_name": "cabinet_2",
                     "region": "kitchen_1"
              }},
              {{
                     "object_type": "glass",
                     "how_many": 2,
                     "furniture_name": "counter_1",
                     "region": "kitchen_1"
              }},
              {{
                     "object_type": "fork",
                     "how_many": 5,
                     "furniture_name": "cabinet_5",
                     "region": "kitchen_1"
              }},

       "final state": [
              {{
                     "object_type": "plate",
                     "how_many": 2,
                     "furniture_name": "table_8",
                     "region": "living_room_1"
              }},
              {{
                     "object_type": "glass",
                     "how_many": 2,
                     "furniture_name": "table_8",
                     "region": "living_room_1"
              }},
              {{
                     "object_type": "fork",
                     "how_many": 2,
                     "furniture_name": "table_8",
                     "region": "living_room_1"
              }},
       ],
       "instruction": "Help me set up a table for dinner in the livingroom for 2 people.
       First place 2 plates on the table. Then, place glasses and forks next to each
       plate.",
       "reason": "The task includes semantically rich descriptions (set up a table) and
       temporal constraints specified by the words 'first' and 'then'."
}}
]

Generate a JSON list with {k} instructions like the examples above.
Your output should only be:
JSON_OUTPUT: result_list
where result_list should be a JSON list with the instructions.
Let's think through this carefully, step by step.
```

**Large-scale task generation prompt**

```
You are a system that generates tasks for robots to perform with humans.
Do not be verbose. Answer the question with no added qualifications or caveats. Directly
provide the answer in JSON.

You will be given a description of a house with objects and furniture. You will also be
given a sample task. Your job is to modify this sample task so that it is applicable to
this house.

Here is an example:

Task: [
{{
    "instruction": "Move the kettle and the jug from the living room to the kitchen and
    fill the kettle with water, then turn on the kettle. Finally clean the living room
    table.",
    "initial_state": [
        {{
            "number": 1,
            "object_classes": [
                "kettle"
            ],
            "furniture_names": [
                "table_0"
            ],
            "allowed_regions": [
                "living_room_0"
            ]
        }},
        {{
            "number": 1,
            "object_classes": [
                "jug"
            ],
            "furniture_names": [
                "table_1"
            ],
            "allowed_regions": [
                "living_room_0"
            ]
        }}
    ]
}}
]

JSON_OUTPUT: [
{{
    "instruction": "Move the jug, kettle, teapot, and cup from the dining table to the
    kitchen and fill all with water. Turn on the lamp, and clean the dining table.",
    "initial_state": [
        {{
            "number": 1,
            "object_classes": [
                "jug"
            ],
            "furniture_names": [
                "table_1"
            ],
            "allowed_regions": [
                "living_room_0"
            ]
        }},
        {{
            "number": 1,
            "object_classes": [
                "kettle"
            ],
            "furniture_names": [
                "table_1"
            ],
            "allowed_regions": [
                "living_room_0"
            ]
        }},
        {{
            "number": 1,
```

```
            "object_classes": [
                "teapot"
            ],
            "furniture_names": [
                "table_1"
            ],
            "allowed_regions": [
                "living_room_0"
            ]
        }},
        {{
            "number": 1,
            "object_classes": [
                "cup"
            ],
            "furniture_names": [
                "table_0"
            ],
            "allowed_regions": [
                "living_room_0"
            ]
        }},
        {{
            "number": 1,
            "object_classes": [
                "lamp"
            ],
            "furniture_names": [
                "table_1"
            ],
            "allowed_regions": [
                "living_room_0"
            ]
        }}
    ]
}}
]

The house has the following rooms, each with the following furniture:
{house_furniture}

You can use the following objects:
{objects_list}

Here is the task:
Task: [
{task}
]

Modify this task to generate a JSON list of tasks, using the rooms and furniture from
this house.

Just change the objects and furniture.

Make sure initial and final locations of objects are different.

Include actions such as turn on/off, fill and clean.

Make tasks multi-step, consisting of more than one object and action.

Your output should only be:
JSON_OUTPUT: result_list
where result_list should be a JSON list with the tasks.
```

## A.14 EVALUATION GENERATION PROMPTS

Here we share the LLM prompts used for proposition generation, temporal constraint prediction, and argument constraint prediction. In each, the task to accomplish is described in the system prompt and between 6-13 few-shot examples follow.

**Evaluation Generation: Propositions**

```
Source: system
```

```
You will be given an instruction describing a household task and a description of the
initial state of the house. You will define a list of python functions that must be
satisfied for the task to be marked complete.

You can call the following functions:
- is_on_top(object_names: str, furniture_name: str)  # any object in object_names is on
top of a furniture
- is_inside(object_names: str, furniture_name: str)  # any object in object_names is
inside of a furniture
- is_in_room(object_names: str, room_name: str)      # any object in object_names is in
a room
- is_on_floor(object_names: str)                     # any object in object_names is on
the floor
- is_next_to(objects_a: str, objects_b: str)         # any object in objects_a is next
to any object in objects_b
- is_clean(object_names: str)                        # any object in object_names is
clean
- is_dirty(object_names: str)                        # any object in object_names is
dirty
- is_filled(object_names: str)                       # any object in object_names is
filled, like with a liquid
- is_empty(object_names: str)                        # any object in object_names is
empty
- is_powered_on(object_names: str)                   # any object in object_names is
powered on
- is_powered_off(object_names: str)                  # any object in object_names is
powered off

Objects in object_names can be expressed as it appears in the objects list
("stuffed_toy_1") or as an OR of object names ("stuffed_toy_1 or stuffed_toy_2").
A furniture_name can be expressed as it appears in the furniture list (e.g. "table") or
as it appears in the furniture-room relation ("table in living_room").

Only use the functions listed above.
Each function should test a single objects/furniture/room relation.
If the instruction is ambiguous such that multiple objects could be used to satisfy a
function (an OR relationship), then include all possible objects.
Define as many functions as necessary.
Write each function on its own line.
It is essential to wrap each function in delimiters [FN] and [/FN].
End your functions with the token [END].

Let's see some examples. Suppose the initial state is:

Objects:
    * pants_1
    * shirt_1
    * shirt_2
    * shirt_3
Furniture:
    * washer_dryer
    * table
Rooms:
    * laundryroom
Object-Furniture-Room Relations:
    * pants_1 on table in laundryroom
    * shirt_1 on table in laundryroom
    * shirt_2 on floor in laundryroom
Furniture-Room Relations:
    * washer_dryer in laundryroom
    * table in laundryroom

Instruction "Put the pants on the table" means

    [FN] is_on_top("pants_1", "table in laundryroom") [/FN]

Instruction "Put the pants in the washer" means

    [FN] is_inside("pants_1", "washer_dryer in laundryroom") [/FN]

Instruction "Put a shirt in the washer" means

    [FN] is_inside("shirt_1 or shirt_2 or shirt_3", "washer_dryer in laundryroom") [/FN]

Instruction "Put all the shirts in the washer" means

    [FN] is_inside("shirt_1", "washer_dryer in laundryroom") [/FN]
    [FN] is_inside("shirt_2", "washer_dryer in laundryroom") [/FN]
    [FN] is_inside("shirt_3", "washer_dryer in laundryroom") [/FN]
```

```
Instruction "Put a shirt on the floor next to the pants" means

    [FN] is_on_floor("shirt_1 or shirt_2 or shirt_3") [/FN]
    [FN] is_next_to("shirt_1 or shirt_2 or shirt_3", "pants_1") [/FN]

Instruction "Bring a shirt to the laundryroom" means

    [FN] is_in_room("shirt_1 or shirt_2 or shirt_3", "laundryroom") [/FN]

<step> Source: user

Objects:
    * casserole_0
    * squeezer_0
Furniture:
    * cabinet
    * table
Rooms:
    * bathroom
    * hallway
    * kitchen
    * living_room
Object-Furniture-Room Relations:
    * casserole_0 on cabinet in kitchen
    * squeezer_0 on cabinet in kitchen
Furniture-Room Relations:
    * table in bathroom
    * table in hallway
    * table in living_room

Instruction: "Get the casserole and squeezer from the kitchen and put them on the table
in the living room."

<step> Source: assistant

[FN] is_on_top("casserole_0", "table in living_room") [/FN]
[FN] is_on_top("squeezer_0", "table in living_room") [/FN]
[END]

<step> Source: user

The initial state is:
Objects:
    * toy_truck_1
    * stuffed_toy_1
    * stuffed_toy_2
Furniture:
    * table
    * bed
    * cabinet
Rooms:
    * bedroom
    * other_room
    * living_room
Object-Furniture-Room Relations:
    * toy_truck_1 on bed in bedroom
    * stuffed_toy_1 on floor in bedroom
    * stuffed_toy_2 on floor in other_room
Furniture-Room Relations:
    * table in living_room

Instruction: "Bring a toy from the bedroom and put it in the living room."

<step> Source: assistant

[FN] is_in_room("toy_truck_1 or stuffed_toy_1", "living_room") [/FN]
[END]

<step> Source: user

The initial state is:
Objects:
    * cushion_1
    * fork_1
    * plate_1
Furniture:
    * bed
    * couch
```

```
        * counter
Rooms:
    * bedroom
    * living_room
Object-Furniture-Room Relations:
    * cushion_1 on couch in living_room
    * fork_1 on couch in living_room
    * plate_1 on couch in living_room
Furniture-Room Relations:
    * bed in bedroom
    * couch in living_room
    * counter in kitchen

Instruction: "Bring the dishes back to the kitchen counter. After that, put the cushion
on the bed."

<step> Source: assistant

[FN] is_on_top("plate_1", "counter in kitchen") [/FN]
[FN] is_on_top("fork_1", "counter in kitchen") [/FN]
[FN] is_on_top("cushion_1", "bed in bedroom") [/FN]
[END]

<step> Source: user

The initial state is:
Objects:
    * candle_0
    * fork_0
    * toy_food_0
Furniture:
    * counter
    * table
Rooms:
    * bedroom
    * kitchen
    * living_room
Object-Furniture-Room Relations:
    * candle_0 on table in living_room
    * fork_0 on counter in kitchen
    * toy_food_0 on counter in kitchen
Furniture-Room Relations:
    * table in bedroom
    * counter in kitchen
    * table in living_room

Instruction: "Put the toy food on the table where the candle is, and place the fork on
the table beside the toy food."

<step> Source: assistant

[FN] is_on_top("toy_food_0", "table in living_room") [/FN]
[FN] is_on_top("fork_0", "table in living_room") [/FN]
[FN] is_next_to("fork_0", "toy_food_0") [/FN]
[END]

<step> Source: user

Objects:
    * toy_airplane_0
    * toy_airplane_1
Furniture:
    * table
Rooms:
    * hallway
    * living_room
Object-Furniture-Room Relations:
    * toy_airplane_0 on table in hallway
    * toy_airplane_1 on table in hallway
Furniture-Room Relations:
    * table in bathroom
    * table in hallway
    * table in living_room

Instruction: "Bring a toy airplane to the living room and put it in on the table."

<step> Source: assistant

[FN] is_on_top("toy_airplane_0 or toy_airplane_1", "table in living_room") [/FN]
```

```
[END]

<step> Source: user

Objects:
    * kettle_0 (empty, dirty)
Furniture:
    * table
Rooms:
    * living_room
Object-Furniture-Room Relations:
    * kettle_0 on table in living_room
    * toy_airplane_0 on table in hallway
    * toy_airplane_1 on table in hallway
Furniture-Room Relations:
    * table in living_room

Instruction: "First, make sure the kettle is clean. Then, fill the kettle with water and
turn it on."

<step> Source: assistant

[FN] is_clean("kettle_0") [/FN]
[FN] is_filled("kettle_0") [/FN]
[FN] is_powered_on("kettle_0") [/FN]
[FN] is_on_top("toy_airplane_0 or toy_airplane_1", "table in living_room") [/FN]
[END]

{TEMPLATE_EXAMPLE}

<step> Source: user

The initial state is:
{INIT_STATE}

Instruction: "{INSTRUCTION}"

<step> Source: assistant
Destination: user

[FN]
```

### Evaluation Generation: Temporal

```
Source: system

You will be given an instruction describing a task to perform in a house and a set of
propositions that define whether the task was done successfully. The task instruction
may say that certain propositions should be completed before others ("then", "after",
"finally"). Your job is to write python code that groups the propositions in the order
in which they must be completed.

The propositions are well-defined python functions that return a boolean value.

You will be given a list of propositions where index i corresponds to the ith
proposition. To solve the task, define the variable proposition_order, which groups
propositions together that can be completed in any order. Each proposition group
appearing in proposition_order must be satisfied before the group that comes after it.
For example,

proposition_order = [
    [0, 1]
]

means that propositions 0 and 1 can be completed in any order. This example

proposition_order = [
    [0],
    [1]
]

means that the proposition 0 must be completed before proposition 1. This example

proposition_order = [
    [0, 1],
    [2]
]
```

```
means that propositions 0 and 1 can be completed in either order, but proposition 2 must
be completed after.

Start by assuming that propositions can be completed in any order. Order matters if the
instruction includes time ordering words such as "then", "finally", or "after". In this
case, propositions should be in multiple groups.

Double check that the index for each proposition is included in proposition_order.

<step> Source: user

Instruction: "Bring an apple to the kitchen table, then bring an orange to the kitchen
counter."

propositions = [
    is_on_top(["apple_1"], ["table_4"]),
    is_on_top(["orange_1"], ["counter_0"])
]

<step> Source: assistant

proposition_order = [
    [0],
    [1]
]

<step> Source: user

Instruction: "Put the toy vehicle and the water bottle in the living room. Next, return
the dish to the kitchen."

propositions = [
    is_in_room(["toy_truck_1"], ["living_room_1"]),
    is_in_room(["cup_0"], ["living_room_1"]),
    is_in_room(["bowl_2"], ["kitchen"])
]

<step> Source: assistant

proposition_order = [
    [0, 1],
    [2]
]

<step> Source: user

Instruction: "Put an apple on the bench in the entryway. Also move the broom to the
closet."

propositions = [
    is_on_top(["apple_0", "apple_1"], ["bench_2"]),
    is_inside(["broom_0"], ["closet_0"])
]

<step> Source: assistant

proposition_order = [
    [0, 1]
]

<step> Source: user

Instruction: "Bring me the toy truck from the bedroom and put it in the living room.
Then put two apples on the kitchen table."

propositions = [
    is_in_room(["toy_truck_1"], ["living_room_1"]),
    is_on_top(["apple_1"], ["table_1"]),
    is_on_top(["apple_2"], ["table_1"])
]

<step> Source: assistant

proposition_order = [
    [0],
    [1, 2]
]
```

```
<step> Source: user

Instruction: "Bring the dishes back to the kitchen counter. After that, put the cushions
on the bed."

propositions = [
    is_on_top(["plate_1"], ["counter_1", "counter_2", "counter_3"]),
    is_on_top(["fork_1"], ["counter_1", "counter_2", "counter_3"]),
    is_on_top(["spoon_0"], ["counter_1", "counter_2", "counter_3"]),
    is_on_top(["cushion_0"], ["bed_1", "bed_2"]),
    is_on_top(["cushion_1"], ["bed_1", "bed_2"]),
]

<step> Source: assistant

proposition_order = [
    [0, 1, 2],
    [3, 4]
]

<step> Source: user

Instruction: "Bring the dishes back to the kitchen counter. Put the cushion on the bed.
Then, move the cushions to the kitchen."

propositions = [
    is_on_top(["plate_1"], ["counter_1", "counter_2", "counter_3"]),
    is_on_top(["fork_1"], ["counter_1", "counter_2", "counter_3"]),
    is_on_top(["cushion_0"], ["bed_1", "bed_2"]),
    is_on_top(["cushion_1"], ["bed_1", "bed_2"]),
    is_in_room(["cushion_0"], ["bedroom_0"]),
    is_in_room(["cushion_1"], ["bedroom_0"]),
]

<step> Source: assistant

proposition_order = [
    [0, 1, 2, 3],
    [4, 5]
]

<step> Source: user

Instruction: "Move the clothes from the bedroom to the washer. After that, Put the
cushion on the bed. Finally, put the book in the living room."

propositions = [
    is_on_top(["shirt_1"], ["washer_dryer_1"]),
    is_on_top(["shirt_2"], ["washer_dryer_1"]),
    is_on_top(["pants_1"], ["washer_dryer_1"]),
    is_on_top(["cushion_1"], ["bed_1", "bed_2"]),
    is_in_room(["book_1"], ["living_room_1"])
]

<step> Source: assistant

proposition_order = [
    [0, 1, 2],
    [3],
    [4]
]

<step> Source: user

Instruction: "First, move the spoon and kettle from the kitchen to the living room and
place them next to each other. Then, place the toy food in the kitchen cabinet."

propositions = [
    is_on_top(['spoon_0'], ['table_1', 'table_2', 'table_3', 'table_4', 'table_5']),
    is_on_top(['kettle_0'], ['table_1', 'table_2', 'table_3', 'table_4', 'table_5']),
    is_next_to(['spoon_0'], ['kettle_0']),
    is_inside(['toy_food_0'], ['cabinet_0'])
]

<step> Source: assistant

proposition_order = [
    [0, 1, 2],
    [3]
```

```
]

<step> Source: user

Instruction: "First, move the phone stand from the bedroom to the living room and place
it on the table next to the lamp. Then, move the file sorter from the living room to the
bedroom and place it on the table next to the phone stand."

propositions = [
    is_on_top(['phone_stand_0'], ['table_1', 'table_2', 'table_3']),
    is_next_to(['phone_stand_0'], ['lamp_0']),
    is_on_top(['file_sorter_0'], ['table_6']),
    is_next_to(['file_sorter_0'], ['phone_stand_0'])
]

<step> Source: assistant

proposition_order = [
    [0, 1],
    [2, 3]
]

<step> Source: user

Instruction: "Help me move the candle and hand towel to the kitchen counter. Place them
next to each other. Then, place
the spatula and c-clamp on the bedside table next to each other."

propositions = [
    is_on_top(['candle_0'], ['counter_0']),
    is_on_top(['hand_towel_0'], ['counter_0']),
    is_next_to(['candle_0'], ['hand_towel_0']),
    is_on_top(['spatula_0'], ['table_6']),
    is_on_top(['c-clamp_0'], ['table_6']),
    is_next_to(['spatula_0'], ['c-clamp_0'])
]

<step> Source: assistant

proposition_order = [
    [0, 1, 2],
    [3, 4, 5]
]

<step> Source: user

Instruction: "First, move the dog bowl, then the placemat, and finally a plush toy from
the living room to the bench in the hallway. Place them next to each other."

propositions = [
    is_on_top(['dog_bowl_0'], ['bench_0']),
    is_on_top(['placemat_0'], ['bench_0']),
    is_next_to(['placemat_0'], ['dog_bowl_0']),
    is_on_top(['plush_toy_0'], ['bench_0']),
    is_next_to(['plush_toy_0'], ['placemat_0'])
]

<step> Source: assistant

proposition_order = [
    [0],
    [1, 2],
    [3, 4]
]

<step> Source: user

Instruction: "Move the toaster and the bread from the pantry to the kitchen and turn the
toaster on, then fill the kettle."

propositions = [
    is_in_room(['toaster_0'], ['kitchen_0']),
    is_in_room(['bread_0'], ['kitchen_0']),
    is_powered_on(['toaster_0']),
    is_filled(['kettle_0'])
]

<step> Source: assistant
```

```
proposition_order = [
    [0, 1, 2],
    [3]
]

<step> Source: user

Instruction: "{INSTRUCTION}"

{PROPOSITIONS}

<step> Source: assistant
Destination: user

proposition_order = [
```

---

**Evaluation Generation: Argument Constraints**

```
Source: system

You will be given a task to perform in a house, and a set of propositions that define
whether the task was done successfully. The task is performed by a human and robot. The
task instruction may imply constraints such that certain groups of propositions should
be satisfied by the same argument or unique arguments. Your job is to write python code
that defines these constraints.

The propositions are well-defined python functions that return a boolean value.

You will be given a list of propositions where index i corresponds to the ith
proposition. To solve the task, define the variable tie_constraints, which is a list of
constraints which can be empty. The constraints you can use are:

SameArgConstraint(
    proposition_indices: List[int],  # indices of propositions that this constraint
    applies to
    arg_index: List[int],  # indices of arguments that should be matched on
)

DifferentArgConstraint(
    proposition_indices: List[int],  # indices of propositions that this constraint
    applies to
    arg_index: List[int],  # indices of arguments that should be matched on
)

Here are some examples:

SameArgConstraint([0, 1], [0, 0])  # means that propositions at index 0 and 1 must have
a matching value in the first argument.
DifferentArgConstraint([0, 1], [0, 0])  # means that propositions at index 0 and 1 must
have different values in the first argument.
SameArgConstraint([0, 1], [1, 1])  # means that propositions at index 0 and 1 must have
a matching value in the second argument.

If no constraints apply to the given instruction, just write an empty list.

<step> Source: user

Instruction: "Bring an apple and an orange to a table in the kitchen."

propositions = [
    is_on_top(["apple_1"], ["table_3", "table_4"]),
    is_on_top(["orange_1"], ["table_3", "table_4"])
]

<step> Source: assistant

tie_constraints = [
    SameArgConstraint([0, 1], [1, 1])
]

<step> Source: user

Instruction: "Put the toy vehicle in the living room and return the dish to the kitchen."

propositions = [
    is_in_room(["toy_truck_1"], ["living_room_1"]),
    is_in_room(["bowl_2"], ["kitchen"])
```

```
]

<step> Source: assistant

tie_constraints = [
]

<step> Source: user

Instruction: "Place the book on the shelf in the bedroom. Place the picture frame next
to it."

propositions = [
    is_on_top(["book_1"], ["shelves_0", "shelves_1"]),
    is_on_top(["picture_frame_0"], ["shelves_0", "shelves_1"]),
    is_next_to(["picture_frame_0"], ["book_1"])
]

<step> Source: assistant

tie_constraints = [
    SameArgConstraint([0, 1], [1, 1])
]

<step> Source: user

Instruction: "Place each candle on its own table in the living room."

propositions = [
    is_on_top(["candle_0"], ["table_0", "table_2", "table_6"]),
    is_on_top(["candle_1"], ["table_0", "table_2", "table_6"]),
    is_on_top(["candle_2"], ["table_0", "table_2", "table_6"])
]

<step> Source: assistant

tie_constraints = [
    DifferentArgConstraint([0, 1, 2], [1, 1])
]

<step> Source: user

Instruction: "Move the clothes from the bedroom to the washer. After that, Put the
cushion on the bed. Finally, put the book in the living room."

propositions = [
    is_on_top(["shirt_1"], ["washer_dryer_1"]),
    is_on_top(["shirt_2"], ["washer_dryer_1"]),
    is_on_top(["pants_1"], ["washer_dryer_1"]),
    is_on_top(["cushion_1"], ["bed_1", "bed_2"]),
    is_in_room(["book_1"], ["living_room_1"])
]

<step> Source: assistant

tie_constraints = [
]

<step> Source: user

Instruction: "{INSTRUCTION}"

{PROPOSITIONS}

<step> Source: assistant
Destination: user

tie_constraints = [
```

## A.15 PROMPTS FOR PLANNING BASELINES

Following prompts were used for various planning baselines.

**Decentralized Single/Multi Agent | ReAct**

```
<|start_header_id|>system<|end_header_id|>You are an agent that solves multi-agent
planning problems. The task assigned to you will be situated in a house and will
generally involve navigating to objects, picking and placing them on different
receptacles to achieve rearrangement. You strictly follow any format specifications and
pay attention to the previous actions taken in order to avoid repeating mistakes. Rooms
do not need to be explored more than once.

There will be another agent trying to solve the same task that you are at the same time.
You may find that that agent has picked up relevant objects or is in the process of
completing parts of the task. If that is the case you may want to move on to a different
part of the task.

Rooms do not need to be explored more than once.
This means if you have explored the living room and have not found the object, then you
should explore the kitchen, if a relevant object is still not found, you should explore
the hallway etc...

{agent_role_description}

Many calls to the same action in a row are a sign that something has gone wrong and you
should try a different action.<|eot_id|>{optional_rag_examples}
<|start_header_id|>user<|end_header_id|>Task: {task_description}

{world_description}

Possible Actions:
{tool_descriptions}

What is the next action to make progress towards completing the task?
Return your response in the following format

Thought: <reasoning for why you are taking the next action>
<next action call>
Assigned!

Here is an example:
Thought: Since there are no objects found I should explore a room I have not explored yet
Explore[<room name>]
Assigned!
<|eot_id|><|start_header_id|>assistant<|end_header_id|>
```

---

**Centralized | ReAct**

```
<|start_header_id|>system<|end_header_id|>You are a system that solves multi-agent
planning tasks. The task assigned to you will be situated in a house and will generally
involve navigating to objects, picking and placing them on different receptacles to
achieve rearrangement. There will be a robot agent (Agent 0) and a human agent (Agent 1)
available for solving the task. Your goal is to assign actions for both of these agents
and solve the task. You strictly follow any format specifications and pay attention to
the previous actions taken in order to avoid repeating mistakes.

You should try and divide the task between the two agents for efficient task completion.
Note that the human agent can wash, clean, fill, pour and turn on/off devices along with
doing object rearrangement. However, the robot can only do object rearrangement i.e.,
navigating to objects, picking and placing them.

In the beginning, you will be provided with the task description and information about
the rooms plus furniture in each room for the house. Object information may or may not
be available. Rooms only need to be explored if there is no information available about
task-relevant objects. Rooms do not need to be explored for identifying which furniture
to to go to. Also, rooms do not need to be explored more than once. This means if one of
your agents has explored the living room and have not found the object, then you should
explore the kitchen, if a relevant object is still not found, you should explore the
hallway etc.

Many calls to the same action in a row are a sign that something has gone wrong and you
should try a different action.

You should try to complete the task in the least amount of actions possible. This means
if there are two objects to be moved you should have one agent navigate to each object
and then move them to the target location a the same time.

If a previous navigation action is still in progress for an agent, you should reassign
that action to the agent till a successful execution is observed in the agent's
observations.
```

```
You should continue to evaluate the task progress and decide the actions for both the
agents. Once both agents are done, you can output "Done[]" to indicate that the agents
have finished the task. Output your response about task completion in the following
format.

Thought: <reasoning about why both agents have completed the entire task successfully>
Done[]

DO NOT output "Done[]" unless you are confident that the whole task is successfully
completed. If one of the agent is done with its part of the task, while the other agent
is still executing, you can assign a "Wait[]" action to the agent who doesnt need to act
anymore. Please re-state the task description and verify it's completion before
outputting "Done[]".{eot_tag}{user_tag}Task: {input}

{world_description}

Possible actions for each agent:
{agent_descriptions}

What is the next action for each agent to make progress towards completing the task?
Return your response in the following format

Thought: <reasoning for why you are taking the next action>
Agent_0_Action: <next action call for robot agent>
Agent_1_Action: <next action call for human agent>
Assigned!

Here is an example:
Thought: Since there are multiple task-relevant objects to be rearranged, I will ask
each agent to go to one of them
Agent_0_Action: Navigate[<obj name1>]
Agent_1_Action: Navigate[<obj name2>]
Assigned!
<|eot_id|><|start_header_id|>assistant<|end_header_id|>
```

The agent role description would be one of the following, depending on if the agent played the role of the human or robot.

## Agent Role Descriptions

### Human Description

```
You are playing the role of the task giver. This means if the instruction says something
like "You should move the object and I will wash it", then the other agent should be
moving the object, and you should washing the it.
```

### Robot Description

```
You are playing the role of the task receiver. This means if the instruction says
something like "You should move the object and I will wash it", then you should move the
object and the other agent should wash it
```

Below are the tool descriptions. Perceptions tools are only included for the React-Tools agents.

## Tool Descriptions

```
- Close: Used for closing an articulated entity. You must provide the name of the
furniture you want to close. Example (Close[chest_of_drawers_1])
- Explore: Search a specific room by visiting various receptacles or furnitures in that
room. The input to the skill is the EXACT name of the room to be visited. Use the room
names provided in the house description. This tool exhaustivly explores the specified
room. Example (Explore[kitchen_1])
- FindAgentActionTool: Should be used to find current and past state history of other
agent.
- FindObjectTool: Used to find the exact name/names of the object/objects of interest.
If you want to find names of objects on specific receptacles or furnitures, please
include that in the query. Example (FindObjectTool[toys on the floor] or
FindObjectTool[apples])
- FindReceptacleTool: Used to know the exact name of a receptacle. A receptacle is a
furniture or entity (like a chair, table, bed etc.) where you can place an object.
Example (FindReceptacleTool[a kitchen counter])
- FindRoomTool: Used to know the exact name of a room in the house. A room is a region
in the house where furniture is placed. Example (FindRoomTool[a room which might have
something to eat])
```

```
- Navigate: Used for navigating to an entity. You must provide the name of the entity
you want to navigate to. Example (Navigate[counter_22])
- Open: Used for opening an articulated entity. You must provide the name of the
furniture you want to open. Example (Open[chest_of_drawers_1])
- Pick: Used for picking up an object. You must provide the name of the object to be
picked. The agent cannot hold more than one object at a time. Example (Pick[cup_1])
- Place: Used for placing an object on a target location. You need to provide the name
of the object to be placed, the name of the furniture where it should be placed, spatial
relation ("on" or "within") describing the relation between the object and furniture.
The object to be placed must already be held by the agent (i.e. picked previously). In
addition to these, you can request to place the object near another object. For that you
can optionally provide a spatial constraints ("next_to") and the name of the reference
object. To place next to an object, the reference object must already be on the target
furniture. API tempate - Place[<object_to_be_placed>, <spatial_relation>, <furniture to
be placed on>, <spatial_constraint>, <reference_object>]. spatial_constraint and
reference_object should be set to "None" when necessary.
- Rearrange: Used for moving an object from its current location to the target location.
You need to provide the name of the object to be moved, the name of the furniture where
is should be moved, spatial relation ("on" or "within") describing the relation between
the object and furniture. This will automatically pick the specified object and move to
the target furniture and attempt to place it. In addition to these, you can request to
place the object near another object. For that you can optionally provide a spatial
constraints ("next_to") and the name of the reference object. To place next to an
object, the reference object must already be on the target furniture. API tempate
Rearrange[<object_to_be_moved>, <spatial_relation>, <furniture to be placed on>,
<spatial_constraint>, <reference_object>]. spatial_constraint and reference_object
should be set to "None" when necessary.
- Wait: Used to make agent stay idle for some time. Example (Wait[])
- Done: Used to indicate that the agent has finished the task. Example (Done[])
```

For LLM agents simulating a human, additionally actions which modify the state of objects are available to be called. For centralized baselines two lists of available actions are provided in the agent description. One for the robot (without state-modifying actions) and one for the human (with state-modifying actions).

**Human Only Tool Descriptions**

```
- Clean: Used for cleaning an object. You need to provide the name of the object to
clean.
- Close: Used for closing an articulated entity. You must provide the name of the
furniture you want to close. Example (Close[chest_of_drawers_1])
- Fill: Used for filling an object. You need to provide the name of the object to fill.
- Pour: Used for pouring from one container to another. This skill will pour into the
specified container from whichever container is currently held by the agent.
- PowerOff: Used for turning off a powered object. You need to provide the name of the
object to be turned off.
- PowerOn: Used for turning on a powered object. You need to provide the name of the
object to be turned on.
```

The world description contains a text description all rooms and their contained furniture, along with currently observed objects. Below is an example for one scene:

**World Description Example**

```
Furniture:
bedroom_1: floor_bedroom_1, chair_41, chair_42, bed_49, table_54, chest_of_drawers_72,
chest_of_drawers_73, chest_of_drawers_75, chest_of_drawers_87
closet_1: floor_closet_1, wardrobe_91
living_room_1: floor_living_room_1, chair_13, chair_14, chair_15, chair_16, chair_17,
chair_18, chair_19, chair_20, chair_21, chair_22, couch_26, couch_28, couch_29,
chair_30, stool_31, stool_32, table_38, table_39, table_48, table_50, stand_52,
counter_78
toilet_1: floor_toilet_1, toilet_43
bedroom_2: floor_bedroom_2, bed_23, chair_46, chair_47, table_53, chest_of_drawers_55,
chest_of_drawers_58, chest_of_drawers_59, chest_of_drawers_60, chest_of_drawers_61
bedroom_3: floor_bedroom_3, bed_37, chair_40, chest_of_drawers_74, table_77,
chest_of_drawers_79, chest_of_drawers_80, chest_of_drawers_89, wardrobe_92
bedroom_4: floor_bedroom_4, chair_27, bed_45, table_51, wardrobe_56, wardrobe_57,
chest_of_drawers_82, chest_of_drawers_83, table_84, table_86
kitchen_1: floor_kitchen_1, shelves_24, shelves_25, chair_33, chair_34, chair_35,
chair_36, cabinet_62, cabinet_63, cabinet_64, cabinet_65, cabinet_66, counter_67,
counter_68, counter_69, cabinet_70, cabinet_71, cabinet_76, cabinet_81, unknown_88,
fridge_90
```

```
other_room_1: floor_other_room_1
other_room_2: floor_other_room_2
bathroom_1: floor_bathroom_1, toilet_44
porch_1: floor_porch_1
entryway_1: floor_entryway_1, table_85

The following furnitures have a faucet: cabinet_70
Objects:
No objects found yet
```

For baselines utilizing a summary of the scene, after each action, we insert a text description of objects in the scene (as perceived by the agent so far).

**Scene Summary Example**

```
Thought: Since there are no objects found, I should explore a room I have not explored
yet. I will start with the living room, as it is one of the rooms mentioned in the task,
and it might contain the vase or other relevant objects.
Explore[living_room_1]
Assigned!<|eot_id|><|start_header_id|>user<|end_header_id|>

Result: Successful execution!
Objects: toy_fruits_1: chest_of_drawers_49 in bedroom_4
laptop_cover_2: cabinet_59 in laundryroom_1
cup_3: cabinet_65 in kitchen_1
vase_0: cabinet_65 in kitchen_1<|eot_id|><|start_header_id|>assistant<|end_header_id|>

Thought: Now that we have found the locations of the objects, I should navigate to the
kitchen to pick up the vase, as it is one of the objects mentioned in the task, and it
is located in the kitchen.
Navigate[cabinet_65]
Assigned!<|eot_id|><|start_header_id|>user<|end_header_id|>

Result: Successful execution!
Objects: toy_fruits_1: chest_of_drawers_49 in bedroom_4
laptop_cover_2: cabinet_59 in laundryroom_1
cup_3: cabinet_65 in kitchen_1
vase_0: cabinet_65 in kitchen_1<|eot_id|>
```

For finetuning, we build the prompt by combining the task description, world description and the previous actions for both agents.

**Finetuning Example**

```
Task: I finished reading my book. Please move the book to the bedroom table. You can
move the vase there too?

Current Environment:
{world_description}

Previous actions:
Agent_Action: Navigate[table_25]
Action Result: Successful execution!
Agent_Action: Navigate[vase_0]
Action Result: Successful execution!
Agent_Action: Pick[vase_0]
Action Result: Successful execution!
Agent_Action: Navigate[table_35]
Action Result: Successful execution!
Other_Agent_Action: Pick[book_1]
Agent_Action: Open[table_35]
Action Result: Successful execution!
Agent_Action: Navigate[table_35]
Action Result: Successful execution!
Agent_Action: Place[vase_0, on, table_35, none, none]
Action Result: Successful execution!
Agent_Action: Wait[]
Action Result: Successful Execution!
Other_Agent_Action: Open[table_35]
Other_Agent_Action: Place[book_1, on, table_35, none, none]

Next Agent_Action:<|reserved_special_token_0|>Done[]<end_act>
```

A.16    FURTHER ANALYSIS OF FAILURES

We describe here in more detail some of the common failures of our proposed LLM-based planner. To better illustrate the failures, we also provide example traces for these.

**Failing to maps objects to their corresponding room leads to unnecessary exploration:** Sometimes objects are detected but assigned to an 'unknown' room, when the perception system fails to assign a room. Note that the object and its location is still added to the graph using RGBD observations, but doesn't have the right parent receptacle or room in the graph. Such situations lead to reasoning problems with the LLM which does not realize that the object is still navigable. In the following example, the LLM keeps exploring rooms looking for the kettle, even though it is already in the graph, but in an 'unknown' room.

---

**Room Association Failure**

```
Objects: 122_clock: 94_bed in bedroom.
126_phone_stand: 94_bed in bedroom.
126_plant_saucer: 65_table in bedroom.
126_kettle: unknown in an unknown
room.<|eot_id|><|start_header_id|>assistant<|end_header_id|>

Thought: I successfully explored the hallway, but I didn't find the kettle. Let me try
to explore the kitchen, as it's a more likely place to find a kettle.
Explore[kitchen]
Assigned!<|eot_id|><|start_header_id|>user<|end_header_id|>

Result: Successful execution!
Objects: 122_clock: 94_bed in bedroom.
126_phone_stand: 94_bed in bedroom.
126_plant_saucer: 65_table in bedroom.
126_kettle: unknown in an unknown
room.<|eot_id|><|start_header_id|>assistant<|end_header_id|>

Thought: I successfully explored the kitchen, but I still didn't find the kettle. Let me
try to explore the bathroom, as it's another possible location.
Explore[bathroom]
Assigned!<|eot_id|><|start_header_id|>user<|end_header_id|>

Result: Unexpected failure! – Skill took too long to finish.
Objects: 122_clock: 94_bed in bedroom.
126_phone_stand: 94_bed in bedroom.
126_plant_saucer: 65_table in bedroom.
126_kettle: unknown in an unknown room.
127_toy_airplane: 67_mirror in
bedroom.<|eot_id|><|start_header_id|>assistant<|end_header_id|>

Thought: The exploration of the bathroom timed out. I still haven't found the kettle.
Since I've already explored the hallway, kitchen, and bathroom, my next step would be to
explore the living room. However, I don't see the living room as an option. Let me try
to explore the bedroom again, as it's a known location and I might have missed something.
Explore[bedroom]
Assigned!<|eot_id|>
```

---

**Failures in communication between the high-level and low-level perception systems result in planner failures**: Our system relies on multiple layers of perception, e.g. high-level perception that tracks the state-of-the-world and low-level perception for skills to act and inform the planner of successful or unsuccessful execution. We see lower-level perception sometimes provides the LLM the wrong feedback about skill execution. Even if the high-level perception gives the correct information, the LLM is not able to reconcile these two sources of information, and tends to get confused. In the attached trace, for example, the LLM planner believes the agent has both ramekin and potato in hand and tries to place them together on the same receptacle. In our simulation, the agent is only allowed to grasp one object at a time, but the low-level perception does not send this error message back to the planner correctly.

---

**High-Level and Low-Level Perception Mismatch**

```
Thought: I have successfully navigated to the living room and I can see the ramekin and
potato on the carpet. I will now try to pick up the ramekin.
Pick[88_ramekin]
Assigned!<|eot_id|><|start_header_id|>user<|end_header_id|>
```

```
Result: Successful execution!

88_ramekin: held by the agent. <-- ramekin in agent's hand (correct high-level)
89_potato: 76_carpet in living_room.
90_ramekin: 37_chest_of_drawers in bedroom.
91_cup: 13_chest_of_drawers in bedroom.
92_book: 24_stool in kitchen.
93_potato: 66_potted_plant in
bedroom.<|eot_id|><|start_header_id|>assistant<|end_header_id|>

Thought: I have successfully picked up the ramekin. Now, I will try to pick up the
potato that is on the carpet in the living room.
Pick[89_potato]
Assigned!<|eot_id|><|start_header_id|>user<|end_header_id|>

Result: Successful execution! <-- agents can not pick up multiple objects; wrong
low-level perception and feedback
Objects: 81_cup: 32_shelves in unknown.
88_ramekin: held by the agent. <-- still held by the agent
89_potato: 76_carpet in living_room. <-- not held by the agent
90_ramekin: 37_chest_of_drawers in bedroom.
91_cup: 13_chest_of_drawers in bedroom.
92_book: 24_stool in kitchen.
93_potato: 66_potted_plant in
bedroom.<|eot_id|><|start_header_id|>assistant<|end_header_id|>

Thought: I have successfully picked up the potato. Now, I will navigate to the table in
the living room to place the ramekin and potato.
Navigate[10_chair]
Assigned!<|eot_id|><|start_header_id|>user<|end_header_id|>

Result: Successful execution!
<summary>

Thought: I have successfully navigated to the chair in the living room, but I need to
place the ramekin and potato on the table. Since the task mentions placing them next to
each other on the chair in the dining room, I will navigate to the dining room instead.
Navigate[47_table]
Assigned!<|eot_id|><|start_header_id|>user<|end_header_id|>

Result: Successful execution!
<summary>

Thought: I have successfully navigated to the table in the dining room. Now, I will
place the ramekin on the table.
Place[88_ramekin,  on, 47_table,  next_to, 89_potato]
Assigned!<|eot_id|><|start_header_id|>user<|end_header_id|>

Result: Unexpected failure! - No valid placements found for entity 47_table. <-- no
placements were found since placement asked for object to be placed next-to 89_potato,
while 89_potato is actually on 76_carpet. Planner keeps repeating this placement on
different receptacles until episode runs out.
```

**Failing to associate objects over time reduces the ability to do task progress:** PARTNR specifically requires agents to reason about task progression and the state of the world as objects are moved around by both agents. Even though the robot agent receives observations from both its own sensors and human's sensor, object temporal reassociation is a hard problem. This leads to complex failures even if the objects are correctly identified. For example, in the trace below, the environment actually only has two bees and 1 toy-truck; however the graph is filled with duplicate copies since the agent fails to associate different toy bee detections to the same instance. This leads to the LLM forgetting the context and the goal, resulting in erroneous episode termination as well as increased inefficiency due to trying rearrangements over objects that do not exist.



**Task Progress Failure**



```
Thought: Now that I have navigated to the shelves in the living room, I should pick up
the toy bee.
Pick[130_toy_bee]
Assigned!<|eot_id|><|start_header_id|>user<|end_header_id|>

Result: Successful execution!
Objects: 122_picture_frame: 67_mirror in bedroom.
124_toy_bee: unknown in an unknown room.
```

```
125_toy_bee: 0_carpet in living_room.
126_toy_bee: 0_carpet in living_room.
127_toy_bee: unknown in an unknown room.
128_toy_bee: 53_carpet in living_room.
129_toy_bee: 53_carpet in living_room.
130_toy_bee: held by the agent.
131_toy_bee: 78_shelves in living_room.
132_toy_bee: 32_chest_of_drawers in living_room.
133_toy_bee: 32_chest_of_drawers in living_room.
134_toy_bee: 19_tv in living_room.
135_toy_bee: 78_shelves in living_room.
136_toy_bee: unknown in an unknown room.
137_toy_bee: 78_shelves in living_room.
139_toy_bee: 78_shelves in living_room.
140_cup: 133_chest_of_drawers in bedroom.
141_toy_bee: 32_chest_of_drawers in living_room.
142_toy_bee: 32_chest_of_drawers in living_room.
143_picture_frame: 4_chest_of_drawers in bedroom.
144_picture_frame: 4_chest_of_drawers in bedroom.
145_picture_frame: unknown in an unknown room.
146_picture_frame: unknown in an unknown room.
147_toy_bee: 19_tv in living_room.
148_toy_bee: 32_chest_of_drawers in living_room.
149_toy_bee: 32_chest_of_drawers in living_room.
150_toy_bee: 0_carpet in living_room.
151_toy_bee: 32_chest_of_drawers in living_room.
152_toy_bee: 0_carpet in living_room.
153_toy_bee: 0_carpet in living_room.
154_toy_bee: 32_chest_of_drawers in living_room.
155_toy_fire_truck: 78_shelves in living_room.
154_toy_fire_truck: 19_tv in living_room.
156_toy_fire_truck: held by the other
agent.<|eot_id|><|start_header_id|>assistant<|end_header_id|>
Thought: I have successfully picked up the toy bee from the shelves in the living room.
My task is now complete. I will end the simulation here.
Done[]
```