# OpenReview forum: "PARTNR: A Benchmark for Planning and Reasoning in Embodied Multi-agent Tasks"
_ICLR.cc/2025/Conference — ICLR 2025 Poster_

### Official Review · Reviewer_LbNu · 2024-10-31

**Soundness:** 3
**Presentation:** 3
**Contribution:** 3
**Rating:** 8
**Confidence:** 2

**Summary:**

This work proposes a benchmark for planning and reasoning tasks in human-robot collaboration, designed to study human-robot coordination in household activities. Numerous experiments have been conducted, which indicate some weaknesses of current LLMs.

**Strengths:**

1. The motivation is reasonable and meaningful.
2. I think this is a large-scale benchmark that makes significant contributions to the community due to its open-source nature.
3. The experiments are sufficient and clear.
4. The paper is well-rounded, demonstrating a huge amount of work, and is practical, potentially being very helpful for research in this direction.

**Weaknesses:**

Please refer to the "Questions" section.

**Questions:**

1. Do you need to use a human-in-the-loop tool to filter all tasks? How much time does it take?
2. Is the human in section 4.2.1 using ReAct to simulate? Does every setting in this section use this model to represent a human? Why did you choose this model for simulating human behavior? Why do you think that LLMs should plan for humans?
3. Is there any analysis of failure cases for task completion?

---

### Official Review · Reviewer_akQq · 2024-11-02

**Soundness:** 2
**Presentation:** 4
**Contribution:** 4
**Rating:** 6
**Confidence:** 3

**Summary:**

This paper presents a benchmark for Planning And Reasoning Tasks in humaN-Robot collaboration (PARTNR), designed to measure collaboration between humans and robots in a household activity setting. The benchmark, semi-automatically generated with the help of LLMs, covers four categories of tasks: (1) constraint-free, (2) spatial tasks, (3) temporal tasks, and (4) heterogeneous tasks. Each task in the benchmark includes instructions and evaluation functions. Across the generated 100,000 tasks, the instructions achieved a 90% accuracy rate, evaluation functions reached 92%, and the overall task validity rate was 83% (90% × 92%). In the final benchmark tests on current mainstream models, it was found that fine-tuning a small model can achieve performance comparable to models nine times larger, while being 8.6 times faster at inference. Additionally, the findings reveal that LLM-guided partners decrease human efficiency compared to working solo.

**Strengths:**

The semi-automatic benchmark generation method proposed in this paper is highly insightful and offers a potential solution to the limited data issue in the field of embodied intelligence. The process for generating test tasks is described in detail, with a comprehensive explanation provided in the appendix, showcasing an ingenious approach to task generation. Additionally, the experimental results validate mainstream models, demonstrating the practical applicability of this benchmark and highlighting the considerable room for improvement in current models.

**Weaknesses:**

The success rate of the tasks generated by this method remains relatively low. Although human-guided data refinement was used to clean the tasks, the effectiveness rate is still only 83%, which could lead to complications in practical applications. Furthermore, the paper does not seem to address how to identify the unsuccessful tasks within the dataset. This issue may interfere with measuring embodied intelligence's success rate in tasks. Since the primary contribution of this method lies in generating large volumes of data, if manual filtering is still required to address task inaccuracies, the overall value of this method may be called into question.

**Questions:**

1） The method ultimately generated 100,000 tasks, with a reported success rate of 83%, but it seem unclear whether the unsuccessful tasks are labeled?
2） Additionally, is there any tendency in the types of tasks generated by this method? Robots may excel in certain task categories, so if the types of tasks generated are those that robots are particularly skilled at—or, conversely, struggle with—would this affect the realism of the test results? In other words, do these tasks' distributions realistically reflect the types of tasks humans encounter in daily life?
3） Can the framework generate task patterns that are related to the robot characteristics?
4)   The author mentioned that the fine tuned model performs better than the large model. But the author trained the model on a specific dataset and tested it on the same dataset. Is this possible to cause overfitting？

---

### Official Review · Reviewer_Htsk · 2024-11-03

**Soundness:** 3
**Presentation:** 3
**Contribution:** 2
**Rating:** 6
**Confidence:** 5

**Summary:**

The work is devoted to the creation of a new benchmark for multiagent planning and execution of textual instructions in a home simulator. The authors proposed a pipeline of task generation and success evaluation scripts using LLM. In the first stage, 1K instructions synchronized with the Habitat 3.0 simulator were obtained mainly due to manual markup, and then 100K instructions were obtained by prompting LLama3-70B-Instruct using 1K as examples. A PDDL-like language with predicates and predicate constraints was proposed for evaluating scripts. 1K scripts were manually marked up, which were then used to generate a complete set with an RAG-like mechanism for the same LLama3-70B-Instruct. The well-known ReAct model and the heuristic expert planner were tested on the resulting dataset. It is shown that centralization, full observability, and accurate execution of skills gives the best result, but there is still a significant gap in the coordination of actions and in the grounding of information.

**Strengths:**

The article is written in clear language, the text is easy to read, and the idea and experimental setting are well described. An important gap in existing datasets for collaborative interaction with a large number of scenes, objects, and a variety of instructions is filled. The resulting dataset will clearly be in demand in the community. The authors use the latest available methods, models, and data: LLama3, Habitat 3, HSSD, OVMM, ReACT, and ConceptGraphs.

**Weaknesses:**

It is not quite clear why the authors specified “applications to robotics, autonomy, planning” as the primary area. There are no new planning methods or robotics applications in the paper, all the baselines used are known. Clearly, the primary topical area should have been “benchmarks and datasets”, within which the work should be evaluated. It is from this point of view that the shortcomings of the work are listed below.

1. In general, the data generation pipelines proposed by the authors cannot be called new. The use of LLM is also found in the works mentioned by the authors both in terms of the generation of the instruction set itself and in terms of the generation of validation scripts. At the same time, the authors do not overcome the main limitation of these approaches - a large amount of manual work of annotation operators. The contribution of LLM is still only related to data scaling, which still requires time-consuming manual validation. In this sense, the work lacks novelty in the sense of data generation pipelines and stands out only for its volume, which is largely obtained by using previously created diverse scenes and collections of objects: HSSD and OVMM.
2. Using a simulator to ground concepts from instructions is reasonable, but this idea is not extended to automatic validation of successful execution of instructions in this environment, for which a heuristic planner could be used. Correctness of instructions and evaluation scripts still relies on manual spot-checking.
3. The authors consider only one single learnable baseline - ReAct, so it is hard to call it a benchmark. There are a huge number of planners for embodied environments, including multi-agent ones [1], with special grounding per scene [2] and with error analysis from lower-level actions [3]. Therefore, the conclusions that the authors draw, based on only one baseline, that this dataset is challenging for state-of-the-art models are not well-founded.
4. It is not clear whether there were instructions that explicitly stated which agent, and which actions should be performed. Such a set of instructions could have been useful for assessing the model's ability to assign roles to a task.
5. The active involvement of human experts requires additional evaluation of the quality of the tasks they perform, both in relation to the generation of instructions and in relation to the manual checking of the correctness of scripts. What criteria were used by the annotation operators and how was their work checked?

[1] https://ieeexplore.ieee.org/abstract/document/10504634/
[2] https://arxiv.org/abs/2310.13255
[3] https://arxiv.org/abs/2307.00329

**Questions:**

1. How serious were ConceptGraph's errors? Can we give an additional metric of its errors in object detection and description to assess whether LLM agents suffer from even small perceptual errors?
2. What criteria were used by the annotation operators and how was their work checked?

---

> ### Comment · Reviewer_Htsk · 2024-11-25
>
> I thank the authors for their detailed responses to my questions and comments. I note that the authors have added three new variations of the ReAct baseline, which at least somewhat expands the set of methods tested on the collected dataset. However, the main drawback of the paper - the lack of novelty in the use of LLM in data collection pipelines - remains. Yes, the authors proposed to do the grounding in the simulator, but from the point of view of the method - this is not a new idea, and in the papers I presented it has already been used. I cannot share the very positive attitude of other reviewers, but I am not against accepting this paper and I am ready to raise my score.

---

### Official Review · Reviewer_63mf · 2024-11-04

**Soundness:** 3
**Presentation:** 4
**Contribution:** 4
**Rating:** 8
**Confidence:** 4

**Summary:**

This paper presents PARTNR, a benchmark for human-robot collaboration. It focuses on evaluating LLM high-level planning.

The paper positions itself against language-based embodied AI benchmarks that don't have interaction, embodied multi-agent benchmarks that have simple, short-horizon environments and closed task sets, and generated-task benchmarks that lack its scale, breadth, and simulator-in-the-loop. It also reviews LLMs for decision making and highlights methods it then uses as baselines.

It then describes the design elements of PARTNR:
- Four task categories with different types of constraints
- 1k human-verified instructions and corresponding evaluation functions, and a prompting + manual annotation pipeline that then leads to 100k total task-evaluation function pairs
- Methods for generating all of these
- Dataset train, validation, and test splits

Finally, the paper goes through significant experimentation investigating the nature of PARTNR tasks. It tests various LLM planning methods in multiple settings, then discusses various conclusions about the nature of PARTNR tasks - difficult for current methods, legitimately encouraging collaboration and coordination, and doable by humans.

**Strengths:**

### Quality
- Well-designed benchmark. The task types, generation pipeline (especially sim-in-the-loop verification), and comparative experiments are clever and useful
- Centralized/decentralized baseline is helpful - even though the centralized method wouldn't make sense as a method benchmarked on PARTNR, using it in comparison to the decentralized method does make an important point about the nature of the tasks. In fact it seems more like an ablation than a benchmark, but point being, it's a good experiment showing that collaboration itself (to the degree you can isolate it when considering generative AI) matters for PARTNR tasks.

### Clarity
- Very well-written paper!

### Originality
As far as I am aware, this is the first multi-agent benchmark of this scale, and with attention paid to these particular aspects. The use of LLM planning modules simplifies the work in many ways - since the methods being used aren't complicated robotic systems, the failures are more constrained to being in semantic space and likely being related to multi-agent coordination, i.e. the benchmark seems especially likely to do what it claims to do.

## Significance
This is a considerable work that will likely enable useful research.

**Weaknesses:**

### Clarity
- More details would help in the main text. I realize benchmark papers require a lot of material, but it would help to have some examples of
  - Human annotation
  - Action selection trajectories - I want to know what it looks like when there's a perception failure, for example
- At times, the writing feels like a laundry list. E.g. section 3.3 - starts to feel like a laundry list of details about the benchmark, without clear direction on why we should care. Dependent rearrangements and multi-step rearrangements of the same object are both interesting, but they aren't brought up again in detail
- Some technical details could use reminder/explanation in main text. E.g.:
  - "LLMs are unable to recover from skill failures" - what exactly is communicated to the LLM here? Is it the new scene graph that doesn't have the intended outcome? What does the failure look like?
  - LLMs struggle to correct errors such as misclassification, making them sensitive to errors in perception - this seems obvious. I'd want to see a trace to know how that plays out, and what this says about *PARTNR* as opposed to the fact that LLMs can't do visual reasoning

### Quality
Overall a convincing paper. Some examples in the main text would help.

**Questions:**

- How is "collaboration data [from] the web" used? I may have missed this
- Does performance improve with more sophisticated collaboration strategies? With a more nuanced understanding of the partner's progress? If not, what's the need for learning these skills?
- How repetitive is the benchmark, fundamentally? Definitely within the 100k, but even within the seed 10k - are there many tasks that require the same exact action trajectories, just with parameters like object ID switched out?

---

### Meta-Review · Area_Chair_UQcm · 2024-12-18

**Metareview:**

The paper presents PARTNR, a large-scale benchmark for studying human-robot collaboration in household activities through planning and reasoning tasks. The work received generally positive reviews, with scores ranging from 6 to 8, and all reviewers recommending acceptance after thorough discussion. Reviewers consistently praised the well-designed benchmark creation pipeline, comprehensive experimental validation, and potential impact on advancing research in human-robot collaboration.

The key strengths highlighted across reviews were the benchmark's scale (100,000 tasks), the innovative simulation-in-the-loop verification approach, and thorough empirical analysis revealing limitations of current LLM-based methods. Reviewers also appreciated the paper's clear presentation and practical significance for the robotics community.

Initial concerns focused on three main areas: the novelty of the LLM-based data generation pipeline, limited baseline comparisons, and lack of detailed analysis of perception failures and task characteristics. The authors provided comprehensive responses, clarifying their novel contributions in grounding LLM outputs through simulation, introducing additional VLM-based baselines, and providing extensive analysis of perception system errors and their impact on task performance. They also clarified the human annotation process and provided detailed failure case analyses.

While some reviewers maintained reservations about the novelty of using LLMs for data generation, they acknowledged that the overall contribution, particularly the simulation-grounded verification and comprehensive evaluation, merits acceptance. The thorough empirical analysis and potential impact on the field outweigh these limitations.

**Additional Comments On Reviewer Discussion:**

None -- see metareview.

---

### Decision · Program_Chairs · 2025-01-22

Accept (Poster)